# On the Linear Speedup of Personalized Federated Reinforcement Learning with Shared Representations

**Guojun Xiong**[1][*]**, Shufan Wang**[2]**, Daniel Jiang**[3]**, Jian Li**[2]
[1]Harvard University, [2]Stony Brook University, [3]Meta

## Abstract

Federated reinforcement learning (FedRL) enables multiple agents to collaboratively learn a policy without needing to share the local trajectories collected during agent-environment interactions. However, in practice, the environments faced by different agents are often heterogeneous, but since existing FedRL algorithms learn a single policy across all agents, this may lead to poor performance. In this paper, we introduce a *personalized* FedRL framework (PFEDRL) by taking advantage of possibly shared common structure among agents in heterogeneous environments. Specifically, we develop a class of PFEDRL algorithms named PFEDRL-REP that learns (1) a shared feature representation collaboratively among all agents, and (2) an agent-specific weight vector personalized to its local environment. We analyze the convergence of PFEDTD-REP, a particular instance of the framework with temporal difference (TD) learning and linear representations. To the best of our knowledge, we are the first to prove a linear convergence speedup with respect to the number of agents in the PFEDRL setting. To achieve this, we show that PFEDTD-REP is an example of federated two-timescale stochastic approximation with Markovian noise. Experimental results demonstrate that PFEDTD-REP, along with an extension to the control setting based on deep Q-networks (DQN), not only improve learning in heterogeneous settings, but also provide better generalization to new environments.

## 1 Introduction

Federated reinforcement learning (FedRL) (Nadiger et al., 2019; Liu et al., 2019; Xu et al., 2021; Zhang et al., 2022a; Jin et al., 2022; Khodadadian et al., 2022; Yuan et al., 2023; Salgia & Chi, 2024; Woo et al., 2024; Zheng et al., 2024; Lan et al., 2024) has recently emerged as a promising framework that blends the distributed nature of federated learning (FL) (McMahan et al., 2017) with reinforcement learning's (RL) ability to make sequential decisions over time (Sutton & Barto, 2018). In FedRL, multiple agents collaboratively learn *a single policy* without sharing individual trajectories that are collected during agent-environment interactions, protecting each agent's privacy.

One key challenge facing FedRL is *environment heterogeneity*, where the collected trajectories may vary to a large extent across agents. To illustrate, consider a few existing applications of FL: on-device NLP applications (e.g., next word prediction, sentence completion, web query suggestions, and speech recognition) from Internet companies (Hard et al., 2018; Yang et al., 2018; Wang et al., 2023b), on-device recommender or ad prediction systems (Maeng et al., 2022; Krichene et al., 2023), and Internet of Things applications like smart healthcare or smart thermostats (Nguyen et al., 2021; Imteaj et al., 2022; Zhang et al., 2022b; Boubouh et al., 2023). Note that *all of the above* (1) exist in settings with environment heterogeneity (heterogeneous users, devices, patients, or homes) and (2) could potentially benefit from an RL problem formulation.

As a result, if all agents collaboratively learn a single policy, which most existing FedRL frameworks do, the learned policy might perform poorly on individual agents. This calls for the design of a *personalized* FedRL (PFEDRL) framework that can provide personalized policies for agents in different environments. Nevertheless, despite the recent advances in FedRL, the design of PFEDRL

---

[*]This work was done when G. Xiong was a PhD student at Stony Brook University.

Table 1: Comparison of existing FedRL frameworks in terms of noise; environments (Homo: homogeneous, Hetero: heterogeneous); using representation learning (Rep.L) or not; timescale (TS), single or two-TS (two) updates; multiple local updates or not; personalization across agents or not; and with or without linear convergence speedup guarantee.

| Method | Noise | Env. | Rep.L | TS | Local updates | Personalized | Linear speedup |
|---|---|---|---|---|---|---|---|
| FedTD & FedQ (Khodadadian et al., 2022) | *Markov* | *Homo.* | ✗ | *Single* | ✓ | ✗ | ✓ |
| FedTD (Dal Fabbro et al., 2023) | *Markov* | *Homo.* | ✗ | *Single* | ✗ | ✗ | ✓ |
| FedTD (Wang et al., 2023a) | *Markov* | *Hetero.* | ✗ | *Single* | ✓ | ✗ | ✓ |
| QAvg & PAvg (Jin et al., 2022) | *i.i.d.* | *Hetero.* | ✗ | *Single* | ✗ | ✗ | ✗ |
| FedQ (Woo et al., 2023) | *Markov* | *Hetero.* | ✗ | *Single* | ✓ | ✗ | ✓ |
| A3C (Shen et al., 2023) | *Markov* | *Homo.* | ✗ | *Two* | ✗ | ✗ | ✓ |
| FedSARSA (Zhang et al., 2024) | *Markov* | *Hetero.* | ✗ | *Single* | ✓ | ✗ | ✓ |
| **PFEDRL-REP** | ***Markov*** | ***Hetero.*** | ✓ | ***Two*** | ✓ | ✓ | ✓ |

and its performance analysis remains, to a large extent, an open question. Motivated by this, the first inquiry we aim to answer in this paper is:

> *Can we design a* PFEDRL *framework for agents in heterogeneous environments that not only collaboratively learns a useful global model without sharing local trajectories, but also learns a personalized policy for each agent?*

We address this question by viewing the PFEDRL problem in heterogeneous environments as $N$ parallel RL tasks with possibly *shared common structure*. This is inspired by observations in centralized learning (Bengio et al., 2013; LeCun et al., 2015) and federated or decentralized learning (Collins et al., 2021; Tziotis et al., 2023; Xiong et al., 2024), where leveraging shared (low-dimensional) representations can improve performance. A theoretical understanding of using shared representations amongst heterogeneous agents has received recent emphasis in the standard *supervised* FL (or decentralized learning) setting (Collins et al., 2021; Tziotis et al., 2023; Xiong et al., 2024).

However, a theoretical analysis of PFEDRL with shared representations is more subtle because each agent in PFEDRL collects data by following its own policy (thereby generating a Markovian trajectory) and simultaneously updates its model parameters. This is in stark contrast to the standard FL paradigm, where data is typically collected in an i.i.d. fashion. Our second research question is:

> *How do the shared representations affect the convergence of* PFEDRL *under Markovian noise, and is it possible to achieve an* $N$-*fold linear convergence speedup?*

Despite the recent progress in the standard supervised FL setting (Collins et al., 2021; Tziotis et al., 2023; Xiong et al., 2024), to the best of our knowledge, this question is still open in the context of learning personalized policies in FedRL under Markovian noise (see Table 1). Motivated by these open questions, our main contributions are:

• **PFEDRL-REP framework.** We propose PFEDRL-REP, a new PFEDRL framework with shared representations. PFEDRL-REP learns a global shared feature representation collaboratively among agents through the aid of a central server, along with agent-specific parameters for personalizing to each agent's local environment. The PFEDRL-REP framework can be paired with a wide range of RL algorithms, including both value-based and policy-based methods with arbitrary feature representations.

• **Linear speedup for TD learning.** We then introduce PFEDTD-REP, an instantiation of the above PFEDRL-REP framework for TD learning (Sutton & Barto, 2018). We analyze its convergence in a linear representation setting, proving the convergence rate of PFEDTD-REP to be $\tilde{\mathcal{O}}\big(N^{-2/3}(T+2)^{-2/3}\big)$, where $N$ is the number of agents and $T$ is the number of communication rounds. This implies a *linear convergence speedup* for PFEDTD-REP with respect to the number of agents, a highly desirable property that allows for massive parallelism in large-scale systems. To our knowledge, this is the first linear speedup result for PFEDRL with shared representations under Markovian noise, providing a theoretical answer to the empirical observations in Mnih et al. (2016) that federated versions of RL algorithms yield faster convergence. To show this result, we make use of two-timescale stochastic approximation theory and address the challenges of Markovian noise through a Lyapunov drift approach.

## 2    PROBLEM FORMULATION

In this section, we first review the standard FedRL framework and then introduce our proposed PFEDRL-REP framework, which incorporates personalization and shared representations. Let $N$ and $T$ be the number of agents and communication rounds, respectively. Denote $[N]$ as the set of integers $\{1, \ldots, N\}$ and $\|\cdot\|$ as the $l_2$-norm. We use boldface to denote matrices and vectors.

### 2.1    PRELIMINARIES: FEDERATED REINFORCEMENT LEARNING

A FedRL system with $N$ agents interacting with $N$ independent heterogeneous environments is modeled as follows. The environment of agent $i \in [N]$ is a Markov decision process (MDP) $\mathcal{M}^i = \langle \mathcal{S}, \mathcal{A}, R^i, P^i, \gamma \rangle$, where $\mathcal{S}$ and $\mathcal{A}$ are finite state and action sets, $R^i$ is the reward function, $P^i$ is the transition kernel, and $\gamma \in (0,1)$ is the discount factor. Suppose agent $i$ is equipped with a policy $\pi^i : \mathcal{S} \to \Delta(\mathcal{A})$ (a mapping from states to probability distributions over $\mathcal{A}$). At each time step $k$, agent $i$ is in state $s_k^i$ and takes action $a_k^i$ according $\pi^i(\cdot \,|\, s_k^i)$, resulting in reward $R^i(s_k^i, a_k^i)$. The environment then transitions to a new state $s_{k+1}^i$ according to $P^i(\cdot \,|\, s_k^i, a_k^i)$. This sequence of states and actions forms a Markov chain, the source of the aforementioned Markovian noise. In this paper, this Markov chain is assumed to be unichain, which is known to asymptotically converge to a steady state. We denote the stationary distribution as $\mu^{i, \pi^i}$.

The value of $\pi^i$ in environment $\mathcal{M}^i$ is defined as $V^{i, \pi^i}(s) = \mathbb{E}_{\pi^i} \left[ \sum_{k=0}^{\infty} \gamma^k R^i(s_k^i, a_k^i) \,|\, s_0^i = s \right]$. In realistic problems with large state spaces, it is infeasible to store $V^{i, \pi^i}(s)$ for all states, so function approximation is often used. One example is $V^{i, \pi^i}(s) \approx \mathbf{\Phi}(s)\boldsymbol{\theta}$, where $\mathbf{\Phi} \in \mathbb{R}^{|\mathcal{S}| \times d}$ is a state feature representation and $\boldsymbol{\theta} \in \mathbb{R}^d$ is an unknown low-dimensional weight vector.

One intermediate goal in RL is to estimate the value function corresponding to a policy $\pi$ using trajectories collected from the environment. This task is called *policy evaluation*, and one widely used approach is *temporal difference* (TD) learning (Sutton, 1988). The FedRL version of TD learning is called FedTD (Khodadadian et al., 2022; Dal Fabbro et al., 2023; Wang et al., 2023a), where $N$ agents collaboratively evaluate a single policy $\pi$ by learning a common (non-personalized) weight vector $\boldsymbol{\theta}$, using trajectories collected from $N$ different environments. More precisely, we have $\pi^i \equiv \pi$ and $\boldsymbol{\theta}^i \equiv \boldsymbol{\theta}, \forall i \in [N]$. Given a feature representation $\mathbf{\Phi}(s), \forall s$, this can be formulated as the following optimization problem:

$$\min_{\boldsymbol{\theta}} \frac{1}{N} \sum_{i=1}^{N} \mathbb{E}_{s \sim \mu^{i, \pi}} \left\| \mathbf{\Phi}(s)\boldsymbol{\theta} - V^{i, \pi}(s) \right\|^2 . \tag{1}$$

Due to space constraints, we focus our presentation on the policy evaluation problem. Note that policy evaluation is an important part of RL and control, since it is a critical step for methods based on policy improvement. Our proposed PFEDRL framework (see Algorithm 1) can be directly applied to control problems as well, but we relegate these discussions to Section 5 and Appendix C.

### 2.2    PERSONALIZED FEDRL WITH SHARED REPRESENTATIONS

Since the local environments are heterogeneous across the $N$ agents, the aforementioned FedRL methods (in Section 2.1) that aim to learn a common weight vector $\boldsymbol{\theta}$ may perform poorly on individual agents. This necessitates the search for personalized local weight vectors $\boldsymbol{\theta}^i$ that can be learned collaboratively among $N$ agents in $N$ heterogeneous environments (without sharing their locally collected trajectories). As alluded to earlier, we view the personalized FedRL (PFEDRL) problem as $N$ parallel RL tasks with possibly *shared common structure*, and we propose that the agents collaboratively learn a common features representation $\mathbf{\Phi}$ in addition to a personalized local weights $\boldsymbol{\theta}^i$. Specifically, the value function of agent $i$ is approximated as $V^{i, \pi^i} \approx f^i(\boldsymbol{\theta}^i, \mathbf{\Phi})$, where $f^i(\cdot, \cdot)$ is a general function parameterized by these two *unknown* parameters.[1] The policy

---

[1] The approximation $f^i(\boldsymbol{\theta}^i, \mathbf{\Phi})$ is general and can take on various forms, including as linear approximations or neural networks. For instance, it can be represented as a linear combination of $\mathbf{\Phi}$ and $\boldsymbol{\theta}^i$, i.e., $f^i(\boldsymbol{\theta}^i, \mathbf{\Phi}) := \mathbf{\Phi}\boldsymbol{\theta}^i$ in TD with linear function approximation (Bhandari et al., 2018). In addition, $f^i(\boldsymbol{\theta}^i, \mathbf{\Phi})$ can represent a deep neural network; see, e.g., our extension of PFEDRL to control problems and its instantiation with DQN (deep Q-networks) (Mnih et al., 2015) in Section 5 and Appendix C.

---

**Algorithm 1** PFEDRL-REP: A General Description

---

**Input:** Sampling policy $\pi^i, \forall i \in [N]$;

1: Initialize the global feature representation $\mathbf{\Phi}_0$ and local weight vector $\boldsymbol{\theta}_0^i, \forall i \in [N]$ randomly;
2: **for** round $t = 0, 1, \ldots, T-1$ **do**
3:     **for** agent $1, \ldots, N$ **do**
4:         $\boldsymbol{\theta}_{t+1}^i = \text{WEIGHT\_UPDATE}(\mathbf{\Phi}_t, \boldsymbol{\theta}_t^i, \alpha_t, K)$;
5:         $\mathbf{\Phi}_{t+1/2}^i = \text{FEATURE\_UPDATE}(\mathbf{\Phi}_t, \boldsymbol{\theta}_{t+1}^i, \beta_t)$;
6:     **end for**
7:     Server computes the new global feature representation $\mathbf{\Phi}_{t+1} = \frac{1}{N} \sum_{i=1}^{N} \mathbf{\Phi}_{t+1/2}^i$.
8: **end for**

---

evaluation problem of (1) can be updated for this new setting as:

$$\min_{\mathbf{\Phi}} \frac{1}{N} \sum_{i=1}^{N} \min_{\{\boldsymbol{\theta}^i, \forall i\}} \mathbb{E}_{s \sim \mu^{i,\pi^i}} \left\| f^i(\boldsymbol{\theta}^i, \mathbf{\Phi}(s)) - V^{i,\pi^i}(s) \right\|^2, \tag{2}$$

where $N$ agents collaboratively learn a shared feature representation $\mathbf{\Phi}$ via a server, along with a personalized local weight vector $\{\boldsymbol{\theta}^i, \forall i\}$ using local trajectories at each agent.

*Remark* 2.1. The learning of a shared feature representation $\mathbf{\Phi}$ in PFEDRL is related to ideas from representation learning theory (Agarwal et al., 2020; 2023), and this is believed to achieve better generalization performance with relatively small training data. In conventional FedRL, the feature representation $\mathbf{\Phi}$ is given and fixed. Indeed, as we numerically verify in Section 4.3, our PFEDRL presents better generalization performance to new environments.

## 3 PFEDRL-REP ALGORITHMS

We now propose a class of algorithms called PFEDRL-REP that realize PFEDRL with shared representations. PFEDRL-REP alternates comprises of three main steps for each agent at each communication round: (1) a local weight vector update; (2) a local feature representation update; and (3) a global feature representation update via the server.

**Steps 1 and 2: Local weight and feature representation updates.** At round $t$, agent $i$ performs an update on its local weight vector given its current global feature representation $\mathbf{\Phi}_t$ and local weight vector $\boldsymbol{\theta}_t^i$. We allow each agent to perform $K$ steps of local weight vector updates. Once the updated local weight vector $\boldsymbol{\theta}_{t+1}^i$ is obtained, each agent $i$ executes a one-step local update on its feature representation to obtain $\mathbf{\Phi}_{t+1/2}^i$. We represent these updates using the following generic notation:

$$\boldsymbol{\theta}_{t+1}^i = \text{WEIGHT\_UPDATE}(\mathbf{\Phi}_t, \boldsymbol{\theta}_t^i, \alpha_t, K) \quad \text{and} \quad \mathbf{\Phi}_{t+1/2}^i = \text{FEATURE\_UPDATE}(\mathbf{\Phi}_t, \boldsymbol{\theta}_{t+1}^i, \beta_t), \tag{3}$$

where $\alpha_t$ and $\beta_t$ are learning rates for the weight and feature updates, respectively. The generic functions WEIGHT\_UPDATE and FEATURE\_UPDATE will be specialized to the particulars of the underlying RL algorithm: in Section 4 we discuss the case of TD with linear function approximation in detail, and in Appendix C, we show instantiations of Q-learning and DQN in our framework.

**Step 3: Server-based global feature representation update.**
The server computes an average of the received local feature representation updates $\mathbf{\Phi}_{t+1/2}^i$ from all agents to obtain the next global feature representation $\mathbf{\Phi}_{t+1}$ as in (4).

$$\mathbf{\Phi}_{t+1} = \frac{1}{N} \sum_{i=1}^{N} \mathbf{\Phi}_{t+1/2}^i. \tag{4}$$

The PFEDRL-REP procedure repeats (3) and (4) and is summarized in Algorithm 1 and Figure 1. We emphasize that because PFEDRL-REP operates in an RL setting, there is no ground truth for the value function and learning occurs through interactions with an MDP environment, resulting in non-i.i.d. data. In contrast, in the standard FL setting (where shared representations have been investigated), there exists a known ground truth and training data are sampled in an i.i.d. fashion (Collins et al., 2021; Tziotis et al., 2023; Xiong et al., 2024). The non-i.i.d. (Markovian) data is the main technical challenge that we need to overcome.

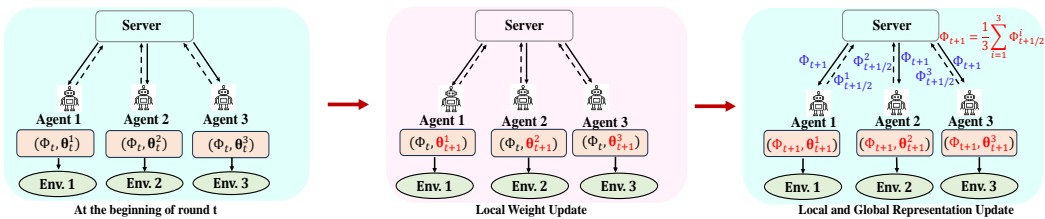

Figure 1: An illustrative example of PFEDRL-REP for 3 agents. (a) At the beginning of round $t$, each agent $i = 1, 2, 3$ has a local weight vector $\boldsymbol{\theta}_t^i$ and a global feature representation $\boldsymbol{\Phi}_t$. (b) Using $(\boldsymbol{\Phi}_t, \boldsymbol{\theta}_t^i)$, each agent $i$ performs a $K$-step update to obtain $\boldsymbol{\theta}_{t+1}^i$ as in (3). Note that $\boldsymbol{\Phi}_t$ remains unchanged at this step. (c) Agent $i$ updates the feature representation by executing a one-step update to obtain $\boldsymbol{\Phi}_{t+1/2}^i$ as in (3), which depends on both $\boldsymbol{\theta}_{t+1}^i$ and $\boldsymbol{\Phi}_t$. Finally, each agent $i$ shares $\boldsymbol{\Phi}_{t+1/2}^i$ with the server, which then executes an averaging step as in (4) to produce $\boldsymbol{\Phi}_{t+1}$. Updated parameters are highlighted in red, while shared parameters (the global feature representation) are in blue.

## 4 PFEDTD-REP WITH LINEAR REPRESENTATION

We present PFEDTD-REP, an instance of PFEDRL-REP paired with TD learning and analyze its convergence in a linear representation setting.

### 4.1 PFEDTD-REP: ALGORITHM DESCRIPTION

Here, the goal of $N$ agents is to collaboratively solve problem (2) when the underlying RL algorithm is TD learning. We first need to specify WEIGHT_UPDATE and FEATURE_UPDATE of Algorithm 1 for the case of TD. At time step $k$, the state of agent $i$ is $s_k^i$, and its value function can be denoted as $V(s_k^i) = \boldsymbol{\Phi}(s_k^i)\boldsymbol{\theta}^i$ in a linear representation setting. By the standard one-step Monte Carlo approximation used in TD, we compute $\hat{V}(s_k^i) = r_k^i + \gamma\boldsymbol{\Phi}(s_{k+1}^i)\boldsymbol{\theta}^i$. The TD error is defined as

$$\delta_k^i := \hat{V}(s_k^i) - V(s_k^i) = r_k^i + \gamma\boldsymbol{\Phi}(s_{k+1}^i)\boldsymbol{\theta}^i - \boldsymbol{\Phi}(s_k^i)\boldsymbol{\theta}^i. \tag{5}$$

The goal of agent $i$ is to minimize the following loss function for every $s_k^i \in \mathcal{S}$

$$\mathcal{L}^i(\boldsymbol{\Phi}(s_k^i), \boldsymbol{\theta}^i) = \frac{1}{2}\big\|V(s_k^i) - \hat{V}(s_k^i)\big\|^2, \tag{6}$$

with $\hat{V}(s_k^i)$ treated as a constant. We now denote the Markovian observations of agent $i$ at the $k$-th time step of communication round $t$ as $X_{t,k}^i := (s_{t,k}^i, r_{t,k}^i, s_{t,k+1}^i)$. Note that the observation sequences $\{X_{t,k}^i, \forall t, k\}$ differ across agents in heterogeneous environments. We assume that $\{X_{t,k}^i, \forall t, k\}$ are statistically independent across all agents.

**Local weight vector update.** As in line 4 of Algorithm 1, given the current global feature representation $\boldsymbol{\Phi}_t$, each agent $i$ takes $K$ local update steps on its local weight vector $\boldsymbol{\theta}_t^i$ as

$$\boldsymbol{\theta}_{t,k}^i = \boldsymbol{\theta}_{t,k-1}^i + \alpha_t\,\mathbf{g}(\boldsymbol{\theta}_{t,k-1}^i, \boldsymbol{\Phi}_t, X_{t,k-1}^i), \tag{7}$$

for $k \in [K]$, where $\mathbf{g}(\boldsymbol{\theta}_{t,k-1}^i, \boldsymbol{\Phi}_t, X_{t,k-1}^i)$ is the negative stochastic gradient of the loss function $\mathcal{L}^i(\boldsymbol{\Phi}_t(s_{t,k-1}^i), \boldsymbol{\theta}_{t,k-1}^i)$ with respect to $\boldsymbol{\theta}$, given the current feature representation $\boldsymbol{\Phi}_t$:

$$\mathbf{g}(\boldsymbol{\theta}_{t,k-1}^i, \boldsymbol{\Phi}_t, X_{t,k-1}^i) := -\nabla_{\boldsymbol{\theta}}\mathcal{L}^i(\boldsymbol{\Phi}_t(s_{t,k-1}^i), \boldsymbol{\theta}_{t,k-1}^i) = \delta_{t,k-1}^i\boldsymbol{\Phi}_t(s_{t,k-1}^i)^\intercal. \tag{8}$$

Since there are $K$ steps of local updates, we denote $\boldsymbol{\theta}_{t+1}^i := \boldsymbol{\theta}_{t,K}^i$. We further add a norm-scaling (i.e., clipping) step for the updated weight vectors $\boldsymbol{\theta}_{t+1}^i$, i.e., enforcing $\|\boldsymbol{\theta}_{t+1}^i\| \leq B$, to stabilize the update. This is essential for the finite-time convergence analysis in Section 4.2, and this technique is widely used in conventional TD learning with linear function approximation (Bhandari et al., 2018).

**Local feature representation update.** As in line 5 of Algorithm 1, given the updated local weight vector $\boldsymbol{\theta}_{t+1}^i$, agent $i$ then executes a one-step local update on the global feature representation:

$$\boldsymbol{\Phi}_{t+1/2}^i = \boldsymbol{\Phi}_t + \beta_t\mathbf{h}(\boldsymbol{\theta}_{t+1}^i, \boldsymbol{\Phi}_t, \{X_{t,k-1}^i\}_{k=1}^K), \tag{9}$$

where $\mathbf{h}(\boldsymbol{\theta}_{t+1}^i, \boldsymbol{\Phi}_t, \{X_{t,k-1}^i\}_{k=1}^K)$ is the negative stochastic gradient of the loss $\mathcal{L}^i(\boldsymbol{\Phi}_t(s_{t,k-1}^i), \boldsymbol{\theta}_{t+1}^i)$ with respect to the current global feature representation $\boldsymbol{\Phi}_t$, satisfying

$$\mathbf{h}(\boldsymbol{\theta}_{t+1}^i, \boldsymbol{\Phi}_t, X_{t,k-1}^i) := -\nabla_{\boldsymbol{\Phi}} \mathcal{L}^i(\boldsymbol{\Phi}_t(s_{t,k-1}^i), \boldsymbol{\theta}_{t+1}^i) = \delta_{t,k-1}^i \boldsymbol{\theta}_{t+1}^{i\mathsf{T}}. \tag{10}$$

**Server-based global feature representation update.** As in line 7 of Algorithm 1, the server then averages the received local feature representation updates in (9) to obtain the next global feature representation:

$$\boldsymbol{\Phi}_{t+1} = \boldsymbol{\Phi}_t + \beta_t \cdot \frac{1}{N} \sum_{j=1}^N \mathbf{h}(\boldsymbol{\theta}_{t+1}^j, \boldsymbol{\Phi}_t, \{X_{t,k-1}^i\}_{k=1}^K). \tag{11}$$

The full pseudo-code of PFEDTD-REP is given in Appendix B.

## 4.2 CONVERGENCE ANALYSIS

The coupled updates in (7) and (11) can be viewed as a federated nonlinear two-timescale stochastic approximation (2TSA) (Doan, 2021) with Markovian noise, with $\boldsymbol{\theta}_t^i$ updating on a faster timescale and $\boldsymbol{\Phi}_t$ on a slower timescale. We aim to establish the finite-time convergence rate of the 2TSA coupled updates (7) and (11). This is equivalent to finding a solution pair $(\boldsymbol{\Phi}^*, \{\boldsymbol{\theta}^{i,*}, \forall i\})$ such that[2]

$$\mathbb{E}_{s_t^i \sim \mu^i, s_{t+1}^i \sim P_{\pi^i}^i(\cdot|s_t^i)}[\mathbf{g}(\boldsymbol{\theta}^{i,*}, \boldsymbol{\Phi}^*, X_t^i)] = 0 \quad \text{and} \quad \mathbb{E}_{s_t^i \sim \mu^i, s_{t+1}^i \sim P_{\pi^i}^i(\cdot|s_t^i)}[\mathbf{h}(\boldsymbol{\theta}^{i,*}, \boldsymbol{\Phi}^*, X_t^i)] = 0 \tag{12}$$

hold for all Markovian observations $X_t^i$. Here, $\mu^i$ is the unknown stationary distribution of state $s_t^i$ of agent $i$ at $t$, and $P_{\pi^i}^i$ is the transition kernel of agent $i$ under policy $\pi^i$.

Although the root $(\boldsymbol{\Phi}^*, \{\boldsymbol{\theta}^{i,*}, \forall i\})$ of the nonlinear 2TSA in (7) and (11) is not unique due to simple permutations (rotations), it is proved in Tsitsiklis & Van Roy (1996) that the standard TD iterates converge asymptotically to a vector $\boldsymbol{\theta}^*$ given a fixed feature representation $\boldsymbol{\Phi}$ almost surely, where $\boldsymbol{\theta}^*$ is the unique solution of a certain projected Bellman equation. Hence, for agent $i$, in order to study the stability of $\boldsymbol{\theta}^i$ when the feature representation $\boldsymbol{\Phi}$ is fixed, we note that there exists a mapping $\boldsymbol{\theta}^i = y^i(\boldsymbol{\Phi})$ that maps $\boldsymbol{\Phi}$ to the unique solution of $\mathbb{E}_{s_t^i \sim \mu^i, s_{t+1}^i \sim P_{\pi^i}^i(\cdot|s_t^i)}[\mathbf{g}(\boldsymbol{\theta}^i, \boldsymbol{\Phi}, X_t^i)] = 0$.

Inspired by Doan (2020), the finite-time analysis of a 2TSA boils down to the choice of two step sizes $\{\alpha_t, \beta_t, \forall t\}$ and a Lyapunov function that couples the two iterates in (7) and (11). We first define the following two error terms:

$$\tilde{\boldsymbol{\Phi}}_t = \boldsymbol{\Phi}_t - \boldsymbol{\Phi}^* \quad \text{and} \quad \tilde{\boldsymbol{\theta}}_t^i = \boldsymbol{\theta}_t^i - y^i(\boldsymbol{\Phi}_t), \ \forall i \in [N], \tag{13}$$

which together characterizes the coupling between $\{\boldsymbol{\theta}_{t+1}^i, \forall i\}$ and $\boldsymbol{\Phi}_t$. If $\{\tilde{\boldsymbol{\theta}}_{t+1}^i, \forall i\}$ and $\tilde{\boldsymbol{\Phi}}_t$ go to zero simultaneously, the convergence of $(\{\boldsymbol{\theta}_{t+1}^i, \forall i\}, \boldsymbol{\Phi}_t)$ to $(\{\boldsymbol{\theta}^{i,*}, \forall i\}, \boldsymbol{\Phi}^*)$ can be established. Thus, to prove the convergence of $(\{\boldsymbol{\theta}_{t+1}^i, \forall i\}, \boldsymbol{\Phi}_t)$ of the 2TSA in (7) and (11) to its true value $(\boldsymbol{\Phi}^*, \{\boldsymbol{\theta}^{i,*}, \forall i\})$, we define the following weighted Lyapunov function to explicitly couple the fast and slow iterates

$$M(\{\boldsymbol{\theta}_{t+1}^i, \forall i\}, \boldsymbol{\Phi}_t) := \|\boldsymbol{\Phi}_t - \boldsymbol{\Phi}^*\|^2 + \frac{\beta_{t-1}}{\alpha_t} \frac{1}{N} \sum_{i=1}^N \|\boldsymbol{\theta}_{t+1}^i - y^i(\boldsymbol{\Phi}_t)\|^2. \tag{14}$$

*Remark* 4.1. Note that the Lyapunov function (14) for 2TSA does not inherently require the solution to be unique. If multiple solutions or equilibria exist, the Lyapunov function should still be able to show that the system will converge to one of these possible equilibria, ensuring that the system's state does not diverge and eventually stabilizes at some equilibrium point, which highly depends on the initialization of $\boldsymbol{\Phi}_0$. To clarify this, in the rest of this paper, we use $\boldsymbol{\Phi}_0^*$ to clearly indicate the dependence of the initialization of $\boldsymbol{\Phi}$, and $\boldsymbol{\Phi}^*$ in (14) is interchangeable with $\boldsymbol{\Phi}_0^*$, which denotes the optimum close to the initial point.

---

[2]The root $(\boldsymbol{\Phi}^*, \{\boldsymbol{\theta}^{i,*}, \forall i\})$ of the nonlinear 2TSA in (7) and (11) can be established by using the ODE method following the solution of suitably defined differential equations (Doan, 2021; 2020; Chen et al., 2019) as in (12).

Our goal is to characterize the finite-time convergence of $\mathbb{E}[M(\{\boldsymbol{\theta}_{t+1}^i, \forall i\}, \boldsymbol{\Phi}_t)]$, the Lyapunov function in (14). We start with some standard assumptions first.

**Assumption 4.2.** The learning rates $\alpha_t$ and $\beta_t$ satisfy the following conditions: (i) $\sum_{t=0}^{\infty} \alpha_t = \infty$, (ii) $\sum_{t=0}^{\infty} \alpha_t^2 < \infty$, (iii) $\sum_{t=0}^{\infty} \beta_t = \infty$, (iv) $\sum_{t=0}^{\infty} \beta_t^2 < \infty$, (v) $\beta_t/\alpha_t$ is non-increasing in $t$, and (vi) $\lim_{t \to \infty} \beta_t/\alpha_t = 0$.

**Assumption 4.3.** Agent $i$'s Markov chain $\{X_t^i\}$ is irreducible and aperiodic. Hence, there exists a unique stationary distribution $\mu^i$ (Levin & Peres, 2017) and constants $C > 0$ and $\rho \in (0, 1)$ such that $d_{TV}(P(X_k^i | X_0^i = x), \mu^i) \le C\rho^k, \forall k \ge 0, x \in \mathcal{X}$, where $d_{TV}(\cdot, \cdot)$ is the total-variation (TV) distance (Levin & Peres, 2017).

*Remark* 4.4. Assumption 4.3 implies that the Markov chain induced by $\pi^i$ admits a unique stationary distribution $\mu^i$. This assumption is commonly used in the asymptotic convergence analysis of stochastic approximation under Markovian noise (Borkar, 2009; Chen et al., 2019).

We can define the steady-state local TD update direction as

$$\bar{\mathbf{g}}(\boldsymbol{\theta}^i, \boldsymbol{\Phi}) := \mathbb{E}_{s_t^i \sim \mu^i, s_{t+1}^i \sim P_{\pi^i}^i(\cdot | s_t^i)}[\mathbf{g}(\boldsymbol{\theta}^i, \boldsymbol{\Phi}, X_t^i)],$$
$$\bar{\mathbf{h}}(\boldsymbol{\theta}^i, \boldsymbol{\Phi}) := \mathbb{E}_{s_t^i \sim \mu^i, s_{t+1}^i \sim P_{\pi^i}^i(\cdot | s_t^i)}[\mathbf{h}(\boldsymbol{\theta}^i, \boldsymbol{\Phi}, X_t^i)]. \tag{15}$$

**Definition 4.5** (Mixing time, similar to Chen et al. (2019)). First, define the discrepancy term

$$\xi_t(\boldsymbol{\theta}^i, \boldsymbol{\Phi}, x) = \max\{\|\mathbb{E}[\mathbf{g}(\boldsymbol{\theta}^i, \boldsymbol{\Phi}, X_t^i) \mid X_0 = x] - \bar{\mathbf{g}}(\boldsymbol{\theta}^i, \boldsymbol{\Phi})\|, \|\mathbb{E}[\mathbf{h}(\boldsymbol{\theta}^i, \boldsymbol{\Phi}, X_t^i) \mid X_0 = x] - \bar{\mathbf{h}}(\boldsymbol{\theta}^i, \boldsymbol{\Phi})\|\}.$$

For $\delta > 0$, the *mixing time* is defined as

$$\tau_\delta = \max_{i \in [N]} \min\{t \ge 1 : \xi_k(\boldsymbol{\theta}^i, \boldsymbol{\Phi}, x) \le \delta(\|\boldsymbol{\Phi} - \boldsymbol{\Phi}^*\| + \|\boldsymbol{\theta}^i - y^i(\boldsymbol{\Phi}^*)\| + 1), \forall k \ge t, \forall (\boldsymbol{\theta}^i, \boldsymbol{\Phi}, x)\},$$

which describes the time it takes for all agents' trajectories (Markov chains) to be well-represented by their stationary distributions.

**Lemma 4.6.** $\mathbf{g}(\boldsymbol{\theta}, \boldsymbol{\Phi}, X)$ *in (8) is globally Lipschitz continuous w.r.t* $\boldsymbol{\theta}$ *and* $\boldsymbol{\Phi}$ *uniformly in* $X$, *i.e.,* $\|\mathbf{g}(\boldsymbol{\theta}_1, \boldsymbol{\Phi}_1, X) - \mathbf{g}(\boldsymbol{\theta}_2, \boldsymbol{\Phi}_2, X)\| \le L_g(\|\boldsymbol{\theta}_1 - \boldsymbol{\theta}_2\| + \|\boldsymbol{\Phi}_1 - \boldsymbol{\Phi}_2\|), \forall X \in \mathcal{X}$.

**Lemma 4.7.** $\mathbf{h}(\boldsymbol{\theta}, \boldsymbol{\Phi}, X)$ *in (10) is globally Lipschitz continuous w.r.t* $\boldsymbol{\theta}$ *and* $\boldsymbol{\Phi}$ *uniformly in* $X$, *i.e.,* $\|\mathbf{h}(\boldsymbol{\theta}_1, \boldsymbol{\Phi}_1, X) - \mathbf{h}(\boldsymbol{\theta}_2, \boldsymbol{\Phi}_2, X)\| \le L_h(\|\boldsymbol{\theta}_1 - \boldsymbol{\theta}_2\| + \|\boldsymbol{\Phi}_1 - \boldsymbol{\Phi}_2\|), \forall X \in \mathcal{X}$.

**Lemma 4.8.** $y^i(\boldsymbol{\Phi}), \forall i$ *is Lipschitz continuous in* $\boldsymbol{\Phi}$, *i.e.,* $\|y^i(\boldsymbol{\Phi}_1) - y^i(\boldsymbol{\Phi}_2)\| \le L_y\|\boldsymbol{\Phi}_1 - \boldsymbol{\Phi}_2\|$.

For notational simplicity, we let $L := \max\{L_g, L_h, L_y\}$ and assume that $L$ is the common Lipschitz constant in Lemmas 4.6-4.8 in the following.

*Remark* 4.9. The Lipschitz continuity of $\mathbf{h}$ guarantees the existence of a solution $\boldsymbol{\Phi}$ to the equilibrium (12) for a fixed $\boldsymbol{\theta}$, while the Lipschitz continuity of $\mathbf{g}$ and $y^i$ ensures the existence of a solution $\boldsymbol{\theta}^i$ of (12) when $\boldsymbol{\Phi}$ is fixed.

**Lemma 4.10.** *There exists a* $\omega > 0$ *such that* $\forall \boldsymbol{\Phi}, \boldsymbol{\theta}$ *and* $\forall i$:

$$\langle \boldsymbol{\Phi} - \boldsymbol{\Phi}_0^*, \bar{\mathbf{h}}(y^i(\boldsymbol{\Phi}), \boldsymbol{\Phi}) \rangle \le -\omega\|\boldsymbol{\Phi}_0^* - \boldsymbol{\Phi}\|^2, \qquad \langle \boldsymbol{\theta}_t^i - y^i(\boldsymbol{\Phi}_{t-1}), \bar{\mathbf{g}}(\boldsymbol{\theta}_t^i, \boldsymbol{\Phi}_{t-1}) \rangle \le -\omega\|\boldsymbol{\theta} - y^i(\boldsymbol{\Phi})\|^2.$$

*Remark* 4.11. Lemma 4.10 guarantees the stability of the two-timescale update in (7) and (11), and can be viewed as the monotone property of nonlinear mappings leveraged in Doan (2020); Chen et al. (2019).

**Lemma 4.12.** *Under Assumption 4.3, and Lemma 4.6 and 4.7, there exist constants* $C > 0$, $\rho \in (0, 1)$ *and* $L_1 = \max(L_g, L_h, \max_X \mathbf{g}(\boldsymbol{\theta}^*, \boldsymbol{\Phi}^*, X), \max_X \mathbf{h}(\boldsymbol{\theta}^*, \boldsymbol{\Phi}^*, X))$ *such that*

$$\tau_\delta \le \frac{\log(1/\delta) + \log(2L_1 Cd)}{\log(1/\rho)} \quad \text{and} \quad \lim_{\delta \to 0} \delta\tau_\delta = 0.$$

### 4.2.1 MAIN RESULTS

We now present our main theoretical results in this work.

**Theorem 4.13.** *Let* $T \ge 2\tau_\delta$ *for some* $\delta > 0$. *Suppose that the learning rates are chosen as*

$$\alpha_t = \alpha_0/(t+2)^{5/6} \quad \text{and} \quad \beta_t = \beta_0/(t+2),$$

*where* $\alpha_0 \leq 1/(2L\sqrt{2(1+L^2)})$, $\beta_0 \leq \omega/2$, *and* $L = \max\{L_g, L_h, L_y\}$. *We have*

$$M(\{\boldsymbol{\theta}_{T+2}^i\}, \boldsymbol{\Phi}_{T+1}) \leq \frac{M(\{\boldsymbol{\theta}_1^i\}, \boldsymbol{\Phi}_0)}{(T+2)^2} + \frac{C_1}{(T+2)^{2/3}}$$

$$+ \frac{C_2}{(T+2)^{2/3}} \left( \mathbb{E}[\|\boldsymbol{\Phi}_0 - \boldsymbol{\Phi}_0^*\|^2] + \frac{1}{N} \mathbb{E} \sum_{i=1}^N \|\boldsymbol{\theta}_1^i - y^i(\boldsymbol{\Phi}_0)\|^2 \right), \quad (16)$$

*with* $C_1 = (4\alpha_0\beta_0 K^2(3\delta^2(1+B^2) + L^2B^2) + 2\alpha_0^2(3K^2B^2 + 3K^2\delta^2 + 2L^2K^2B^2) + 8\alpha_0\beta_0\delta^2)$ *and* $C_2 = (144\tau_\delta^2 K^2 L^2\delta^2 + 4L^2/N)\alpha_0\beta_0$.

The first term of the right-hand side of (16) corresponds to the bias due to initialization, which goes to zero at a rate $\mathcal{O}(1/T^2)$. The second term is due to the variance of the Markovian noise. The third term corresponds to the accumulated estimation error of the two-timescale update. The second and third terms decay at a rate $\mathcal{O}(1/T^{2/3})$, and hence dominate the overall convergence rate in (16).

*Remark* 4.14. Doan (2020) provided the first finite-time analysis for general nonlinear 2TSA under i.i.d. noise, and then extended it to the Markovian noise setting under the assumptions that both $\bar{\mathbf{g}}$ and $\bar{\mathbf{h}}$ functions are monotone in both parameters (Doan, 2021). Since Doan (2021) leverages the methods from Doan (2020), it needs a detailed characterization of the covariance between the error induced by Markovian noise and the residual error of the parameters in (13), rendering the convergence analysis much more intricate. To address this, we take inspiration from (Srikant & Ying, 2019) (which operates in the single-timescale SA setting) and use a Lyapunov drift approach to capture the evolution of two coupled parameters under Markovian noise. Characterizing the impact of a norm-scaling step further distinguishes our work.

**Corollary 4.15.** *Suppose that* $\beta_0 = o(N^{-2/3})$ *and that* $T^2 > N$. *Then, we have*

$$M(\{\boldsymbol{\theta}_{t+2}^i\}, \boldsymbol{\Phi}_{t+1}) \leq \mathcal{O}\left(\frac{1}{N^{2/3}(T+2)^{2/3}}\right).$$

*Remark* 4.16. Corollary 4.15 indicates that to attain an $\epsilon$ accuracy, it requires $T = \mathcal{O}\left(N^{-1}\epsilon^{-3/2}\right)$ steps. In this sense, we prove that PFEDTD-REP achieves a *linear convergence speedup* with respect to the number of agents $N$, i.e., the number of steps until $\epsilon$-convergence is multiplied by a factor of $N^{-1}$. In other words, we can proportionally decrease $T$ as $N$ increases. To our knowledge, this is the first linear speedup result for personalized FedRL with shared representations under Markovian noise, which is highly desirable since it implies that one can efficiently leverage the massive parallelism in large-scale systems. Recently, Shen et al. (2023) considered a 2TSA in a federated RL setting and achieved a convergence rate of $\mathcal{O}\left(T^{-2/5}\right)$ and thus a sample complexity of $\mathcal{O}\left(\epsilon^{-5/2}\right)$. In contrast, our method can converge quicker and enjoys a lower sample complexity, and the convergence speed matches the best-known convergence speed for non-linear 2TSA under even i.i.d. noise (Doan, 2020). In addition, we note that single-timescale (SA) methods may enjoy a faster convergence speed and a lower sample complexity. However, the analysis of the 2TSA setting is more involved, as there are two parameters to be updated in a coupled and asynchronous manner. To our knowledge, there are no existing works in the 2TSA settings that achieve the same convergence rate or sample complexity as those in the SA settings. It may be an interesting direction to investigate for the community. Finally, similar to FL settings (Collins et al., 2021), the local step does not hurt the global convergence with a proper learning rate choice.

*Remark* 4.17. The gradient tracking technique discussed in Zeng & Doan (2024) could potentially be effective in handling Markovian sampling and improving the current convergence rate to $\mathcal{O}(T^{-1})$. However, it is unclear if it can be applied to PFEDRL-REP, because Zeng & Doan (2024) only considers single-agent settings under i.i.d. noise. Furthermore, Zeng & Doan (2024) assumes a second-order variance bound of the stochastic function, while our analysis does not include such an assumption. Investigating these is out of the scope of this work, which already considers a very challenging setting.

### 4.2.2 INTUITIONS AND PROOF SKETCH

We highlight the key ideas and challenges behind the convergence rate analysis of PFEDTD-REP with two coupled parameters, which is an example of a federated nonlinear 2TSA. With the defined

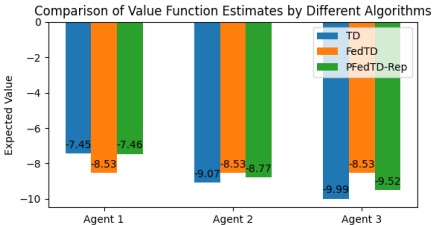
(a) Value function estimates.

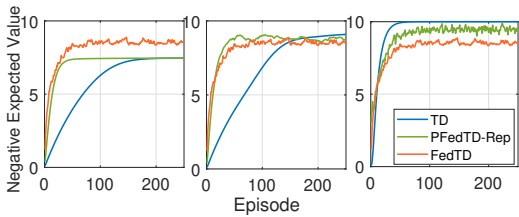
(b) Convergence speed of value function learning.

Figure 2: Comparisons in a CliffWalking Environment with 3 agents.

Lyapunov function in (14), the key is to find the drift between $M(\{\boldsymbol{\theta}_{t+1}^i, \forall i\}, \boldsymbol{\Phi}_t)$ in the $t$-th communication round and $M(\{\boldsymbol{\theta}_t^i, \forall i\}, \boldsymbol{\Phi}_{t-1})$ in the $(t-1)$-th communication round. To achieve this, we separately characterize the drift between $\boldsymbol{\Phi}_{t+1}$ and $\boldsymbol{\Phi}_t$, and the drift between $\boldsymbol{\theta}_{t+1}^i$ and $\boldsymbol{\theta}_t^i, \forall i$. We emphasize the three main challenges in characterizing the drift: (i) how to bound the stochastic gradient with Markovian samples; (ii) how to leverage the mixing time $\tau$ to handle the biased parameter updates due to Markovian noise; and (iii) how to deal with multiple local updates for the local weight vector $\boldsymbol{\theta}^i$.

By the mixing time property of MDPs, we have that the gap between the biased gradient at each time step and the true gradient can be bounded when the time step exceeds the mixing time $\tau$, as defined in Definition 4.5. To characterize the effect of local updates, the key idea is to bound the gradient at the initial local step and the gradient at the final local steps, which can be done by leveraging the Lipschitz property of those gradient functions in Lemmas 4.6, 4.7 and 4.8. See Appendix F.1.1 and Appendix F.1.2 for details. Once we establish the drift of the Lyapunov function, the remaining task is to select suitable *dynamic two-timescale learning rates* $\{\alpha_t, \forall t\}$ and $\{\beta_t, \forall t\}$ for the weight vector update in (7) and the feature representation update in (9), respectively. See Appendix F.1.3 for details.

## 4.3 NUMERICAL EVALUATION

We empirically evaluate the performance of PFEDTD-REP. We consider a tabular CliffWalking environment (Brockman et al., 2016) with a $4 \times 12$ grid world, where 3 agents evaluate 3 different policies. The dimension for the feature representation and weight vector is set to be 6. We compare PFEDTD-REP with (i) "TD": each agent independently leverages the conventional TD without communication; and (ii) "FedTD" without personalization (Khodadadian et al., 2022; Dal Fabbro et al., 2023) as listed in Table 1. As shown in Figure 2a, PFEDTD-REP ensures personalization among all agents while FedTD tends to converge uniformly among all agents. Further, PFEDTD-REP attains values much closer to the ground-truth achieved by TD for each agent compared to FedTD; and PFEDTD-REP converges much faster than TD. For instance, agent 1 only needs 50 episodes to converge under PFEDTD-REP, while it takes more than 150 episodes to converge under TD, as illustrated in Figure 2b. The improved convergence performance of PFEDTD-REP further supports our theoretical findings that leveraging shared representations not only provides personalization among agents in heterogeneous environments but yield faster convergence.

## 5 APPLICATION TO CONTROL PROBLEMS

In this section, we briefly discuss how our proposed PFEDRL-REP framework can be applied to the control problems in RL. More details are provided in Appendices C.3 (i.e., Algorithm 4) and G.

PFEDDQN-REP (an instance of PFEDRL-REP paired with DQN) leverages shared representations to learn a common feature space that captures the underlying dynamics and features relevant across different but related tasks encountered by various agents. In PFEDDQN-REP, the target network is a critical component that provides stability to the learning process by serving as a relatively static benchmark for calculating the loss during training updates (Mnih et al., 2015). The target network's architecture mirrors that of the main network, including the shared representation model. However, its parameters are updated less frequently. This setup ensures that the calculated target values, which guide the policy updates, are based on a consistent representation of the environment's state, as encoded by the shared representation model. The synergy between the target network and the repre-

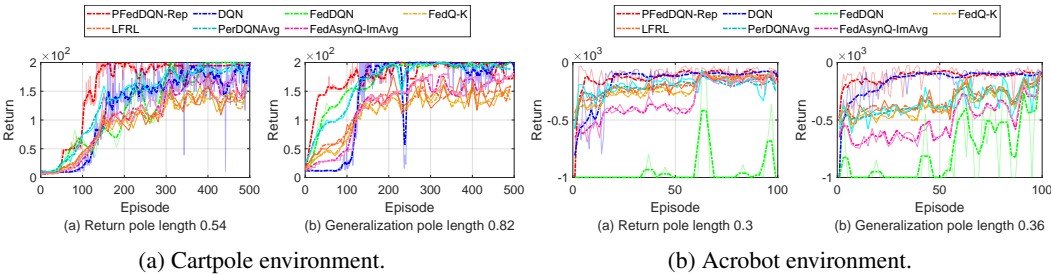

(a) Cartpole environment.    (b) Acrobot environment.

Figure 3: Comparisons in control problems.

sentation model is thus central to achieving stable and convergent learning. In Line 13 of Algorithm 4, the algorithm performs a scheduled update of the shared representation $\mathbf{\Phi}$ of the main network's parameters with the guidance of the target network. In Line 18 of Algorithm 4, every $T_{\text{target}}$ steps, the algorithm performs a scheduled update of the target network's parameters by copying over the parameters from the main network. This step is essential for maintaining the stability of the learning process, as it ensures that the target values against which the policy updates are computed remain consistent and reflects the most recent knowledge encoded in the shared representation. The update frequency is carefully chosen to balance learning stability with model adaptivity.

**Numerical evaluation.** We consider a modified CartPole environment (Brockman et al., 2016) by changing the length of pole to create heterogenous environments (Jin et al., 2022). Specifically, we consider 10 agents with varying pole lengths from $0.38$ to $0.74$ with a step size of $0.04$. We compare PFEDDQN-REP with (i) a conventional DQN where each agent learns its own environment independently; (ii) a federated version of DQN (FedDQN) that allows all agents to collaboratively learn a single policy (without personalization); (iii) two federated algorithms without personalizing, FedQ-K (Khodadadian et al., 2022) and LFRL (Liu et al., 2019); and (iv) two personalized algorithms, PerDQNAvg (Jin et al., 2022) and FedAsynQ-ImAvg (Woo et al., 2023). We randomly choose one agent and present its performance in Figure 3a. We observe that our PFEDDQN-REP obtains larger reward than the baselines without personalization and achieves the maximal return faster than existing personalized algorithms. We further evaluate the effectiveness of the shared representation learned by PFEDDQN-REP when generalizing to a new environment. As shown in Figure 3a, PFEDDQN-REP generalizes quickly to the new environment. Similar observations can be made from Figure 3b using Acrobot environments (see details in Appendix G). In summary, the significance of our PFEDRL-REP framework lies in its superior performance in heterogeneous environments compared to existing algorithms that do not incorporate personalization. Additionally, our PFEDRL-REP framework also enables quick adaptation to new, previously unobserved environments.

**Limitations and open problems.** In this paper, we characterize the finite-time convergence rate for PFEDTD-REP with linear feature representation. However, our analysis is not directly applicable to control problems. Consider Q-learning. The difficulty arises because the update for Q-learning is not a linear operation with respect to the shared representation and local weights. Additionally, Q-learning is typically combined with deep neural networks, where the Q-function is approximated by a neural network as in PFEDDQN-REP. The complexity is further compounded in personalized federated RL frameworks, where multiple agents share a common representation while maintaining personalized local weights. Given the promising experimental results on control, whether we can extend our theoretical results to the control setting remains an open problem.

ACKNOWLEDGEMENTS

This work was supported in part by the National Science Foundation (NSF) grants 2148309, 2315614 and 2337914, and was supported in part by funds from OUSD R&E, NIST, and industry partners as specified in the Resilient & Intelligent NextG Systems (RINGS) program. Any opinions, findings, and conclusions or recommendations expressed in this material are those of the authors and do not necessarily reflect the views of the funding agencies.

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

# A    RELATED WORK

**Single-agent reinforcement learning.** RL is a machine learning paradigm that trains agents to make sequences of decisions by rewarding desired behaviors and/or penalizing undesired ones in a given environment (Sutton & Barto, 2018). Starting from Temporal Difference (TD) Learning (Sutton, 1988), which introduced the concept of learning from the discrepancy between predicted and actual rewards through episodes, the widely used Q-Learning (Watkins & Dayan, 1992) emerged, advancing the field with an off-policy algorithm that learns action-value functions and enables policy improvement without needing a model of the environment. Later on, the introduction of Deep Q-Networks (DQN) (Mnih et al., 2015) marked a significant leap, integrating deep neural networks with Q-Learning to handle high-dimensional state spaces, thus enabling RL to tackle complex problems. Subsequently, policy-based algorithms such as Proximal Policy Optimization (PPO) (Schulman et al., 2017) and deep Deterministic Policy Gradients (DDPG) (Silver et al., 2014), leverage the Actor-Critic framework to provide more stable and robust ways to directly optimize the policy, overcoming challenges related to action space and variance.

**Federated reinforcement learning.** Jin et al. (2022) introduced a FedRL framework with $N$ agents collaboratively learning a policy by averaging their Q-values or policy gradients. Khodadadian et al. (2022) provided a convergence analysis of federated TD (FedTD) and Q-learning (FedQ) when $N$ agents interact with homogeneous environments. A similar FedTD was considered in Dal Fabbro et al. (2023), and expanded to heterogeneous environments in Wang et al. (2023a). Woo et al. (2023) analyzed (a)synchronous variants of FedQ in heterogeneous settings, and an asynchronous actor-critic method was considered in Shen et al. (2023) with linear speedup guarantee only under i.i.d. samples. Zhang et al. (2024) provided a finite-time analysis of FedSARSA with linear function approximation (i.e., fixed feature representation). To facilitate personalization in heterogeneous settings, Jin et al. (2022) proposed a heuristic personalized FedRL method where agents share a common model, but make use of individual environment embeddings. There is also a related paper Fan et al. (2021), which considers a special setting where each agent can be Byzantine and suffers random failure in every round. In Fan et al. (2021), convergence was established based on i.i.d. noise.

**Personalized federated learning (PFL).** In contrast to standard FL where a single model is learned, PFL aims to learn $N$ models specialized for $N$ local datasets. Many PFL methods have been developed, including but not limited to multi-task learning (Smith et al., 2017), meta-learning (Chen et al., 2018), and various personalization techniques such as local fine-tuning (Fallah et al., 2020), layer personalization (Arivazhagan et al., 2019), and model compression (Bergou et al., 2022). Another line of work (Collins et al., 2021; Xiong et al., 2024) leveraged the common representation among agents in heterogeneous environments to guarantee personalized models for federated supervised learning.

**Representation learning in MDP.** Representation learning aims to transform high-dimensional observation to low-dimensional embedding to enable efficient learning, and has received increasing attention in Markov decision processs (MDP) settings, such as linear MDPs (Jin et al., 2020), low-rank MDPs (Modi et al., 2021; Agarwal et al., 2020) and block MDPs (Zhang et al., 2022c). However, it is open in the context of leveraging representation learning in PFedFL. In this work, we prove that representation augmented PFedFL forms a general framework as a federated two-timescale stochastic approximation with Markovian noise, which differs significantly from existing works, and hence necessitates different proof techniques.

**Multi-agent reinforcement learning versus federated reinforcement learning.** The advent of Multi-Agent Reinforcement Learning (MARL) expanded RL's applications, allowing multiple agents to learn from interactions in cooperative, competitive, or mixed settings, opening new avenues for complex applications and research (Zhang et al., 2021). Multi-agent Reinforcement Learning (MARL) addresses scenarios where multiple agents operate within a shared or interrelated environment, potentially engaging in both cooperative and competitive behaviors. The complexity arises from each agent needing to consider the strategies and actions of others, making the learning process highly dynamic. Federated Reinforcement Learning (FedRL) (Qi et al., 2021), contrasts with MARL by focusing on privacy-preserving, distributed learning across agents that do not share their raw data. Instead, these agents might contribute towards a centralized learning model without compromising individual data privacy, addressing the unique challenges of learning from decentralized data sources.

---

**Algorithm 2** PFEDTD-REP

---

1: **Input:** Sampling policy $\pi^i, \forall i \in [N]$;
2: Initialize $\boldsymbol{\theta}_0^i = \mathbf{0}$, $S_0^i, \forall i \in [N]$, and randomly generate $\boldsymbol{\Phi} \in \mathbb{R}^{|\mathcal{S}| \times d}$ with each row being unit-norm vector;
3: **for** $t = 0, 1, ..., T - 1$ **do**
4:   **for** $i = 1, \ldots, N$ **do**
5:     **for** $k = 1, \ldots, K$ **do**
6:       Sample observations $X_{t,k-1}^i$;
7:       Set $\boldsymbol{\theta}_{t,k}^i = \boldsymbol{\theta}_{t,k-1}^i + \alpha_t \mathbf{g}(\boldsymbol{\theta}_{t,k-1}^i, \boldsymbol{\Phi}_t, X_{t,k-1}^i)$;
8:     **end for**
9:     Scale $\|\boldsymbol{\theta}_{t+1}^i\|$ to $B$ if $\|\boldsymbol{\theta}_{t+1}^i\| > B$, otherwise keep it unchanged;
10:    Set $\boldsymbol{\Phi}_{t+1/2}^i = \boldsymbol{\Phi}_t + \beta_t \mathbf{h}(\boldsymbol{\theta}_{t+1}^i, \boldsymbol{\Phi}_t, \{X_{t,k-1}^i\}_{k=1}^K)$;
11:    Normalize $\boldsymbol{\Phi}_{t+1/2}^i$ as $\boldsymbol{\Phi}_{t+1/2}^i \leftarrow \frac{\boldsymbol{\Phi}_{t+1/2}^i}{\|\boldsymbol{\Phi}_{t+1/2}^i\|}$;
12:  **end for**
13:    $\boldsymbol{\Phi}_{t+1} = \frac{1}{N} \sum_{i=1}^N \boldsymbol{\Phi}_{t+1/2}^i$.
14: **end for**

---

## B    PSEUDOCODE OF PFEDTD-REP

In this section, we present the pseudocode of PFEDQ-REP as summarized in Algorithm 2.

## C    APPLICATION TO CONTROL TASKS IN RL

The Q-function of agent $i$ in environment $\mathcal{M}^i$ under policy $\pi^i$ is defined as $Q^{i,\pi^i}(s,a) = \mathbb{E}_{\pi^i}\left[\sum_{k=0}^\infty \gamma^k R^i(s_k^i, a_k^i) | s_0^i = s, a_0^i = a\right]$. When the state and action spaces are large, it is computationally infeasible to store $Q^{i,\pi^i}(s,a)$ for all state-action pairs. One way to deal with is to approximate the Q-function as $Q^{i,\pi^i}(s,a) \approx \boldsymbol{\Phi}(s,a)\boldsymbol{\theta}$, where $\boldsymbol{\Phi} \in \mathbb{R}^{|\mathcal{S}| \times |\mathcal{A}| \times d}$ is a feature representation corresponding to state-actions, and $\boldsymbol{\theta} \in \mathbb{R}^d$ is a low-dimensional unknown weight vector. When $\boldsymbol{\Phi}$ is given and known, this falls under the paradigm of RL or FedRL with function approximation.

### C.1    PRELIMINARIES: CONTROL IN FEDERATED REINFORCEMENT LEARNING

Another task in RL is to search for an optimal policy, which is called *a control problem*, and one commonly used approach is Q-learning (Watkins & Dayan, 1992). Under the FedRL framework, the goal of a control problem is to let $N$ agents collaboratively learn a policy $\pi^*$ that performs uniformly well across $N$ different environments, i.e., $\pi^* = \arg\max_\pi \frac{1}{N} \sum_{i=1}^N \mathbb{E}_{\pi^i}\left[V^{i,\pi^i}(s_0^i)|s_0^i \sim d_0\right]$, where $d_0$ is the common initial state distribution in these $N$ environments. Similar to (1), this can be formulated as the optimization problem in (17) to collaboratively learn a common (non-personalized) weight vector $\boldsymbol{\theta} \equiv \boldsymbol{\theta}^i, \forall i \in [N]$ when the feature representation $\boldsymbol{\Phi}(s,a), \forall s, a$ are given.

$$\mathcal{L}(\boldsymbol{\theta}) := \min_{\boldsymbol{\theta}} \frac{1}{N} \sum_{i=1}^N \mathbb{E}_{\substack{s \sim \mu^{i,\pi^*} \\ a \sim \pi^*(\cdot|s)}} \left\| \boldsymbol{\Phi}(s,a)\boldsymbol{\theta} - Q^{i,\pi^*}(s,a) \right\|^2. \tag{17}$$

Again, we use the superscript $i$ to highlight heterogeneous environments $P^i$ among agents.

## C.2 Control in Personalized FedRL with Shared Representations

The control problem in (17) aims to learn $\boldsymbol{\Phi}$ and $\{\boldsymbol{\theta}^i, \forall i\}$ simultaneously among all $N$ agents via solving the following optimization problem:

$$\mathcal{L}(\boldsymbol{\Phi}, \{\boldsymbol{\theta}^i, \forall i\}) := \min_{\boldsymbol{\Phi}} \frac{1}{N} \sum_{i=1}^{N} \min_{\{\boldsymbol{\theta}^i, \forall i\}} \mathbb{E}_{\substack{s \sim \mu^{i, \pi^{i,*}} \\ a \sim \pi^{i,*}(\cdot|s)}} \left\| f^i(\boldsymbol{\theta}^i, \boldsymbol{\Phi}(s,a)) - Q^{i, \pi^{i,*}}(s,a) \right\|^2. \quad (18)$$

## C.3 Algorithms

In this subsection, we present two realizations of our proposed PFEDRL-REP in Algorithm 1, one is PFEDQ-REP as summarized in Algorithm 3, federated Q-learning with shared representations, and the other is PFEDDQN-REP as outlined in Algorithm 4, federated DQN with shared representations.

---

**Algorithm 3** PFEDQ-REP

**Input:** Sampling policy $\pi^i, \forall i \in [N]$.

1: Initialize $\boldsymbol{\theta}_0^i = \mathbf{0}$, and $s_0^i, \forall i \in [N]$, and randomly generate $\boldsymbol{\Phi} \in \mathbb{R}^{|\mathcal{S}||\mathcal{A}| \times d}$ with each row being unit-norm vector.
2: **for** $t = 0, 1, ..., T-1$ **do**
3:     **for** $i = 1, \ldots, N$ **do**
4:         **for** $k = 1, \ldots, K$ **do**
5:             Sample observations $X_{t,k-1}^i = (s_{t,k}^i, s_{t,k-1}^i, a_{t,k-1}^i)$;
6:             With fixed $\boldsymbol{\Phi}_t$, update $\boldsymbol{\theta}_{t,k}^i \leftarrow \boldsymbol{\theta}_{t,k-1}^i + \alpha_t \cdot (r_{t,k-1}^i + \gamma \max_a \boldsymbol{\Phi}_t(s_{t,k+1}^i, a)\boldsymbol{\theta}_{t,k-1}^i - \boldsymbol{\Phi}_t(s_{t,k-1}^i)\boldsymbol{\theta}_{t,k-1}^i) \cdot \boldsymbol{\Phi}_t(s_{t,k-1}^i, a_{t,k-1}^i)$;
7:         **end for**
8:         Scale $\|\boldsymbol{\theta}_{t+1}^i\|$ to $B$ if $\|\boldsymbol{\theta}_{t+1}^i\| > B$, otherwise keep it unchanged.
9:         **if** $(s,a) \in X_{t,k}^i, \exists k \in \{0, \ldots, K-1\}$ **then**
10:           Update $\boldsymbol{\Phi}_{t+1/2}^i(s,a) = \boldsymbol{\Phi}_t^i(s,a) + \beta_t(r(s,a) + \gamma \max_a \boldsymbol{\Phi}_t(s',a)\boldsymbol{\theta}_{t+1}^i - \boldsymbol{\Phi}_t(s,a)^{\mathsf{T}}\boldsymbol{\theta}_{t+1}^i) \cdot \boldsymbol{\theta}_{t+1}^i$;
11:         **else**
12:           $\boldsymbol{\Phi}_{t+1/2}^i(s,a) = \boldsymbol{\Phi}_t^i(s,a)$;
13:         **end if**
14:         Normalize $\boldsymbol{\Phi}_{t+1/2}^i$ as $\boldsymbol{\Phi}_{t+1/2}^i \leftarrow \frac{\boldsymbol{\Phi}_{t+1/2}^i}{\|\boldsymbol{\Phi}_{t+1/2}^i\|}$;
15:     **end for**
16:     $\boldsymbol{\Phi}_{t+1} \leftarrow \frac{1}{N} \sum_{i=1}^{N} \boldsymbol{\Phi}_{t+1/2}^i, \forall i \in [N]$.
17: **end for**

---

# D Figure Illustrations

We present some figures to further highlight the proposed personalized FedRL (PFEDRL) framework with shared representations.

**Schematic framework of conventional FedRL.** We begin by introducing the conventional FedRL framework (Khodadadian et al., 2022), where $N$ agents collaboratively learn a common policy (or optimal value functions) via a server while engaging with homogeneous environments. Each agent generates independent Markovian trajectories, as depicted in Figure 4.

**Schematic framework for our proposed PFEDRL with shared representations.** We introduce our proposed personalized FedRL (PFEDRL) framework with shared representations in Figure 5. In PFEDRL, $N$ agents independently interact with their own environments and execute actions according to their individual RL component parameterized by $\boldsymbol{\Phi}$ and $\boldsymbol{\theta}^i$. Each agent $i$ performs local update on its local weight vector $\boldsymbol{\theta}_i$, while jointly updating the global shared feature representation $\boldsymbol{\Phi}$ through the server. Similarly, the update follows the Markovian trajectories.

**Motivation of personalized FedRL.** In the following, we also want to provide some examples showing that the conventional FedRL framework may fail, as depicted in Figure 6. In Figure 6a,

---

**Algorithm 4** PFEDDQN-REP

---

**Initialize:** The parameters $(\mathbf{\Phi}, \boldsymbol{\theta}^i)$ for each Q network $Q^i(s,a)$, the replay buffer $\mathcal{R}^i$, and copy the same parameter from Q network to initialize the target Q network $Q^{i,\prime}(s,a)$ for agent $i, \forall i \in [N]$;

1: **for** episode $e = 1, \ldots, E$ **do**
2:     Get the initial state of the environment;
3:     **for** $t = 0, 1, ..., T-1$ **do**
4:         **for** $i = 1, \ldots, N$ **do**
5:             **for** $k = 1, \ldots, K$ **do**
6:                 Select action $a_{t,k-1}$ according to $\epsilon$-greedy policy with the current network $Q^i(s_{t,k-1}, a)$;
7:                 Execute action $a_{t,k-1}$, receive the reward $r(s_{t,k-1}, a_{t,k-1})$, and the environment state transits to $s_{t,k}$;
8:                 Store the tuple $(s_{t,k-1}, a_{t,k-1}, r(s_{t,k-1}, a_{t,k-1}, s_{t,k})$ into the replay buffer $\mathcal{R}^i$;
9:                 Sample $N$ data tuples from the replay buffer $\mathcal{R}^i$;
10:            Update the local weight $\boldsymbol{\theta}^i(t,k)$ by minimizing the loss compared with the target network $Q^{i,\prime}$ with fixed representation $\mathbf{\Phi}_t$;
11:         **end for**
12:         Sample $N$ data tuples from replay buffer $\mathcal{R}^i$;
13:         Update representation model locally by minimizing the loss compared with the target network $Q^{i,\prime}$ with fixed weights $\boldsymbol{\theta}_{t+1}$, and yield $\mathbf{\Phi}^i_{t+1/2}$;
14:         **end for**
15:     Average the representation model from all agents, i.e., $\mathbf{\Phi}_{t+1} := \frac{1}{N} \sum_{i=1}^{N} \mathbf{\Phi}^i_{t+1/2}$;
16:     **end for**
17:     **if** $mod(t, T_{target}) = 0$ **then**
18:         update the target network $Q^{i,*}$ be copy the up-to-date parameters of Q network $Q^i, \forall i \in [N]$;
19:     **end if**
20: **end for**

---

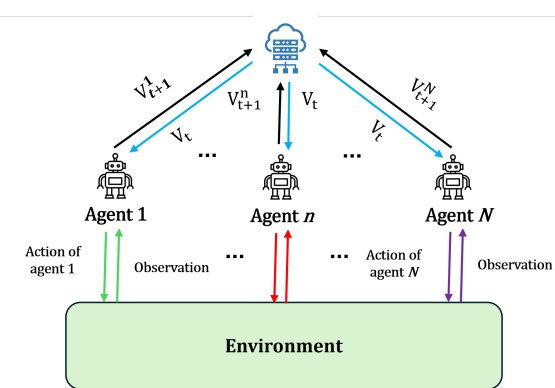

Figure 4: Schematic representation of FedRL, where $N$ agents interact with homogeneous environments.

we provide an example where three agents assess distinct policies within the same environment. In the traditional FedRL framework, agents exchange the evaluated value functions via a central server, leading to a unified consensus on value functions for three different policies. This enforced consensus on value functions, despite the diversity in policies, is not optimal. In another scenario depicted in Figure 6b, three agents each interact with their unique environments. The objective for each agent is to learn an optimal policy tailored to its specific environment. However, within the traditional FedRL framework, the central server mandates a uniform policy across all three agents, which clearly contradicts the intended goal of achieving environment-specific optimization. This highlights the necessity for personalized decision-making, a feature that conventional FedRL frameworks do not accommodate.

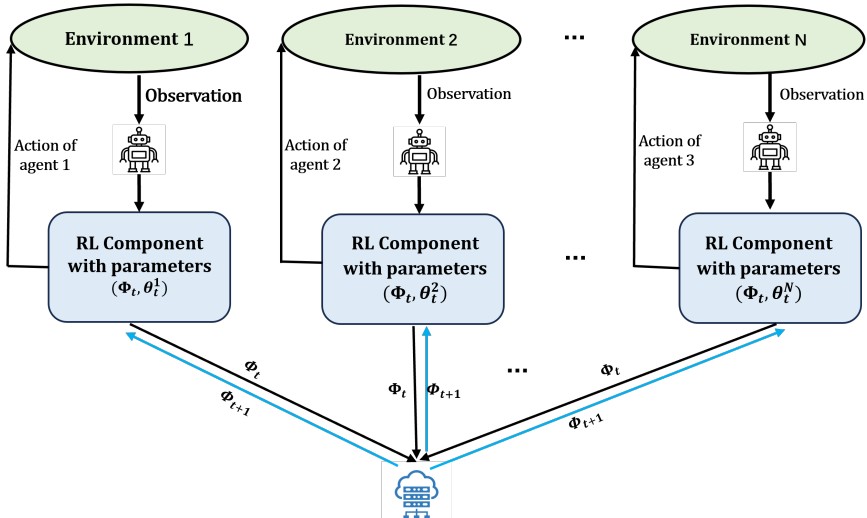

Figure 5: Our proposed PFEDRL-REP framework where $N$ agents independently interact with their own environments and take actions according to their individual RL component parameterized by $\boldsymbol{\Phi}$ and $\boldsymbol{\theta}^i$. Agent $i$ locally update weight vector $\boldsymbol{\theta}_i$ while jointly updating the shared feature representation $\boldsymbol{\Phi}$ through the central server. The update follows the Markovian trajectories.

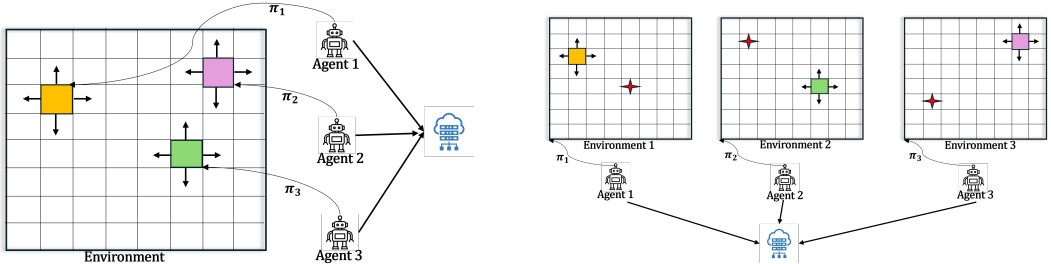

(a) Agents evaluate difference policies in the same environment.

(b) Agents learn optimal policies for heterogeneous environments.

Figure 6: *An illustrative example with three agents that demonstrates the conventional FedRL framework fails to work.*

**Example of RL components that fit the proposed PFEDRL with shared representations.** In the following, we aim to showcase examples of RL components that are compatible with our proposed PFEDRL framework featuring shared representations. An illustrative example of this framework is presented in Figure 7. It is important to note that both the DQN architecture in Figure 7a and the policy gradient (PG) approach in Figure 7b seamlessly integrate into our proposed framework. This integration is achieved by designating the parameters of the feature extraction network as the shared feature representation $\boldsymbol{\Phi}$, and the parameters of the fully connected network, which either predict the Q-values or determine the policy, as the local weight vector $\boldsymbol{\theta}$. This arrangement underscores the adaptability of our framework to various RL methodologies, facilitating personalized learning while maintaining a common foundation of shared representations.

# E    PROOF OF LEMMAS IN SECTION 4.2

## E.1    PROOF OF LEMMA 4.6

*Proof.* Recall that for any observation $X = (s, a, s')$, the function $\mathbf{g}(\boldsymbol{\theta}, \boldsymbol{\Phi}, X)$ defined in (8) is expressed as

$$\mathbf{g}(\boldsymbol{\theta}, \boldsymbol{\Phi}, X) := (r(s,a) + \gamma \boldsymbol{\Phi}(s')\boldsymbol{\theta} - \boldsymbol{\Phi}(s)\boldsymbol{\theta}) \cdot \boldsymbol{\Phi}(s)^{\mathsf{T}},$$

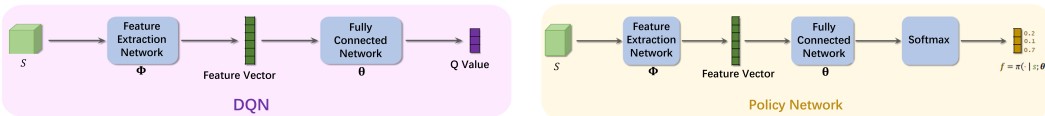

(a) When DQN meets the proposed framework.    (b) When PG meets the proposed framework.

Figure 7: *An illustrative example for the proposed framework. Notice that both the DQN in (a) and policy gradient (PG) in (b) can be fitted into the proposed framework by treating the parameters of the feature extraction network as the shared feature representation $\boldsymbol{\Phi}$ and the parameter of the fully connected network which maps to the Q value of policy as the local weight vector $\boldsymbol{\theta}$.*

and hence we have the following inequality for any parameter pairs $(\boldsymbol{\theta}_1, \boldsymbol{\Phi}_1)$ and $(\boldsymbol{\theta}_2, \boldsymbol{\lambda}_2)$ with $X = (s, a, s')$,

$$\|\mathbf{g}(\boldsymbol{\theta}_1, \boldsymbol{\Phi}_1, X) - \mathbf{g}(\boldsymbol{\theta}_2, \boldsymbol{\Phi}_2, X)\|$$
$$= \|(r(s,a) + \gamma\boldsymbol{\Phi}_1(s')\boldsymbol{\theta}_1 - \boldsymbol{\Phi}_1(s)\boldsymbol{\theta}_1) \cdot \boldsymbol{\Phi}_1(s)^{\mathsf{T}} - (r(s,a) + \gamma\boldsymbol{\Phi}_2(s')\boldsymbol{\theta}_2 - \boldsymbol{\Phi}_2(s)\boldsymbol{\theta}_2) \cdot \boldsymbol{\Phi}_2(s)^{\mathsf{T}}\|$$
$$\overset{(a_1)}{\leq} \|(\gamma\boldsymbol{\Phi}_1(s')\boldsymbol{\theta}_1 - \boldsymbol{\Phi}_1(s)\boldsymbol{\theta}_1) \cdot \boldsymbol{\Phi}_1(s)^{\mathsf{T}} - (\gamma\boldsymbol{\Phi}_1(s')\boldsymbol{\theta}_2 - \boldsymbol{\Phi}_1(s)\boldsymbol{\theta}_2) \cdot \boldsymbol{\Phi}_1(s)^{\mathsf{T}}\|$$
$$+ \|(\gamma\boldsymbol{\Phi}_1(s')\boldsymbol{\theta}_2 - \boldsymbol{\Phi}_1(s)\boldsymbol{\theta}_2) \cdot \boldsymbol{\Phi}_1(s)^{\mathsf{T}} - (\gamma\boldsymbol{\Phi}_2(s')\boldsymbol{\theta}_2 - \boldsymbol{\Phi}_2(s)\boldsymbol{\theta}_2) \cdot \boldsymbol{\Phi}_2(s)^{\mathsf{T}}\|$$
$$\overset{(a_2)}{\leq} \|(\gamma\boldsymbol{\Phi}_1(s')\boldsymbol{\theta}_1 - \boldsymbol{\Phi}_1(s)\boldsymbol{\theta}_1) - (\gamma\boldsymbol{\Phi}_1(s')\boldsymbol{\theta}_2 - \boldsymbol{\Phi}_1(s)\boldsymbol{\theta}_2)\| \cdot \|\boldsymbol{\Phi}_1(s)\|$$
$$+ \|(\gamma\boldsymbol{\Phi}_1(s')\boldsymbol{\theta}_2 - \boldsymbol{\Phi}_1(s)\boldsymbol{\theta}_2) \cdot \boldsymbol{\Phi}_1(s)^{\mathsf{T}} - (\gamma\boldsymbol{\Phi}_2(s')\boldsymbol{\theta}_2 - \boldsymbol{\Phi}_2(s)\boldsymbol{\theta}_2) \cdot \boldsymbol{\Phi}_2(s)^{\mathsf{T}}\|$$
$$\overset{(a_3)}{\leq} (1 + \gamma) \|\boldsymbol{\theta}_1 - \boldsymbol{\theta}_2\| + \|(\gamma\boldsymbol{\Phi}_1(s')\boldsymbol{\theta}_2 - \boldsymbol{\Phi}_1(s)\boldsymbol{\theta}_2) \cdot \boldsymbol{\Phi}_1(s)^{\mathsf{T}} - (\gamma\boldsymbol{\Phi}_2(s')\boldsymbol{\theta}_2 - \boldsymbol{\Phi}_2(s)\boldsymbol{\theta}_2) \cdot \boldsymbol{\Phi}_2(s)^{\mathsf{T}}\|$$
$$\overset{(a_4)}{\leq} (1 + \gamma) \|\boldsymbol{\theta}_1 - \boldsymbol{\theta}_2\| + \|(\gamma\boldsymbol{\Phi}_1(s')\boldsymbol{\theta}_2 - \boldsymbol{\Phi}_1(s)\boldsymbol{\theta}_2) \cdot \boldsymbol{\Phi}_1(s)^{\mathsf{T}} - (\gamma\boldsymbol{\Phi}_1(s')\boldsymbol{\theta}_2 - \boldsymbol{\Phi}_1(s)\boldsymbol{\theta}_2) \cdot \boldsymbol{\Phi}_2(s)^{\mathsf{T}}\|$$
$$+ \|(\gamma\boldsymbol{\Phi}_1(s')\boldsymbol{\theta}_2 - \boldsymbol{\Phi}_1(s)\boldsymbol{\theta}_2) \cdot \boldsymbol{\Phi}_2(s)^{\mathsf{T}} - (\gamma\boldsymbol{\Phi}_2(s')\boldsymbol{\theta}_2 - \boldsymbol{\Phi}_2(s)\boldsymbol{\theta}_2) \cdot \boldsymbol{\Phi}_2(s)^{\mathsf{T}}\|$$
$$\overset{(a_5)}{\leq} (1 + \gamma) \|\boldsymbol{\theta}_1 - \boldsymbol{\theta}_2\| + \left\|(\gamma\boldsymbol{\Phi}_1(s')\boldsymbol{\theta}_2 - \boldsymbol{\Phi}_1(s)\boldsymbol{\theta}_2)\right\| \cdot \left\|\boldsymbol{\Phi}_1(s) - \boldsymbol{\Phi}_2(s)\right\|$$
$$+ \|(\gamma\boldsymbol{\Phi}_1(s')\boldsymbol{\theta}_2 - \boldsymbol{\Phi}_1(s)\boldsymbol{\theta}_2) - (\gamma\boldsymbol{\Phi}_2(s')\boldsymbol{\theta}_2 - \boldsymbol{\Phi}_2(s)\boldsymbol{\theta}_2)\| \cdot \|\boldsymbol{\Phi}_2(s)\|$$
$$\overset{(a_6)}{\leq} (1 + \gamma) \left\|\boldsymbol{\theta}_1 - \boldsymbol{\theta}_2\right\| + \left\|(\gamma\boldsymbol{\Phi}_1(s')\boldsymbol{\theta}_2 - \boldsymbol{\Phi}_1(s)\boldsymbol{\theta}_2)\right\| \cdot \|\boldsymbol{\Phi}_1(s) - \boldsymbol{\Phi}_2(s)\|$$
$$+ \|\boldsymbol{\Phi}_1(s') - \boldsymbol{\Phi}_2(s')\| \cdot \|\gamma\boldsymbol{\theta}_2\| + \|\boldsymbol{\Phi}_1(s) - \boldsymbol{\Phi}_2(s)\| \cdot \|\boldsymbol{\theta}_2\|$$
$$\leq (1 + \gamma) \|\boldsymbol{\theta}_1 - \boldsymbol{\theta}_2\| + (2 + 2\gamma) \|\boldsymbol{\theta}_2\| \cdot \|\boldsymbol{\Phi}_1 - \boldsymbol{\Phi}_2\|$$
$$\overset{(a_7)}{\leq} L_g \left(\|\boldsymbol{\theta}_1 - \boldsymbol{\theta}_2\| + \|\boldsymbol{\Phi}_1 - \boldsymbol{\Phi}_2\|\right),$$

$(a_1)$ is due to the fact that $\|\mathbf{x} + \mathbf{y}\| \leq \|\mathbf{x}\| + \|\mathbf{y}\|, \forall \mathbf{x}, \mathbf{y} \in \mathbb{R}^d$, $(a_2)$ holds due to $\|\mathbf{x} \cdot \mathbf{y}\| \leq \|\mathbf{x}\| \cdot \|\mathbf{y}\|, \forall \mathbf{x}, \mathbf{y} \in \mathbb{R}^d$, $(a_3)$ comes from the fact and $\|\boldsymbol{\Phi}_1(s)\| \leq 1, \|\boldsymbol{\Phi}_2(s)\| \leq 1 \forall s$. $(a_4) - (a_6)$ holds for the same reason as $(a_1) - (a_3)$. The last inequalty $(a_7)$ comes from the fact that $\boldsymbol{\theta}$ is bounded by norm $B$ and by setting $L_g := \max(1 + \gamma, (2 + 2\gamma)B)$. $\qquad\square$

### E.2   PROOF OF LEMMA 4.7

*Proof.* Recall that for any observation $X = (s, a, s')$, the function $\mathbf{h}(\boldsymbol{\theta}, \boldsymbol{\Phi}, X)$ defined in (10) is expressed as

$$\mathbf{h}(\boldsymbol{\theta}, \boldsymbol{\Phi}, X) := (r(s,a) + \gamma\boldsymbol{\Phi}(s')\boldsymbol{\theta} - \boldsymbol{\Phi}(s)\boldsymbol{\theta}) \cdot \boldsymbol{\theta}^{\mathsf{T}},$$

and hence we have the following inequality for any parameter pairs $(\boldsymbol{\theta}_1, \boldsymbol{\Phi}_1)$ and $(\boldsymbol{\theta}_2, \boldsymbol{\lambda}_2)$ with $X = (s, a, s')$,

$$\|\mathbf{h}(\boldsymbol{\theta}_1, \boldsymbol{\Phi}_1, X) - \mathbf{h}(\boldsymbol{\theta}_2, \boldsymbol{\Phi}_2, X)\|$$
$$= \|(r(s,a) + \gamma\boldsymbol{\Phi}_1(s')\boldsymbol{\theta}_1 - \boldsymbol{\Phi}_1(s)\boldsymbol{\theta}_1) \cdot \boldsymbol{\theta}_1^{\mathsf{T}} - (r(s,a) + \gamma\boldsymbol{\Phi}_2(s')\boldsymbol{\theta}_2 - \boldsymbol{\Phi}_2(s)\boldsymbol{\theta}_2) \cdot \boldsymbol{\theta}_2^{\mathsf{T}}\|$$

$$\overset{(b_1)}{\leq} \|(\gamma\mathbf{\Phi}_1(s')\boldsymbol{\theta}_1 - \mathbf{\Phi}_1(s)\boldsymbol{\theta}_1) \cdot \boldsymbol{\theta}_1^\mathsf{T} - (\gamma\mathbf{\Phi}_2(s')\boldsymbol{\theta}_1 - \mathbf{\Phi}_2(s)\boldsymbol{\theta}_1) \cdot \boldsymbol{\theta}_1^\mathsf{T}\|$$
$$+ \|(\gamma\mathbf{\Phi}_2(s')\boldsymbol{\theta}_1 - \mathbf{\Phi}_2(s)\boldsymbol{\theta}_1) \cdot \boldsymbol{\theta}_1^\mathsf{T} - (\gamma\mathbf{\Phi}_2(s')\boldsymbol{\theta}_2 - \mathbf{\Phi}_2(s)\boldsymbol{\theta}_2) \cdot \boldsymbol{\theta}_2^\mathsf{T}\|$$

$$\overset{(b_2)}{\leq} \|(\gamma\mathbf{\Phi}_1(s')\boldsymbol{\theta}_1 - \mathbf{\Phi}_1(s)\boldsymbol{\theta}_1) - (\gamma\mathbf{\Phi}_2(s')\boldsymbol{\theta}_1 - \mathbf{\Phi}_2(s)\boldsymbol{\theta}_1)\| \cdot \|\boldsymbol{\theta}_1\|$$
$$+ \|(\gamma\mathbf{\Phi}_2(s')\boldsymbol{\theta}_1 - \mathbf{\Phi}_2(s)\boldsymbol{\theta}_1) \cdot \boldsymbol{\theta}_1^\mathsf{T} - (\gamma\mathbf{\Phi}_2(s')\boldsymbol{\theta}_2 - \mathbf{\Phi}_2(s)\boldsymbol{\theta}_2) \cdot \boldsymbol{\theta}_2^\mathsf{T}\|$$

$$\overset{(b_3)}{\leq} (1+\gamma)\|\boldsymbol{\theta}_1\|^2 \cdot \|\mathbf{\Phi}_1 - \mathbf{\Phi}_2\| + \|(\gamma\mathbf{\Phi}_2(s')\boldsymbol{\theta}_1 - \mathbf{\Phi}_2(s)\boldsymbol{\theta}_1) \cdot \boldsymbol{\theta}_1^\mathsf{T} - (\gamma\mathbf{\Phi}_2(s')\boldsymbol{\theta}_2 - \mathbf{\Phi}_2(s)\boldsymbol{\theta}_2) \cdot \boldsymbol{\theta}_2^\mathsf{T}\|$$

$$\overset{(b_4)}{\leq} (1+\gamma)\|\boldsymbol{\theta}_1\|^2 \cdot \|\mathbf{\Phi}_1 - \mathbf{\Phi}_2\| + \|(\gamma\mathbf{\Phi}_2(s')\boldsymbol{\theta}_1 - \mathbf{\Phi}_2(s)\boldsymbol{\theta}_1) \cdot \boldsymbol{\theta}_1^\mathsf{T} - (\gamma\mathbf{\Phi}_2(s')\boldsymbol{\theta}_1 - \mathbf{\Phi}_2(s)\boldsymbol{\theta}_1) \cdot \boldsymbol{\theta}_2^\mathsf{T}\|$$
$$+ \|(\gamma\mathbf{\Phi}_2(s')\boldsymbol{\theta}_1 - \mathbf{\Phi}_2(s)\boldsymbol{\theta}_1) \cdot \boldsymbol{\theta}_2^\mathsf{T} - (\gamma\mathbf{\Phi}_2(s')\boldsymbol{\theta}_2 - \mathbf{\Phi}_2(s)\boldsymbol{\theta}_2) \cdot \boldsymbol{\theta}_2^\mathsf{T}\|$$

$$\overset{(b_5)}{\leq} (1+\gamma)\|\boldsymbol{\theta}_1\|^2 \cdot \|\mathbf{\Phi}_1 - \mathbf{\Phi}_2\| + \|(\gamma\mathbf{\Phi}_2(s')\boldsymbol{\theta}_1 - \mathbf{\Phi}_2(s)\boldsymbol{\theta}_1)\| \cdot \|\boldsymbol{\theta}_1 - \boldsymbol{\theta}_2\|$$
$$+ \|(\gamma\boldsymbol{\phi}_2(s')\boldsymbol{\theta}_1 - \mathbf{\Phi}_2(s)\boldsymbol{\theta}_1) - (\gamma\mathbf{\Phi}_2(s')\boldsymbol{\theta}_2 - \mathbf{\Phi}_2(s)\boldsymbol{\theta}_2)\| \cdot \|\boldsymbol{\theta}_2\|$$

$$\overset{(b_6)}{\leq} (1+\gamma)\|\boldsymbol{\theta}_1\|^2 \cdot \|\mathbf{\Phi}_1 - \mathbf{\Phi}_2\| + (1+\gamma)\|\boldsymbol{\theta}_1\| \cdot \|\boldsymbol{\theta}_1 - \boldsymbol{\theta}_2\| + (1+\gamma)\|\boldsymbol{\theta}_2\| \cdot \|\boldsymbol{\theta}_1 - \boldsymbol{\theta}_2\|$$
$$\leq (1+\gamma)\|\boldsymbol{\theta}_1\|^2 \cdot \|\mathbf{\Phi}_1 - \mathbf{\Phi}_2\| + (1+\gamma)(\|\boldsymbol{\theta}_1\| + \|\boldsymbol{\theta}_2\|) \cdot \|\boldsymbol{\theta}_1 - \boldsymbol{\theta}_2\|$$

$$\overset{(b_7)}{\leq} L_h(\|\boldsymbol{\theta}_1 - \boldsymbol{\theta}_2\| + \|\mathbf{\Phi}_1 - \mathbf{\Phi}_2\|),$$

$(b_1)$ is due to the fact that $\|\mathbf{x} + \mathbf{y}\| \leq \|\mathbf{x}\| + \|\mathbf{y}\|, \forall \mathbf{x}, \mathbf{y} \in \mathbb{R}^d$, $(b_2)$ holds due to $\|\mathbf{x} \cdot \mathbf{y}\| \leq \|\mathbf{x}\| \cdot \|\mathbf{y}\|, \forall \mathbf{x}, \mathbf{y} \in \mathbb{R}^d$, $(b_3)$ comes from the fact and $\|\mathbf{\Phi}_1(s)\| \leq 1, \|\mathbf{\Phi}_2(s)\| \leq 1 \forall s$. $(b_4) - (b_6)$ holds for the same reason as $(b_1) - (b_3)$. The last inequalty $(b_7)$ comes from by setting $L_h := \max((1+\gamma)B^2, (2+2\gamma)B)$. $\quad\square$

### E.3  PROOF OF LEMMA 4.8

*Proof.* Due to the norm-scale step (step 9) in Algorithm 2, we have

$$\|y^i(\mathbf{\Phi}_1) - y^i(\mathbf{\Phi}_2)\| \leq \max_{(\|\boldsymbol{\theta}\| \leq B, \|\boldsymbol{\theta}'\| \leq B)} \|\boldsymbol{\theta} - \boldsymbol{\theta}'\| \leq 2B. \tag{19}$$

Since the representation matrices $\mathbf{\Phi}_1$ and $\mathbf{\Phi}_2$ are of unit-norm in each row, there exists a positive constant $L_y$ such that

$$\|y^i(\mathbf{\Phi}_1) - y^i(\mathbf{\Phi}_2)\| \leq L_y\|\mathbf{\Phi}_1 - \mathbf{\Phi}_2\|. \tag{20}$$
$$\square$$

### E.4  PROOF OF LEMMA 4.10

*Proof.* In the TD learning setting for our PFEDTD-REP, at time step $k$, the state of agent $i$ is $s_k^i$, and its value function can be denoted as $V(s_k^i) = \mathbf{\Phi}(s_k^i)\boldsymbol{\theta}^i$ in a linear representation, where $\mathbf{\Phi}(s_k^i)$ is a feature vector and $\boldsymbol{\theta}^i$ is a weight vector. The goal of agent $i$ is to minimize the following loss function for every $s_k^i \in \mathcal{S}$:

$$\mathcal{L}^i(\mathbf{\Phi}(s_k^i), \boldsymbol{\theta}^i) = \frac{1}{2}\left|V(s_k^i) - \hat{V}(s_k^i)\right|^2,$$

with $\hat{V}(s_k^i) = r_k^i + \gamma\Phi(s_{k+1}^i)\theta^i$ being a constant. Therefore, to update $\Phi(s)$ and $\boldsymbol{\theta}$, we just take the natural gradient descent. Specifically, we update $\boldsymbol{\theta}$ according to (7) by taking a gradient descent step with respect to $\boldsymbol{\theta}$, with fixed $\mathbf{\Phi}$. Similarly, we update $\mathbf{\Phi}(s)$ according to (9) by taking a gradient descent step with respect to $\mathbf{\Phi}(s)$, with fixed $\boldsymbol{\theta}$.

Next, we show the convexity of the loss function $\mathcal{L}^i(\mathbf{\Phi}(s_k^i), \boldsymbol{\theta}^i)$ with respect to the feature representation $\mathbf{\Phi}(s_k^i)$ under a fixed $\boldsymbol{\theta}^i$. Since the estimated value function is approximated as $V(s_k^i) = \mathbf{\Phi}(s_k^i)\boldsymbol{\theta}^i$, where $\boldsymbol{\theta}^i$ is a fixed parameter. Taking the second-order derivative of $\mathcal{L}^i(\mathbf{\Phi}(s_k^i), \boldsymbol{\theta}^i)$ w.r.t. $\mathbf{\Phi}(s_k^i)$ will involve $\boldsymbol{\theta}^i\boldsymbol{\theta}^{i\mathsf{T}}$, which is a positive semi-definite matrix as long as $\boldsymbol{\theta}^i \neq \mathbf{0}$. Positive semi-definiteness of the Hessian implies convexity. Hence, $\mathcal{L}^i(\mathbf{\Phi}(s_k^i), \boldsymbol{\theta}^i)$ is convex on $\mathbf{\Phi}(s_k^i)$ under a fixed $\boldsymbol{\theta}^i$. This property holds vice versa, i.e., $\mathcal{L}^i(\mathbf{\Phi}(s_k^i), \boldsymbol{\theta}^i)$ is convex on $\boldsymbol{\theta}^i$ under a fixed $\mathbf{\Phi}(s_k^i)$.

Recall that the optimal solution $\boldsymbol{\Phi}_0^*$ and $\boldsymbol{\theta}^*$ is defined as the set of possible values that make the expectation of stochastic gradient $\mathbf{g}$ and $\mathbf{h}$ tends to be 0, as defined in (12), which is analogy to make the first-order gradient of loss function be 0 and achieve the local minima. The inequalities in Lemma 4.10 denote that the updates made to the feature matrix $\boldsymbol{\Phi}$ for fixed $\boldsymbol{\theta}$ in the first equation and the parameters $\boldsymbol{\theta}$ for fixed $\boldsymbol{\Phi}$ in the second equation is directed towards reducing the deviation from the optimal solutions close to initial point. As we only care about the solution to make stochastic gradients be 0, for a fixed $\boldsymbol{\theta}$, the loss function $\mathcal{L}$ is convex w.r.t. $\boldsymbol{\Phi}$, the learning process of $\boldsymbol{\Phi}$ is guaranteed to move towards decreasing the difference from an optimal point. This also holds for the update of $\boldsymbol{\theta}$. $\qquad\square$

### E.5 PROOF OF LEMMA 4.12

*Proof.* Under Lemma 4.6, we have

$$\|\mathbf{g}(\boldsymbol{\theta}, \boldsymbol{\Phi}, X) - \mathbf{g}(y^i(\boldsymbol{\Phi}^*), \boldsymbol{\Phi}^*, X)\| \le L(\|\boldsymbol{\theta} - y^i(\boldsymbol{\Phi}^*)\| + \|\boldsymbol{\Phi} - \boldsymbol{\Phi}^*\|), \forall i \in [N]. \qquad (21)$$

Similarly, under Lemma 4.7, we have

$$\|\mathbf{h}(\boldsymbol{\theta}, \boldsymbol{\Phi}, X) - \mathbf{h}(y^i(\boldsymbol{\Phi}^*), \boldsymbol{\Phi}^*, X)\| \le L(\|\boldsymbol{\theta} - y^i(\boldsymbol{\Phi}^*)\| + \|\boldsymbol{\Phi} - \boldsymbol{\Phi}^*\|), \forall i \in [N]. \qquad (22)$$

Let $L_1 = \max(L, \max_X \mathbf{g}(y^i(\boldsymbol{\Phi}^*), \boldsymbol{\Phi}^*, X), \max_X \mathbf{h}(y^i(\boldsymbol{\Phi}^*), \boldsymbol{\Phi}^*, X))$, then according to (21)-(22), we have

$$\|\mathbf{g}(\boldsymbol{\theta}, \boldsymbol{\Phi})\| \le L_1(\|\boldsymbol{\theta} - y^i(\boldsymbol{\Phi}^*)\| + \|\boldsymbol{\Phi} - \boldsymbol{\Phi}^*\| + 1),$$

and

$$\|\mathbf{h}(\boldsymbol{\theta}, \boldsymbol{\Phi})\| \le L_1(\|\boldsymbol{\theta} - y^i(\boldsymbol{\Phi}^*)\| + \|\boldsymbol{\Phi} - \boldsymbol{\Phi}^*\| + 1).$$

Denote $h^j(\boldsymbol{\theta}, \boldsymbol{\phi}, X)$ as the $j$-th element of $\mathbf{h}(\boldsymbol{\theta}, \boldsymbol{\Phi}, X)$. Following Chen et al. (2019), we can show that $\boldsymbol{\theta} \in \mathbb{R}^d$, $\boldsymbol{\Phi} \in \mathbb{R}^{|\mathcal{S}| \times d}$, and $x \in \mathcal{X}$,

$$\|\mathbb{E}[\mathbf{h}(\boldsymbol{\theta}, \boldsymbol{\Phi}, X)|X_0 = x] - \mathbb{E}_\mu[\mathbf{h}(\boldsymbol{\theta}, \boldsymbol{\Phi}, X)]\|$$

$$\le \sum_{j=1}^d |\mathbb{E}[h^j(\boldsymbol{\theta}, \boldsymbol{\lambda}, X)|X_0 = x] - \mathbb{E}_\mu[h^j(\boldsymbol{\theta}, \boldsymbol{\Phi}, X)]|$$

$$\le 2L_1(\|\boldsymbol{\theta} - y^i(\boldsymbol{\Phi}^*)\| + \|\boldsymbol{\Phi} - \boldsymbol{\Phi}^*\| + 1) \sum_{j=1}^d \left| \mathbb{E}\left[ \frac{h^j(\boldsymbol{\theta}, \boldsymbol{\Phi}, X)}{2L_1(\|\boldsymbol{\theta} - y^i(\boldsymbol{\Phi}^*)\| + \|\lambda - \lambda^*\| + 1)} \Big| X_0 = x \right] \right.$$

$$\left. - \mathbb{E}_\mu\left[ \frac{h^j(\boldsymbol{\theta}, \boldsymbol{\Phi}, X)}{2L_1(\|\boldsymbol{\theta} - y^i(\boldsymbol{\Phi}^*)\| + \|\lambda - \lambda^*\| + 1)} \right] \right|$$

$$\le 2L_1(\|\boldsymbol{\theta} - y^i(\boldsymbol{\Phi}^*)\| + \|\boldsymbol{\Phi} - \boldsymbol{\Phi}^*\| + 1)dC_1\rho_1^k,$$

where the last inequality holds due to Assumption 4.3 with constants $C_1 > 0$ and $\rho_1 \in (0, 1)$. To guarantee $2L_1(\|\boldsymbol{\theta} - y^i(\boldsymbol{\Phi}^*)\| + \|\boldsymbol{\Phi} - \boldsymbol{\Phi}^*\| + 1)dC_1\rho_1^k \le \delta(\|\boldsymbol{\theta} - y^i(\boldsymbol{\Phi}^*)\| + \|\boldsymbol{\Phi} - \boldsymbol{\Phi}^*\| + 1)$, we have

$$\tau_\delta \le \frac{\log(1/\delta) + \log(2L_1C_1d)}{\log(1/\rho_1)}. \qquad (23)$$

Using the same procedures we can show that

$$\|\mathbb{E}[\mathbf{g}(\boldsymbol{\theta}, \boldsymbol{\Phi}, X)|X_0 = x] - \mathbb{E}_\mu[\mathbf{g}(\boldsymbol{\theta}, \boldsymbol{\Phi}, X)]\| \le 2L_1(\|\boldsymbol{\theta} - y^i(\boldsymbol{\Phi}^*)\| + \|\boldsymbol{\Phi} - \boldsymbol{\Phi}^*\| + 1)dC_2\rho_2^k,$$

hence we have

$$\tau_\delta \le \frac{\log(1/\delta) + \log(2L_1C_2d)}{\log(1/\rho_2)}. \qquad (24)$$

By setting $\tau_\delta$ as the largest value in (23) and (24), we arrive at the final result in Lemma 4.12. $\qquad\square$

# F PROOFS OF MAIN RESULTS

## F.1 PROOF OF THEOREM 4.13

For notational simplicity, in the proofs, we use $\mathbf{h}(\boldsymbol{\theta}_{t+1}^i, \boldsymbol{\Phi}_t)$ to denote $\mathbf{h}(\boldsymbol{\theta}_{t+1}^i, \boldsymbol{\Phi}_t, \{X_{t,k-1}^i\}_{k=1}^K)$, and $\mathbf{g}(\boldsymbol{\theta}_{t,k-1}^i, \boldsymbol{\Phi}_t)$ to denote $\mathbf{g}(\boldsymbol{\theta}_{t,k-1}^i, \boldsymbol{\Phi}_t, X_{t,k-1}^i)$. In the following, we first focus on the update of the global representation $\boldsymbol{\Phi}_t$ and characterize the drift of it.

### F.1.1 DRIFT OF $\boldsymbol{\Phi}_t$

The drift of $\boldsymbol{\Phi}_t$ is given in the following lemma.

**Lemma F.1.** *The drift between $\boldsymbol{\Phi}_{t+1}$ and $\boldsymbol{\Phi}_t$ is given by*

$$\mathbb{E}[\|\boldsymbol{\Phi}_{t+1} - \boldsymbol{\Phi}^*\|^2]$$

$$= \mathbb{E}[\|\boldsymbol{\Phi}_t - \boldsymbol{\Phi}^*\|^2] + \underbrace{\frac{\beta_t^2}{N^2}\mathbb{E}\left[\left\|\sum_{i=1}^N \mathbf{h}(\boldsymbol{\theta}_{t+1}^i, \boldsymbol{\Phi}_t)\right\|^2\right]}_{\text{Term 1}} + \underbrace{2\beta_t \mathbb{E}\left[\left\langle \boldsymbol{\Phi}^* - \boldsymbol{\Phi}_t, \frac{-1}{N}\sum_{i=1}^N \bar{\mathbf{h}}(\boldsymbol{\theta}_{t+1}^i, \boldsymbol{\Phi}_t)\right\rangle\right]}_{\text{Term 2}}$$

$$+ \underbrace{2\beta_t \mathbb{E}\left[\left\langle \boldsymbol{\Phi}_t - \boldsymbol{\Phi}^*, \frac{1}{N}\sum_{i=1}^N \mathbf{h}(\boldsymbol{\theta}_{t+1}^i, \boldsymbol{\Phi}_t) - \bar{\mathbf{h}}(\boldsymbol{\theta}_{t+1}^i, \boldsymbol{\Phi}_t)\right\rangle\right]}_{\text{Term 3}}. \tag{25}$$

*Proof.* Based on the update of $\boldsymbol{\Phi}_t$ in (11), We have the following equation

$$\mathbb{E}[\|\boldsymbol{\Phi}_{t+1} - \boldsymbol{\Phi}^*\|^2] - \mathbb{E}[\|\boldsymbol{\Phi}_t - \boldsymbol{\Phi}^*\|^2]$$

$$= \mathbb{E}[\|\boldsymbol{\Phi}^*\|^2 + \|\boldsymbol{\Phi}_{t+1}\|^2 - 2\langle \boldsymbol{\Phi}^*, \boldsymbol{\Phi}_{t+1}\rangle] - \mathbb{E}[\|\boldsymbol{\Phi}^*\|^2 + \|\boldsymbol{\Phi}_t\|^2 - 2\langle \boldsymbol{\Phi}^*, \boldsymbol{\Phi}_t\rangle]$$

$$= \mathbb{E}[\|\boldsymbol{\Phi}_{t+1}\|^2] - \mathbb{E}[\|\boldsymbol{\Phi}_t\|^2] - 2\langle \boldsymbol{\Phi}^*, \boldsymbol{\Phi}_{t+1} - \boldsymbol{\Phi}_t\rangle]$$

$$= \mathbb{E}[\langle \boldsymbol{\Phi}_{t+1} - \boldsymbol{\Phi}_t, \boldsymbol{\Phi}_{t+1} + \boldsymbol{\Phi}_t\rangle] - 2\langle \boldsymbol{\Phi}^*, \boldsymbol{\Phi}_{t+1} - \boldsymbol{\Phi}_t\rangle]$$

$$= \mathbb{E}[\langle \boldsymbol{\Phi}_{t+1} - \boldsymbol{\Phi}_t, \boldsymbol{\Phi}_{t+1} - \boldsymbol{\Phi}_t\rangle] + 2\mathbb{E}[\langle \boldsymbol{\Phi}_{t+1} - \boldsymbol{\Phi}_t, \boldsymbol{\Phi}_t\rangle] - 2\langle \boldsymbol{\Phi}^*, \boldsymbol{\Phi}_{t+1} - \boldsymbol{\Phi}_t\rangle]$$

$$= \frac{\beta_t^2}{N^2}\mathbb{E}\left[\left\|\sum_{i=1}^N \mathbf{h}(\boldsymbol{\theta}_{t+1}^i, \boldsymbol{\Phi}_t)\right\|^2\right] - 2\beta_t \mathbb{E}\left[\left\langle \boldsymbol{\Phi}^* - \boldsymbol{\Phi}_t, \frac{1}{N}\sum_{i=1}^N \mathbf{h}(\boldsymbol{\theta}_{t+1}^i, \boldsymbol{\Phi}_t)\right\rangle\right], \tag{26}$$

which directly leads to

$$\mathbb{E}[\|\boldsymbol{\Phi}_{t+1} - \boldsymbol{\Phi}^*\|^2]$$

$$= \mathbb{E}[\|\boldsymbol{\Phi}_t - \boldsymbol{\Phi}^*\|^2] + \frac{\beta_t^2}{N^2}\mathbb{E}\left[\left\|\sum_{i=1}^N \mathbf{h}(\boldsymbol{\theta}_{t+1}^i, \boldsymbol{\Phi}_t)\right\|^2\right] - 2\beta_t \mathbb{E}\left[\left\langle \boldsymbol{\Phi}^* - \boldsymbol{\Phi}_t, \frac{1}{N}\sum_{i=1}^N \mathbf{h}(\boldsymbol{\theta}_{t+1}^i, \boldsymbol{\Phi}_t)\right\rangle\right]. \tag{27}$$

Rearranging the last term yields the desired result. $\qquad\square$

In the following, we separately bound Term 1 to Term 3. We first bound Term 1 as follows.

**Lemma F.2.** *For any $t \geq \tau$, we have*

$$\text{Term 1} \leq 4\beta_t^2(L^2 + L^4)\mathbb{E}[\|\boldsymbol{\Phi}^* - \boldsymbol{\Phi}_t\|^2] + \frac{4\beta_t^2 L^2}{N}\mathbb{E}\left[\sum_{i=1}^N \|\boldsymbol{\theta}_{t+1} - y^i(\boldsymbol{\Phi}_t)\|^2\right] + 4\beta_t^2 \delta^2 \tag{28}$$

*Proof.* Note that

$$\text{Term 1} = \frac{\beta_t^2}{N^2}\mathbb{E}\left[\left\|\sum_{i=1}^N \mathbf{h}(\boldsymbol{\theta}_{t+1}^i, \boldsymbol{\Phi}_t) - \sum_{i=1}^N \mathbf{h}(y^i(\boldsymbol{\Phi}_t), \boldsymbol{\Phi}^*) + \sum_{i=1}^N \mathbf{h}(y^i(\boldsymbol{\Phi}_t), \boldsymbol{\Phi}^*)\right\|^2\right]$$

$$\overset{\text{triangle inequality}}{\leq} \frac{2\beta_t^2}{N^2}\mathbb{E}\left[\underbrace{\left\|\sum_{i=1}^N \mathbf{h}(\boldsymbol{\theta}_{t+1}^i, \boldsymbol{\Phi}_t) - \sum_{i=1}^N \mathbf{h}(y^i(\boldsymbol{\Phi}_t), \boldsymbol{\Phi}^*)\right\|^2}_{\text{Lipschitz property of } \mathbf{h}}\right]$$

$$+ \frac{2\beta_t^2}{N^2}\mathbb{E}\left[\left\|\sum_{i=1}^N \mathbf{h}(y^i(\boldsymbol{\Phi}_t), \boldsymbol{\Phi}^*)\right\|^2\right]$$

$$\overset{(a_1)}{\leq} \frac{2\beta_t^2 L^2}{N^2}\mathbb{E}\left[2N\sum_{i=1}^N \left\|(\boldsymbol{\theta}_{t+1}^i - y^i(\boldsymbol{\Phi}_t))\right\|^2 + 2N^2\left\|(\boldsymbol{\Phi}_t - \boldsymbol{\Phi}^*)\right\|^2\right]$$

$$+ \frac{2\beta_t^2}{N^2}\mathbb{E}\left[\left\|\sum_{i=1}^N \mathbf{h}(y^i(\boldsymbol{\Phi}_t), \boldsymbol{\Phi}^*) - \sum_{i=1}^N \mathbf{h}(y^i(\boldsymbol{\Phi}^*), \boldsymbol{\Phi}^*) + \sum_{i=1}^N \mathbf{h}(y^i(\boldsymbol{\Phi}^*), \boldsymbol{\Phi}^*)\right\|^2\right]$$

$$\leq 4\beta_t^2 L^2\mathbb{E}[\|\boldsymbol{\Phi}^* - \boldsymbol{\Phi}_t\|^2] + \frac{4\beta_t^2 L^2}{N}\mathbb{E}\left[\sum_{i=1}^N \|\boldsymbol{\theta}_{t+1}^i - y^i(\boldsymbol{\Phi}_t)\|^2\right]$$

$$+ \frac{4\beta_t^2}{N^2}\mathbb{E}\left[\underbrace{\left\|\sum_{i=1}^N \mathbf{h}(y^i(\boldsymbol{\Phi}_t), \boldsymbol{\Phi}^*) - \sum_{i=1}^N \mathbf{h}(y^i(\boldsymbol{\Phi}^*), \boldsymbol{\Phi}^*)\right\|^2}_{\text{Lipschitz of } \mathbf{h}, \, y^i}\right]$$

$$+ \frac{4\beta_t^2}{N^2}\mathbb{E}\left[\left\|\sum_{i=1}^N \mathbf{h}(y^i(\boldsymbol{\Phi}^*), \boldsymbol{\Phi}^*)\right\|^2\right]$$

$$\overset{(a_2)}{\leq} 4\beta_t^2 L^2\mathbb{E}[\|\boldsymbol{\Phi}^* - \boldsymbol{\Phi}_t\|^2] + \frac{4\beta_t^2 L^2}{N}\mathbb{E}\left[\sum_{i=1}^N \|\boldsymbol{\theta}_{t+1}^i - y^i(\boldsymbol{\Phi}_t)\|^2\right]$$

$$+ 4\beta_t^2 L^4\mathbb{E}\left[\|\boldsymbol{\Phi}_t - \boldsymbol{\Phi}^*\|^2\right] + \frac{4\beta_t^2}{N^2}\mathbb{E}\left[\left\|\sum_{i=1}^N \mathbf{h}(y^i(\boldsymbol{\Phi}^*), \boldsymbol{\Phi}^*) - \sum_{i=1}^N \bar{\mathbf{h}}(y^i(\boldsymbol{\Phi}^*), \boldsymbol{\Phi}^*)\right\|^2\right]$$

$$\overset{(a_3)}{\leq} 4\beta_t^2(L^2 + L^4)\mathbb{E}[\|\boldsymbol{\Phi}^* - \boldsymbol{\Phi}_t\|^2] + \frac{4\beta_t^2 L^2}{N}\mathbb{E}\left[\sum_{i=1}^N \|\boldsymbol{\theta}_{t+1} - y^i(\boldsymbol{\Phi}_t)\|^2\right] + 4\beta_t^2\delta^2,$$

where the $(a_1)$ is due to $\|\sum_{i=1}^N \mathbf{x}_i\|^2 \leq N\sum_{i=1}^N \|\mathbf{x}_i\|^2$, $(a_2)$ is due to the Lipschitz property of functions $\mathbf{h}$ and $y^i$, and $(a_3)$ holds based on the mixing time property in Definition 4.3.

$\square$

Next, we bound Term 2 in the following lemma.

**Lemma F.3.** *We have*

$$\text{Term 2} \leq \beta_t(L/\alpha_t - 2\omega)\mathbb{E}[\|\boldsymbol{\Phi}^* - \boldsymbol{\Phi}_t\|^2] + \frac{\beta_t\alpha_t L}{N}\mathbb{E}\left[\sum_{i=1}^N \|\boldsymbol{\theta}_{t+1}^i - y^i(\boldsymbol{\Phi}_t)\|^2\right]. \tag{29}$$

*Proof.* We have

$$\text{Term 2} = 2\beta_t\mathbb{E}\left[\langle \boldsymbol{\Phi}^* - \boldsymbol{\Phi}_t, \frac{-1}{N}\sum_{i=1}^N \bar{\mathbf{h}}(\boldsymbol{\theta}_{t+1}^i, \boldsymbol{\Phi}_t)\rangle\right]$$

$$= 2\beta_t\mathbb{E}\left[\langle \boldsymbol{\Phi}^* - \boldsymbol{\Phi}_t, \frac{-1}{N}\sum_{i=1}^N \bar{\mathbf{h}}(y^i(\boldsymbol{\Phi}_t), \boldsymbol{\Phi}_t)\rangle\right]$$

$$+ 2\beta_t \mathbb{E}\left[\left\langle \mathbf{\Phi}^* - \mathbf{\Phi}_t, \underbrace{\frac{1}{N}\sum_{i=1}^{N} \bar{\mathbf{h}}(y^i(\mathbf{\Phi}_t), \mathbf{\Phi}_t) - \bar{\mathbf{h}}(\boldsymbol{\theta}_{t+1}^i, \mathbf{\Phi}_t)}_{\text{Lipschitz of } \mathbf{h}}\right\rangle\right]$$

$$\leq 2\beta_t \mathbb{E}\left[\left\langle \mathbf{\Phi}^* - \mathbf{\Phi}_t, \frac{-1}{N}\sum_{i=1}^{N} \bar{\mathbf{h}}(y^i(\mathbf{\Phi}_t), \mathbf{\Phi}_t)\right\rangle\right] + 2\beta_t L \mathbb{E}\left[\left\langle \mathbf{\Phi}^* - \mathbf{\Phi}_t, \frac{1}{N}\sum_{i=1}^{N}(y^i(\mathbf{\Phi}_t) - \boldsymbol{\theta}_{t+1}^i)\right\rangle\right]$$

$$\overset{(b_1)}{\leq} 2\beta_t \mathbb{E}\left[\left\langle \mathbf{\Phi}^* - \mathbf{\Phi}_t, \frac{-1}{N}\sum_{i=1}^{N} \bar{\mathbf{h}}(y^i(\mathbf{\Phi}_t), \mathbf{\Phi}_t)\right\rangle\right] + \beta_t L/\alpha_t \mathbb{E}[\|\mathbf{\Phi}^* - \mathbf{\Phi}_t\|^2]$$

$$+ \frac{\beta_t \alpha_t L}{N^2} \mathbb{E}\left[\|\sum_{i=1}^{N}(\boldsymbol{\theta}_{t+1}^i - y^i(\mathbf{\Phi}_t))\|^2\right]$$

$$\overset{(b_2)}{\leq} 2\beta_t \mathbb{E}\left[\left\langle \mathbf{\Phi}_t - \mathbf{\Phi}^*, \frac{1}{N}\sum_{i=1}^{N} \bar{\mathbf{h}}(y^i(\mathbf{\Phi}_t), \mathbf{\Phi}_t)\right\rangle\right] + \beta_t L/\alpha_t \mathbb{E}[\|\mathbf{\Phi}^* - \mathbf{\Phi}_t\|^2]$$

$$+ \frac{\beta_t \alpha_t L}{N} \mathbb{E}\left[\sum_{i=1}^{N} \|\boldsymbol{\theta}_{t+1}^i - y^i(\mathbf{\Phi}_t)\|^2\right]$$

$$\leq \beta_t(L/\alpha_t - 2\omega)\mathbb{E}[\|\mathbf{\Phi}^* - \mathbf{\Phi}_t\|^2] + \frac{\beta_t \alpha_t L}{N} \mathbb{E}\left[\sum_{i=1}^{N} \|\boldsymbol{\theta}_{t+1}^i - y^i(\mathbf{\Phi}_t)\|^2\right],$$

where $(b_1)$ holds because $2\mathbf{x}^T\mathbf{y} \leq \beta\|\mathbf{x}\|^2 + 1/\beta\|\mathbf{y}\|^2, \forall \beta > 0$, $(b_2)$ is due to $\|\sum_{i=1}^{N} \mathbf{x}_i\|^2 \leq N\sum_{i=1}^{N}\|\mathbf{x}_i\|^2$, and the last inequality is due to Assumption 4.10.

$\square$

Next, we bound Term 3 in the following lemmas.

**Lemma F.4.** *For all $t \geq \tau$ we have*

$$\text{Term 3} \leq (7\beta_t/\alpha_t + 2\beta_t\alpha_t L^2 + 6\beta_t\alpha_t\delta^2)\mathbb{E}[\|\mathbf{\Phi}_{t-\tau} - \mathbf{\Phi}_t\|^2]$$
$$+ (6\beta_t/\alpha_t + 6\beta_t\alpha_t\delta^2(1+L^2) + 4\beta_t\alpha_t L^2(3+4L^2))\mathbb{E}[\|\mathbf{\Phi}_t - \mathbf{\Phi}^*\|^2]$$
$$+ \frac{16\beta_t\alpha_t L^2 + 6\beta_t\alpha_t\delta^2}{N}\mathbb{E}\left[\sum_{i=1}^{N}\|\boldsymbol{\theta}^{i,*} - \boldsymbol{\theta}_{t+1}\|^2\right] + 11\beta_t\alpha_t\delta^2. \tag{30}$$

*Proof.* We first decompose Term 3 as follows

$$\text{Term 3} = 2\beta_t \mathbb{E}\left[\left\langle \mathbf{\Phi}_t - \mathbf{\Phi}^*, \frac{1}{N}\sum_{i=1}^{N} \mathbf{h}(\boldsymbol{\theta}_{t+1}^i, \mathbf{\Phi}_t) - \bar{\mathbf{h}}(\boldsymbol{\theta}_{t+1}^i, \mathbf{\Phi}_t)\right\rangle\right]$$

$$= 2\beta_t \mathbb{E}\left[\left\langle \mathbf{\Phi}_t - \mathbf{\Phi}_{t-\tau}, \frac{1}{N}\sum_{i=1}^{N} \mathbf{h}(\boldsymbol{\theta}_{t+1}^i, \mathbf{\Phi}_t) - \bar{\mathbf{h}}(\boldsymbol{\theta}_{t+1}^i, \mathbf{\Phi}_t)\right\rangle\right]$$

$$+ 2\beta_t \mathbb{E}\left[\left\langle \mathbf{\Phi}_{t-\tau} - \mathbf{\Phi}^*, \frac{1}{N}\sum_{i=1}^{N} \mathbf{h}(\boldsymbol{\theta}_{t+1}^i, \mathbf{\Phi}_t) - \bar{\mathbf{h}}(\boldsymbol{\theta}_{t+1}^i, \mathbf{\Phi}_t)\right\rangle\right]$$

$$= \underbrace{2\beta_t \mathbb{E}\left[\left\langle \mathbf{\Phi}_t - \mathbf{\Phi}_{t-\tau}, \frac{1}{N}\sum_{i=1}^{N} \mathbf{h}(\boldsymbol{\theta}_{t+1}^i, \mathbf{\Phi}_t) - \bar{\mathbf{h}}(\boldsymbol{\theta}_{t+1}^i, \mathbf{\Phi}_t)\right\rangle\right]}_{C_1}$$

$$+ \underbrace{2\beta_t \mathbb{E}\left[\left\langle \mathbf{\Phi}_{t-\tau} - \mathbf{\Phi}^*, \frac{1}{N}\sum_{i=1}^{N} \mathbf{h}(\boldsymbol{\theta}_{t+1}^i, \mathbf{\Phi}_t) - \frac{1}{N}\sum_{i=1}^{N} \mathbf{h}(\boldsymbol{\theta}_{t+1}^i, \mathbf{\Phi}_{t-\tau})\right\rangle\right]}_{C_2}$$

$$+ 2\beta_t \mathbb{E}\left[\underbrace{\langle \boldsymbol{\Phi}_{t-\tau} - \boldsymbol{\Phi}^*, \frac{1}{N}\sum_{i=1}^N \mathbf{h}(\boldsymbol{\theta}_{t+1}^i, \boldsymbol{\Phi}_{t-\tau}) - \frac{1}{N}\sum_{i=1}^N \bar{\mathbf{h}}(\boldsymbol{\theta}_{t+1}^i, \boldsymbol{\Phi}_{t-\tau})\rangle}_{C_3}\right]$$

$$+ 2\beta_t \mathbb{E}\left[\underbrace{\langle \boldsymbol{\Phi}_{t-\tau} - \boldsymbol{\Phi}^*, \frac{1}{N}\sum_{i=1}^N \bar{\mathbf{h}}(\boldsymbol{\theta}_{t+1}^i, \boldsymbol{\Phi}_{t-\tau}) - \frac{1}{N}\sum_{i=1}^N \bar{\mathbf{h}}(\boldsymbol{\theta}_{t+1}^i, \boldsymbol{\Phi}_t)\rangle}_{C_4}\right].$$

Next, we bound $C_1$ as

$$C_1 = 2\beta_t \mathbb{E}\left[\langle \boldsymbol{\Phi}_t - \boldsymbol{\Phi}_{t-\tau}, \frac{1}{N}\sum_{i=1}^N \mathbf{h}(\boldsymbol{\theta}_{t+1}^i, \boldsymbol{\Phi}_t) - \bar{\mathbf{h}}(\boldsymbol{\theta}_{t+1}^i, \boldsymbol{\Phi}_t)\rangle\right]$$

$$\leq \beta_t/\alpha_t \mathbb{E}[\|\boldsymbol{\Phi}_t - \boldsymbol{\Phi}_{t-\tau}\|^2] + \beta_t \alpha_t \mathbb{E}\left[\left\|\frac{1}{N}\sum_{i=1}^N \mathbf{h}(\boldsymbol{\theta}_{t+1}^i, \boldsymbol{\Phi}_t) - \bar{\mathbf{h}}(\boldsymbol{\theta}_{t+1}^i, \boldsymbol{\Phi}_t) + \bar{h}(y^i(\boldsymbol{\Phi}^*), \boldsymbol{\Phi}^*)\right\|^2\right]$$

$$\leq \beta_t/\alpha_t \mathbb{E}[\|\boldsymbol{\Phi}_t - \boldsymbol{\Phi}_{t-\tau}\|^2] + 2\beta_t \alpha_t \mathbb{E}\left[\left\|\frac{1}{N}\sum_{i=1}^N \mathbf{h}(\boldsymbol{\theta}_{t+1}^i, \boldsymbol{\Phi}_t)\right\|^2\right]$$

$$+ 2\beta_t \alpha_t \mathbb{E}\left[\left\|\frac{1}{N}\sum_{i=1}^N \bar{\mathbf{h}}(y^i(\boldsymbol{\Phi}^*), \boldsymbol{\Phi}^*) - \bar{\mathbf{h}}(\boldsymbol{\theta}_{t+1}^i, \boldsymbol{\Phi}_t)\right\|^2\right]$$

$$= \beta_t/\alpha_t \mathbb{E}[\|\boldsymbol{\Phi}_t - \boldsymbol{\Phi}_{t-\tau}\|^2] + \frac{2\beta_t \alpha_t}{N^2} \mathbb{E}\left[\left\|\sum_{i=1}^N \mathbf{h}(\boldsymbol{\theta}_{t+1}^i, \boldsymbol{\Phi}_t)\right\|^2\right]$$

$$+ 2\beta_t \alpha_t \mathbb{E}\left[\left\|\frac{1}{N}\sum_{i=1}^N \bar{\mathbf{h}}(y^i(\boldsymbol{\Phi}^*), \boldsymbol{\Phi}^*) - \bar{\mathbf{h}}(\boldsymbol{\theta}_{t+1}^i, \boldsymbol{\Phi}_t)\right\|^2\right]$$

$$\overset{\text{Lemma F.2}}{\leq} \beta_t/\alpha_t \mathbb{E}[\|\boldsymbol{\Phi}_t - \boldsymbol{\Phi}_{t-\tau}\|^2] + 8\beta_t \alpha_t (L^2 + L^4) \mathbb{E}[\|\boldsymbol{\Phi}^* - \boldsymbol{\Phi}_t\|^2]$$

$$+ \frac{8\beta_t \alpha_t L^2}{N} \mathbb{E}\left[\sum_{i=1}^N \|\boldsymbol{\theta}_{t+1}^i - y^i(\boldsymbol{\Phi}_t)\|^2\right] + 8\beta_t \alpha_t \delta^2$$

$$+ 2\beta_t \alpha_t \underbrace{\mathbb{E}\left[\left\|\frac{1}{N}\sum_{i=1}^N \bar{\mathbf{h}}(y^i(\boldsymbol{\Phi}^*), \boldsymbol{\Phi}^*) - \bar{\mathbf{h}}(\boldsymbol{\theta}_{t+1}^i, \boldsymbol{\Phi}_t)\right\|^2\right]}_{\text{Lipschitz of } \mathbf{h}}$$

$$\leq \beta_t/\alpha_t \mathbb{E}[\|\boldsymbol{\Phi}_t - \boldsymbol{\Phi}_{t-\tau}\|^2] + 8\beta_t \alpha_t (L^2 + L^4) \mathbb{E}[\|\boldsymbol{\Phi}^* - \boldsymbol{\Phi}_t\|^2]$$

$$+ \frac{8\beta_t \alpha_t L^2}{N} \mathbb{E}\left[\sum_{i=1}^N \|\boldsymbol{\theta}_{t+1}^i - y^i(\boldsymbol{\Phi}_t)\|^2\right] + 8\beta_t \alpha_t \delta^2$$

$$+ 2\beta_t \alpha_t L^2 \mathbb{E}\left[\left\|\frac{1}{N}\sum_{i=1}^N 2(\boldsymbol{\Phi}^* - \boldsymbol{\Phi}_t) + 2(\boldsymbol{\theta}_{t+1}^i - y^i(\boldsymbol{\Phi}^*))\right\|^2\right]$$

$$\leq \beta_t/\alpha_t \mathbb{E}[\|\boldsymbol{\Phi}_t - \boldsymbol{\Phi}_{t-\tau}\|^2] + 8\beta_t \alpha_t (L^2 + L^4) \mathbb{E}[\|\boldsymbol{\Phi}^* - \boldsymbol{\Phi}_t\|^2]$$

$$+ \frac{8\beta_t \alpha_t L^2}{N} \mathbb{E}\left[\sum_{i=1}^N \|\boldsymbol{\theta}_{t+1}^i - y^i(\boldsymbol{\Phi}_t)\|^2\right] + 8\beta_t \alpha_t \delta^2$$

$$+ 4\beta_t \alpha_t L^2 \mathbb{E}[\|\boldsymbol{\Phi}^* - \boldsymbol{\Phi}_t\|^2] + \frac{4\beta_t \alpha_t L^2}{N} \mathbb{E}\left[\sum_{i=1}^N \|\boldsymbol{\theta}_{t+1}^i - y^i(\boldsymbol{\Phi}^*)\|^2\right]$$

$$= \beta_t/\alpha_t \mathbb{E}[\|\boldsymbol{\Phi}_t - \boldsymbol{\Phi}_{t-\tau}\|^2] + 8\beta_t \alpha_t (L^2 + L^4) \mathbb{E}[\|\boldsymbol{\Phi}^* - \boldsymbol{\Phi}_t\|^2]$$

$$+ \frac{8\beta_t\alpha_t L^2}{N} \mathbb{E}\left[\sum_{i=1}^{N} \|\boldsymbol{\theta}_{t+1}^i - y^i(\boldsymbol{\Phi}_t)\|^2\right] + 8\beta_t\alpha_t\delta^2 + 4\beta_t\alpha_t L^2 \mathbb{E}[\|\boldsymbol{\Phi}^* - \boldsymbol{\Phi}_t\|^2]$$

$$+ \frac{4\beta_t\alpha_t L^2}{N} \mathbb{E}\left[\sum_{i=1}^{N} \|\boldsymbol{\theta}_{t+1}^i - y^i(\boldsymbol{\Phi}_t) + y^i(\boldsymbol{\Phi}_t) - y^i(\boldsymbol{\Phi}^*)\|^2\right]$$

$$\leq \beta_t/\alpha_t \mathbb{E}[\|\boldsymbol{\Phi}_t - \boldsymbol{\Phi}_{t-\tau}\|^2] + 8\beta_t\alpha_t(L^2 + L^4)\mathbb{E}[\|\boldsymbol{\Phi}^* - \boldsymbol{\Phi}_t\|^2]$$

$$+ \frac{8\beta_t\alpha_t L^2}{N} \mathbb{E}\left[\sum_{i=1}^{N} \|\boldsymbol{\theta}_{t+1}^i - y^i(\boldsymbol{\Phi}_t)\|^2\right] + 8\beta_t\alpha_t\delta^2$$

$$+ 4\beta_t\alpha_t L^2 \mathbb{E}[\|\boldsymbol{\Phi}^* - \boldsymbol{\Phi}_t\|^2] + \frac{8\beta_t\alpha_t L^2}{N} \mathbb{E}\left[\sum_{i=1}^{N} \|\boldsymbol{\theta}_{t+1}^i - y^i(\boldsymbol{\Phi}_t)\|^2\right]$$

$$+ 8\beta_t\alpha_t L^4 \mathbb{E}[\|\boldsymbol{\Phi}^* - \boldsymbol{\Phi}_t\|^2]$$

$$= \beta_t/\alpha_t \mathbb{E}[\|\boldsymbol{\Phi}_t - \boldsymbol{\Phi}_{t-\tau}\|^2] + 4\beta_t\alpha_t L^2(3 + 4L^2)\mathbb{E}[\|\boldsymbol{\Phi}^* - \boldsymbol{\Phi}_t\|^2]$$

$$+ \frac{16\beta_t\alpha_t L^2}{N} \mathbb{E}\left[\sum_{i=1}^{N} \|\boldsymbol{\theta}_{t+1}^i - y^i(\boldsymbol{\Phi}_t)\|^2\right] + 8\beta_t\alpha_t\delta^2,$$

where the last inequality is due to the Lipschitz of the function $y^i$.

Next, we bound $C_2$ as follows.

$$C_2 = 2\beta_t\mathbb{E}\left[\langle\boldsymbol{\Phi}_{t-\tau} - \boldsymbol{\Phi}^*, \frac{1}{N}\sum_{i=1}^{N}\mathbf{h}(\boldsymbol{\theta}_{t+1}^i, \boldsymbol{\Phi}_t) - \frac{1}{N}\sum_{i=1}^{N}\mathbf{h}(\boldsymbol{\theta}_{t+1}^i, \boldsymbol{\Phi}_{t-\tau})\rangle\right]$$

$$\leq \beta_t/\alpha_t\mathbb{E}[\|\boldsymbol{\Phi}_{t-\tau} - \boldsymbol{\Phi}^*\|^2] + \beta_t\alpha_t\mathbb{E}\underbrace{\left[\left\|\frac{1}{N}\sum_{i=1}^{N}\mathbf{h}(\boldsymbol{\theta}_{t+1}^i, \boldsymbol{\Phi}_t) - \frac{1}{N}\sum_{i=1}^{N}\mathbf{h}(\boldsymbol{\theta}_{t+1}^i, \boldsymbol{\Phi}_{t-\tau})\right\|^2\right]}_{\text{Lipschitz of } \mathbf{h}}$$

$$\leq \beta_t/\alpha_t\mathbb{E}[\|\boldsymbol{\Phi}_{t-\tau} - \boldsymbol{\Phi}^*\|^2] + \beta_t\alpha_t L^2\mathbb{E}[\|\boldsymbol{\Phi}_t - \boldsymbol{\Phi}_{t-\tau}\|^2]$$

$$= \beta_t/\alpha_t\mathbb{E}[\|\boldsymbol{\Phi}_{t-\tau} - \boldsymbol{\Phi}_t + \boldsymbol{\Phi}_t - \boldsymbol{\Phi}^*\|^2] + \beta_t\alpha_t L^2\mathbb{E}[\|\boldsymbol{\Phi}_t - \boldsymbol{\Phi}_{t-\tau}\|^2]$$

$$\leq 2\beta_t/\alpha_t\mathbb{E}[\|\boldsymbol{\Phi}_{t-\tau} - \boldsymbol{\Phi}_t\|^2] + 2\beta_t/\alpha_t\mathbb{E}[\|\boldsymbol{\Phi}_t - \boldsymbol{\Phi}^*\|^2] + \beta_t\alpha_t L^2\mathbb{E}[\|\boldsymbol{\Phi}_t - \boldsymbol{\Phi}_{t-\tau}\|^2]$$

$$= (2\beta_t/\alpha_t + \beta_t\alpha_t L^2)\mathbb{E}[\|\boldsymbol{\Phi}_{t-\tau} - \boldsymbol{\Phi}_t\|^2] + 2\beta_t/\alpha_t\mathbb{E}[\|\boldsymbol{\Phi}_t - \boldsymbol{\Phi}^*\|^2].$$

Similarly, $C_4$ is bounded exactly same as $C_2$, i.e.,

$$C_4 \leq (2\beta_t/\alpha_t + \beta_t\alpha_t L^2)\mathbb{E}[\|\boldsymbol{\Phi}_{t-\tau} - \boldsymbol{\Phi}_t\|^2] + 2\beta_t/\alpha_t\mathbb{E}[\|\boldsymbol{\Phi}_t - \boldsymbol{\Phi}^*\|^2].$$

Next, we bound $C_3$ as follows.

$$C_3 = 2\beta_t\mathbb{E}\left[\langle\boldsymbol{\Phi}_{t-\tau} - \boldsymbol{\Phi}^*, \frac{1}{N}\sum_{i=1}^{N}\mathbf{h}(\boldsymbol{\theta}_{t+1}^i, \boldsymbol{\Phi}_{t-\tau}) - \frac{1}{N}\sum_{i=1}^{N}\bar{\mathbf{h}}(\boldsymbol{\theta}_{t+1}^i, \boldsymbol{\Phi}_{t-\tau})\rangle\right]$$

$$\leq \beta_t/\alpha_t\mathbb{E}[\|\boldsymbol{\Phi}_{t-\tau} - \boldsymbol{\Phi}^*\|^2] + \beta_t\alpha_t\frac{1}{N^2}\mathbb{E}\left[\left\|\sum_{i=1}^{N}\mathbf{h}(\boldsymbol{\theta}_{t+1}^i, \boldsymbol{\Phi}_{t-\tau}) - \sum_{i=1}^{N}\bar{\mathbf{h}}(\boldsymbol{\theta}_{t+1}^i, \boldsymbol{\Phi}_{t-\tau})\right\|^2\right]$$

$$\overset{\text{Definition 4.3}}{\leq} \beta_t/\alpha_t\mathbb{E}[\|\boldsymbol{\Phi}_{t-\tau} - \boldsymbol{\Phi}^*\|^2]$$

$$+ \beta_t\alpha_t\frac{1}{N^2}\mathbb{E}\left[\left(N\delta\|\boldsymbol{\Phi}_{t-\tau} - \boldsymbol{\Phi}^*\| + N\delta + \delta\sum_{i=1}^{N}\|\boldsymbol{\theta}_{t+1}^i - y^i(\boldsymbol{\Phi}^*)\|\right)^2\right]$$

$$\leq \beta_t/\alpha_t\mathbb{E}[\|\boldsymbol{\Phi}_{t-\tau} - \boldsymbol{\Phi}^*\|^2] + 3\beta_t\alpha_t\delta^2\mathbb{E}\left[\|\boldsymbol{\Phi}_{t-\tau} - \boldsymbol{\Phi}^*\|^2\right] + 3\beta_t\alpha_t\delta^2$$

$$+ \frac{3\beta_t\alpha_t\delta^2}{N}\mathbb{E}\left[\sum_{i=1}^{N}\|\boldsymbol{\theta}_{t+1}^i - y^i(\boldsymbol{\Phi}^*)\|^2\right]$$

$$= \beta_t/\alpha_t \mathbb{E}[\|\boldsymbol{\Phi}_{t-\tau} - \boldsymbol{\Phi}^*\|^2] + 3\beta_t\alpha_t\delta^2 \mathbb{E}\left[\|\boldsymbol{\Phi}_{t-\tau} - \boldsymbol{\Phi}^*\|^2\right] + 3\beta_t\alpha_t\delta^2$$

$$+ \frac{3\beta_t\alpha_t\delta^2}{N}\mathbb{E}\left[\sum_{i=1}^N \|\boldsymbol{\theta}_{t+1}^i - y^i(\boldsymbol{\Phi}_t) + y^i(\boldsymbol{\Phi}_t) - y^i(\boldsymbol{\Phi}^*)\|^2\right]$$

$$\leq \beta_t/\alpha_t \mathbb{E}[\|\boldsymbol{\Phi}_{t-\tau} - \boldsymbol{\Phi}^*\|^2] + 3\beta_t\alpha_t\delta^2 \mathbb{E}\left[\|\boldsymbol{\Phi}_{t-\tau} - \boldsymbol{\Phi}^*\|^2\right] + 3\beta_t\alpha_t\delta^2$$

$$+ \frac{6\beta_t\alpha_t\delta^2}{N}\mathbb{E}\left[\sum_{i=1}^N \|\boldsymbol{\theta}_{t+1}^i - y^i(\boldsymbol{\Phi}_t)\|^2\right] + 6\beta_t\alpha_t L^2\delta^2 \mathbb{E}\left[\|\boldsymbol{\Phi}_t - \boldsymbol{\Phi}^*\|^2\right]$$

$$\leq (2\beta_t/\alpha_t + 6\beta_t\alpha_t\delta^2)\mathbb{E}[\|\boldsymbol{\Phi}_{t-\tau} - \boldsymbol{\Phi}_t\|^2] + (2\beta_t/\alpha_t + 6\beta_t\alpha_t\delta^2 + 6\beta_t\alpha_t L^2\delta^2)\mathbb{E}[\|\boldsymbol{\Phi}_t - \boldsymbol{\Phi}^*\|^2]$$

$$+ 3\beta_t\alpha_t\delta^2 + \frac{6\beta_t\alpha_t\delta^2}{N}\mathbb{E}\left[\sum_{i=1}^N \|\boldsymbol{\theta}_{t+1}^i - y^i(\boldsymbol{\Phi}_t)\|^2\right],$$

where the last inequality comes from $\mathbb{E}[\|\boldsymbol{\Phi}_{t-\tau} - \boldsymbol{\Phi}^*\|^2] \leq 2\mathbb{E}[\|\boldsymbol{\Phi}_{t-\tau} - \boldsymbol{\Phi}_t\|^2] + 2\mathbb{E}[\|\boldsymbol{\Phi}_t - \boldsymbol{\Phi}^*\|^2]$. Hence, we can write Term 3 as follows

$$\text{Term 3} = C_1 + C_2 + C_3 + C_4$$

$$\leq \beta_t/\alpha_t \mathbb{E}[\|\boldsymbol{\Phi}_t - \boldsymbol{\Phi}_{t-\tau}\|^2] + 4\beta_t\alpha_t L^2(3 + 4L^2)\mathbb{E}[\|\boldsymbol{\Phi}^* - \boldsymbol{\Phi}_t\|^2]$$

$$+ \frac{16\beta_t\alpha_t L^2}{N}\mathbb{E}\left[\sum_{i=1}^N \|\boldsymbol{\theta}_{t+1}^i - y^i(\boldsymbol{\Phi}_t)\|^2\right] + 8\beta_t\alpha_t\delta^2$$

$$+ (2\beta_t/\alpha_t + \beta_t\alpha_t L^2)\mathbb{E}[\|\boldsymbol{\Phi}_{t-\tau} - \boldsymbol{\Phi}_t\|^2] + 2\beta_t/\alpha_t \mathbb{E}[\|\boldsymbol{\Phi}_t - \boldsymbol{\Phi}^*\|^2]$$

$$+ (2\beta_t/\alpha_t + 6\beta_t\alpha_t\delta^2)\mathbb{E}[\|\boldsymbol{\Phi}_{t-\tau} - \boldsymbol{\Phi}_t\|^2]$$

$$+ (2\beta_t/\alpha_t + 6\beta_t\alpha_t\delta^2 + 6\beta_t\alpha_t L^2\delta^2)\mathbb{E}[\|\boldsymbol{\Phi}_t - \boldsymbol{\Phi}^*\|^2]$$

$$+ 3\beta_t\alpha_t\delta^2 + \frac{6\beta_t\alpha_t\delta^2}{N}\mathbb{E}\left[\sum_{i=1}^N \|\boldsymbol{\theta}_{t+1}^i - y^i(\boldsymbol{\Phi}_t)\|^2\right]$$

$$+ (2\beta_t/\alpha_t + \beta_t\alpha_t L^2)\mathbb{E}[\|\boldsymbol{\Phi}_{t-\tau} - \boldsymbol{\Phi}_t\|^2] + 2\beta_t/\alpha_t \mathbb{E}[\|\boldsymbol{\Phi}_t - \boldsymbol{\Phi}^*\|^2]$$

$$\leq (7\beta_t/\alpha_t + 2\beta_t\alpha_t L^2 + 6\beta_t\alpha_t\delta^2)\mathbb{E}[\|\boldsymbol{\Phi}_{t-\tau} - \boldsymbol{\Phi}_t\|^2]$$

$$+ (6\beta_t/\alpha_t + 6\beta_t\alpha_t\delta^2(1 + L^2) + 4\beta_t\alpha_t L^2(3 + 4L^2))\mathbb{E}[\|\boldsymbol{\Phi}_t - \boldsymbol{\Phi}^*\|^2]$$

$$+ \frac{16\beta_t\alpha_t L^2 + 6\beta_t\alpha_t\delta^2}{N}\mathbb{E}\left[\sum_{i=1}^N \|y^i(\boldsymbol{\Phi}_t) - \boldsymbol{\theta}_{t+1}^i\|^2\right] + 11\beta_t\alpha_t\delta^2,$$

which completes the proof. $\qquad\qquad\square$

To bound Term 3, we need to bound $\mathbb{E}[\|\boldsymbol{\Phi}_t - \boldsymbol{\Phi}_{t-\tau}\|^2]$, which is shown in the following lemma.

**Lemma F.5.** *We have* $\forall t \geq 2\tau$

$$\mathbb{E}[\|\boldsymbol{\Phi}_t - \boldsymbol{\Phi}_{t-\tau}\|^2] \leq 4\tau^2\beta_0^2/\alpha_0^2 \mathbb{E}[\|\boldsymbol{\Phi}^* - \boldsymbol{\Phi}_t\|^2] + 8\beta_0^2 L^2 B^2\tau^2 + 8\beta_0^2\delta^2\tau^2. \tag{31}$$

*Proof.* The proof is similar to that of Lemma 3 in Dal Fabbro et al. (2023). Starting with

$$\|\boldsymbol{\Phi}^* - \boldsymbol{\Phi}_{t+1}\|^2 = \|\boldsymbol{\Phi}^* - \boldsymbol{\Phi}_t\|^2 + \frac{\beta_t^2}{N^2}\left\|\sum_{i=1}^N \mathbf{h}(\boldsymbol{\theta}_{t+1}^i, \boldsymbol{\Phi}_t)\right\|^2 - 2\beta_t\langle\boldsymbol{\Phi}^* - \boldsymbol{\Phi}_t, \frac{1}{N}\sum_{i=1}^N \mathbf{h}(\boldsymbol{\theta}_{t+1}^i, \boldsymbol{\Phi}_t)\rangle$$

$$\leq (1 + \beta_t/\alpha_0)\|\boldsymbol{\Phi}^* - \boldsymbol{\Phi}_t\|^2 + \frac{(\beta_t\alpha_0 + \beta_t^2)}{N^2}\left\|\sum_{i=1}^N \mathbf{h}_t^i(\boldsymbol{\theta}_{t+1}^i, \boldsymbol{\Phi}_t)\right\|^2$$

$$\leq (1 + \beta_t/\alpha_0)\|\boldsymbol{\Phi}^* - \boldsymbol{\Phi}_t\|^2 + \frac{2\beta_t\alpha_0}{N^2}\left\|\sum_{i=1}^N \mathbf{h}_t^i(\boldsymbol{\theta}_{t+1}^i, \boldsymbol{\Phi}_t)\right\|^2, \tag{32}$$

where the first inequality holds due to $2\mathbf{x}^T\mathbf{y} \leq \gamma\|\mathbf{x}\|^2 + 1/\gamma\|\mathbf{y}\|^2, \forall \gamma > 0$, and the second inequality holds since $\beta_t\alpha_0 \geq \beta_t^2$. We then have the following inequality according to Lemma F.2,

$$\mathbb{E}\left[\|\mathbf{\Phi}^* - \mathbf{\Phi}_{t+1}\|^2\right] \leq (1 + \beta_t/\alpha_0 + 8\beta_t\alpha_0 L^2(1 + L^2))\mathbb{E}\left[\|\mathbf{\Phi}^* - \mathbf{\Phi}_t\|^2\right]$$
$$+ \frac{8\beta_t\alpha_0 L^2}{N}\mathbb{E}\left[\sum_{i=1}^N \|\boldsymbol{\theta}_{t+1} - y^i(\mathbf{\Phi}_t)\|^2\right] + 8\beta_t\alpha_0\delta^2$$
$$\leq (1 + \beta_t/\alpha_0 + 8\beta_t\alpha_0 L^2(1 + L^2))\mathbb{E}\left[\|\mathbf{\Phi}^* - \mathbf{\Phi}_t\|^2\right] + 8\beta_t\alpha_0(L^2 B^2 + \delta^2). \tag{33}$$

By letting $\alpha_0 \leq \frac{1}{2L\sqrt{2(1+L^2)}}$, we have $\beta_t/\alpha_0 \geq 8\beta_t\alpha_0 L^2(1 + L^2)$, and hence

$$\mathbb{E}\left[\|\mathbf{\Phi}^* - \mathbf{\Phi}_{t+1}\|^2\right] \leq (1 + 2\beta_0/\alpha_0)\mathbb{E}\left[\|\mathbf{\Phi}^* - \mathbf{\Phi}_t\|^2\right] + 8\beta_0\alpha_0(L^2 B^2 + \delta^2). \tag{34}$$

Therefore, for all $t'$ such that $t - \tau \leq t' \leq t$,

$$\mathbb{E}[\|\mathbf{\Phi}^* - \mathbf{\Phi}_{t'}\|^2] \leq (1 + 2\beta_0/\alpha_0)^\tau \mathbb{E}[\|\mathbf{\Phi}^* - \mathbf{\Phi}_{t-\tau}\|^2] + 8\beta_0\alpha_0(L^2 B^2 + \delta^2)\sum_{\ell=0}^{\tau-1}(1 + 2\beta_0/\alpha_0)^\ell. \tag{35}$$

Using the fact that $(1 + x) \leq e^x$ (Dal Fabbro et al., 2023), if we let $\beta_0/\alpha_0 \leq \frac{1}{8\tau}$, we have

$$(1 + 2\beta_0/\alpha_0)^\ell \leq (1 + 2\beta_0/\alpha_0)^\tau \leq e^{0.25} \leq 2,$$

and

$$\sum_{\ell=0}^{\tau-1}(1 + 32\beta^2)^\ell \leq 2\tau.$$

Hence, we have

$$\mathbb{E}[\|\mathbf{\Phi}^* - \mathbf{\Phi}_{t'}\|^2] \leq 2\mathbb{E}[\|\mathbf{\Phi}^* - \mathbf{\Phi}_{t-\tau}\|^2] + 16\beta_0\alpha_0\tau(L^2 B^2 + \delta^2).$$

Since $\|\mathbf{\Phi}_t - \mathbf{\Phi}_{t-\tau}\|^2 \leq \tau \sum_{\ell=t-\tau}^{t-1}\|\mathbf{\Phi}_{\ell+1} - \mathbf{\Phi}_\ell\|^2 = \tau\frac{\beta^2}{N^2}\sum_{\ell=t-\tau}^{t-1}\|\sum_{i=1}^N \mathbf{h}_\ell^i(\boldsymbol{\theta}_{\ell+1}^i, \mathbf{\Phi}_\ell)\|^2$, when $t \geq 2\tau$, we have $\ell \geq \tau$ and thus

$$\mathbb{E}[\|\mathbf{\Phi}_t - \mathbf{\Phi}_{t-\tau}\|^2]$$
$$\leq \tau\frac{\beta^2}{N^2}\sum_{\ell=t-\tau}^{t-1}\|\sum_{i=1}^N \mathbf{h}_\ell^i(\boldsymbol{\theta}_{\ell+1}^i, \mathbf{\Phi}_\ell)\|^2$$
$$\leq \tau\sum_{\ell=t-\tau}^{t-1}((4\beta_0^2(L^2 + L^4)\mathbb{E}[\|\mathbf{\Phi}^* - \mathbf{\Phi}_\ell\|^2] + 4\beta_0^2 L^2 B^2\tau^2 + 4\beta_0^2\delta^2\tau^2$$
$$\leq 4\beta_0^2(L^2 + L^4)\tau^2(2\mathbb{E}[\|\mathbf{\Phi}^* - \mathbf{\Phi}_{t-\tau}\|^2] + 16\beta_0\alpha_0\tau(L^2 B^2 + \delta^2)) + 4\beta_0^2 L^2 B^2\tau^2 + 4\beta_0^2\delta^2\tau^2$$
$$= 8\beta_0^2(L^2 + L^4)\tau^2\mathbb{E}[\|\mathbf{\Phi}^* - \mathbf{\Phi}_{t-\tau}\|^2] + 4\beta_0^2 L^2 B^2\tau^2 + 4\beta_0^2\delta^2\tau^2$$
$$\leq \tau^2\beta_0^2/\alpha_0^2\mathbb{E}[\|\mathbf{\Phi}^* - \mathbf{\Phi}_{t-\tau}\|^2] + 4\beta_0^2 L^2 B^2\tau^2 + 4\beta_0^2\delta^2\tau^2$$
$$\leq 2\tau^2\beta_0^2/\alpha_0^2\mathbb{E}[\|\mathbf{\Phi}^* - \mathbf{\Phi}_t\|^2] + 2\tau^2\beta_0^2/\alpha_0^2\mathbb{E}[\|\mathbf{\Phi}_t - \mathbf{\Phi}_{t-\tau}\|^2] + 4\beta_0^2 L^2 B^2\tau^2 + 4\beta_0^2\delta^2\tau^2.$$

Since $2\tau^2\beta_0^2/\alpha_0^2 \leq 1/2$ when $\beta_0/\alpha_0 \leq \frac{1}{8\tau}$, we have

$$\mathbb{E}[\|\mathbf{\Phi}_t - \mathbf{\Phi}_{t-\tau}\|^2] \leq 4\tau^2\beta_0^2/\alpha_0^2\mathbb{E}[\|\mathbf{\Phi}^* - \mathbf{\Phi}_t\|^2] + 8\beta_0^2 L^2 B^2\tau^2 + 8\beta_0^2\delta^2\tau^2.$$

This completes the proof. $\qquad\square$

**Lemma F.6.** Term 3 *is bounded as follows*

$$Term_3 \leq (7\beta_t/\alpha_t + 2\beta_t\alpha_t L^2 + 6\beta_t\alpha_t\delta^2)(4\tau^2\beta_0^2/\alpha_0^2\mathbb{E}[\|\mathbf{\Phi}^* - \mathbf{\Phi}_t\|^2] + 8\beta_0^2 L^2 B^2\tau^2 + 8\beta_0^2\delta^2\tau^2)$$
$$+ (6\beta_t/\alpha_t + 6\beta_t\alpha_t\delta^2(1 + L^2) + 4\beta_t\alpha_t L^2(3 + 4L^2))\mathbb{E}[\|\mathbf{\Phi}_t - \mathbf{\Phi}^*\|^2]$$
$$+ \frac{16\beta_t\alpha_t L^2 + 6\beta_t\alpha_t\delta^2}{N}\mathbb{E}\left[\sum_{i=1}^N \|\boldsymbol{\theta}^{i,*} - \boldsymbol{\theta}_{t+1}\|^2\right] + 11\beta_t\alpha_t\delta^2.$$

*Proof.* Substituting the bound of $\mathbb{E}[\|\boldsymbol{\Phi}_t - \boldsymbol{\Phi}_{t-\tau}\|^2]$ in (31) into Term 3 in Lemma F.4 yield the final results. $\qquad\square$

Provided Term 1 in Lemma F.2, Term 2 in Lemma F.3, and Term 3 in Lemma F.6, we have the following lemma to characterize the drift between $\boldsymbol{\Phi}_{t+1}$ and $\boldsymbol{\Phi}_t$.

**Lemma F.7.** *For $t \geq 2\tau$, the following holds*

$$
\mathbb{E}[\|\boldsymbol{\Phi}^* - \boldsymbol{\Phi}_{t+1}\|^2]
$$
$$
\begin{aligned}
\leq &\; (1 + 4\beta_t^2(L^2 + L^4) + (7\beta_t/\alpha_t + 2\beta_t\alpha_t L^2 + 6\beta_t\alpha_t\delta^2)4\tau^2\beta_0^2/\alpha_0^2 \\
&\; + (6\beta_t/\alpha_t + 6\beta_t\alpha_t\delta^2(1 + L^2) + 4\beta_t\alpha_t L^2(3 + 4L^2)) + \beta_t(L/\alpha_t - 2\omega))\mathbb{E}[\|\boldsymbol{\Phi}_t - \boldsymbol{\Phi}^*\|^2] \\
&\; + \frac{4\beta_t^2 L^2 + \beta_t\alpha_t L + 16\beta_t\alpha_t L^2 + 6\beta_t\alpha_t\delta^2}{N}\mathbb{E}\left[\sum_{i=1}^N \|\boldsymbol{\theta}^{i,*} - \boldsymbol{\theta}_{t+1}\|^2\right] \\
&\; + (7\beta_t/\alpha_t + 2\beta_t\alpha_t L^2 + 6\beta_t\alpha_t\delta^2)(8\beta_0^2 L^2 B^2\tau^2 + 8\beta_0^2\delta^2\tau^2) + 4\beta_t^2\delta^2 + 11\beta_t\alpha_t\delta^2.
\end{aligned}
$$

*Proof.* Substituting $Term_1, Term_2$ and $Term_3$ back into Lemma F.1, we have

$$
\mathbb{E}[\|\boldsymbol{\Phi}^* - \boldsymbol{\Phi}_{t+1}\|^2]
$$
$$
\begin{aligned}
\leq &\; \mathbb{E}[\|\boldsymbol{\Phi}^* - \boldsymbol{\Phi}_t\|^2] + 4\beta_t^2(L^2 + L^4)\mathbb{E}[\|\boldsymbol{\Phi}^* - \boldsymbol{\Phi}_t\|^2] + \frac{4\beta_t^2 L^2}{N}\mathbb{E}\left[\sum_{i=1}^N \|\boldsymbol{\theta}_{t+1} - y^i(\boldsymbol{\Phi}_t)\|^2\right] + 4\beta_t^2\delta^2 \\
&\; + \beta_t(L/\alpha_t - 2\omega)\mathbb{E}[\|\boldsymbol{\Phi}^* - \boldsymbol{\Phi}_t\|^2] + \frac{\beta_t\alpha_t L}{N}\mathbb{E}\left[\sum_{i=1}^N \|\boldsymbol{\theta}_{t+1}^i - y^i(\boldsymbol{\Phi}_t)\|^2\right] \\
&\; + (7\beta_t/\alpha_t + 2\beta_t\alpha_t L^2 + 6\beta_t\alpha_t\delta^2)(4\tau^2\beta_0^2/\alpha_0^2\mathbb{E}[\|\boldsymbol{\Phi}^* - \boldsymbol{\Phi}_t\|^2] + 8\beta_0^2 L^2 B^2\tau^2 + 8\beta_0^2\delta^2\tau^2) \\
&\; + (6\beta_t/\alpha_t + 6\beta_t\alpha_t\delta^2(1 + L^2) + 4\beta_t\alpha_t L^2(3 + 4L^2))\mathbb{E}[\|\boldsymbol{\Phi}_t - \boldsymbol{\Phi}^*\|^2] \\
&\; + \frac{16\beta_t\alpha_t L^2 + 6\beta_t\alpha_t\delta^2}{N}\mathbb{E}\left[\sum_{i=1}^N \|\boldsymbol{\theta}^{i,*} - \boldsymbol{\theta}_{t+1}\|^2\right] + 11\beta_t\alpha_t\delta^2 \\
= &\; (1 + 4\beta_t^2(L^2 + L^4) + (7\beta_t/\alpha_t + 2\beta_t\alpha_t L^2 + 6\beta_t\alpha_t\delta^2)4\tau^2\beta_0^2/\alpha_0^2 \\
&\; + (6\beta_t/\alpha_t + 6\beta_t\alpha_t\delta^2(1 + L^2) + 4\beta_t\alpha_t L^2(3 + 4L^2)) + \beta_t(L/\alpha_t - 2\omega))\mathbb{E}[\|\boldsymbol{\Phi}_t - \boldsymbol{\Phi}^*\|^2] \\
&\; + \frac{4\beta_t^2 L^2 + \beta_t\alpha_t L + 16\beta_t\alpha_t L^2 + 6\beta_t\alpha_t\delta^2}{N}\mathbb{E}\left[\sum_{i=1}^N \|\boldsymbol{\theta}^{i,*} - \boldsymbol{\theta}_{t+1}\|^2\right] \\
&\; + (7\beta_t/\alpha_t + 2\beta_t\alpha_t L^2 + 6\beta_t\alpha_t\delta^2)(8\beta_0^2 L^2 B^2\tau^2 + 8\beta_0^2\delta^2\tau^2) + 4\beta_t^2\delta^2 + 11\beta_t\alpha_t\delta^2.
\end{aligned}
$$

This completes the proof.

$\qquad\square$

### F.1.2 DRIFT OF $\boldsymbol{\theta}_t^i, \forall i$.

Next, we characterize the drift between $\boldsymbol{\theta}_{t+1}^i$ and $\boldsymbol{\theta}_t^i$.

**Lemma F.8.** *The drift between $\boldsymbol{\theta}_{t+1}^i$ and $\boldsymbol{\theta}_t^i, \forall i$ is given by*

$$
\mathbb{E}[\|\boldsymbol{\theta}_{t+1}^i - y^i(\boldsymbol{\Phi}_t)\|^2] = \underbrace{\mathbb{E}\left[\left\|\boldsymbol{\theta}_t^i - y^i(\boldsymbol{\Phi}_{t-1}) + \alpha_t\sum_{k=1}^K \mathbf{g}(\boldsymbol{\theta}_{t,k-1}^i, \boldsymbol{\Phi}_t)\right\|^2\right]}_{Term_4} + \underbrace{\mathbb{E}\left[\left\|y^i(\boldsymbol{\Phi}_{t-1}) - y^i(\boldsymbol{\Phi}_t)\right\|^2\right]}_{Term_5}
$$
$$
+ \underbrace{2\mathbb{E}\left[\left\langle \boldsymbol{\theta}_t^i - y^i(\boldsymbol{\Phi}_{t-1}) + \alpha_t\sum_{k=1}^K \mathbf{g}(\boldsymbol{\theta}_{t,k-1}^i, \boldsymbol{\Phi}_t), y^i(\boldsymbol{\Phi}_{t-1}) - y^i(\boldsymbol{\Phi}_t)\right\rangle\right]}_{Term_6}.
$$

$$
\tag{36}
$$

*Proof.* According to the update of $\boldsymbol{\theta}_t^i$ in (7), we have

$$
\mathbb{E}[\|\boldsymbol{\theta}_{t+1}^i - y^i(\boldsymbol{\Phi}_t)\|^2] = \mathbb{E}\left[\left\|\boldsymbol{\theta}_t^i - y^i(\boldsymbol{\Phi}_{t-1}) + \alpha_t \sum_{k=1}^{K} \mathbf{g}(\boldsymbol{\theta}_{t,k-1}^i, \boldsymbol{\Phi}_t) + y^i(\boldsymbol{\Phi}_{t-1}) - y^i(\boldsymbol{\Phi}_t)\right\|^2\right]
$$

$$
= \underbrace{\mathbb{E}\left[\left\|\boldsymbol{\theta}_t^i - y^i(\boldsymbol{\Phi}_{t-1}) + \alpha_t \sum_{k=1}^{K} \mathbf{g}(\boldsymbol{\theta}_{t,k-1}^i, \boldsymbol{\Phi}_t)\right\|^2\right]}_{Term_4} + \underbrace{\mathbb{E}\left[\left\|y^i(\boldsymbol{\Phi}_{t-1}) - y^i(\boldsymbol{\Phi}_t)\right\|^2\right]}_{Term_5}
$$

$$
+ \underbrace{2\mathbb{E}\left[\left\langle \boldsymbol{\theta}_t^i - y^i(\boldsymbol{\Phi}_{t-1}) + \alpha_t \sum_{k=1}^{K} \mathbf{g}(\boldsymbol{\theta}_{t,k-1}^i, \boldsymbol{\Phi}_t), y^i(\boldsymbol{\Phi}_{t-1}) - y^i(\boldsymbol{\Phi}_t) \right\rangle\right]}_{Term_6},
$$

$$
\tag{37}
$$

where the second inequality holds due to $\|\mathbf{x} + \mathbf{y}\|^2 = \|\mathbf{x}\|^2 + \|\mathbf{y}\|^2 + 2\langle \mathbf{x}, \mathbf{y} \rangle$.

$\square$

We next analyze each term in (37). First, we bound Term 4 in the following lemma.

**Lemma F.9.** *With $t \geq \tau$, we have* Term 4 *bounded as*

$$
\text{Term } 4 \leq (1 + 2\beta_{t-1}/\alpha_t - 2\alpha_t K\omega)\mathbb{E}\left[\left\|\boldsymbol{\theta}_t^i - y^i(\boldsymbol{\Phi}_{t-1})\right\|^2\right]
$$

$$
+ (12\alpha_t^2\delta^2 K^2 + 6K^2\delta^2\alpha_t^3/\beta_{t-1})\mathbb{E}[\|\boldsymbol{\Phi}_{t-1} - \boldsymbol{\Phi}^*\|^2]
$$

$$
+ (12\alpha_t^2\delta^2 K^2 + 2L^2\alpha_t^3/\beta_{t-1} + 6K^2\delta^2\alpha_t^3/\beta_{t-1})\mathbb{E}[\|\boldsymbol{\Phi}_t - \boldsymbol{\Phi}_{t-1}\|^2]
$$

$$
+ 6\alpha_t^2\delta^2 K^2(1 + B^2) + 2\alpha_t^2 K^2 L^2 B^2 + 2L^2 K^2 B^2\alpha_t^3/\beta_{t-1} + \alpha_t^3/\beta_{t-1}(3K^2 B^2 + 3K^2\delta^2).
$$

$$
\tag{38}
$$

*Proof.* According to the definition of Term 4, we have

$$
\text{Term } 4 = \mathbb{E}\left[\left\|\boldsymbol{\theta}_t^i - y^i(\boldsymbol{\Phi}_{t-1}) + \alpha_t \sum_{k=1}^{K} \mathbf{g}(\boldsymbol{\theta}_{t,k-1}^i, \boldsymbol{\Phi}_t)\right\|^2\right]
$$

$$
= \mathbb{E}\left[\left\|\boldsymbol{\theta}_t^i - y^i(\boldsymbol{\Phi}_{t-1})\right\|^2\right] + \alpha_t^2\mathbb{E}\left[\left\|\sum_{k=1}^{K} \mathbf{g}(\boldsymbol{\theta}_{t,k-1}^i, \boldsymbol{\Phi}_t)\right\|^2\right]
$$

$$
+ 2\alpha_t\left\langle \boldsymbol{\theta}_t^i - y^i(\boldsymbol{\Phi}_{t-1}), \sum_{k=1}^{K} \mathbf{g}(\boldsymbol{\theta}_{t,k-1}^i, \boldsymbol{\Phi}_t) \right\rangle
$$

$$
= \mathbb{E}\left[\left\|\boldsymbol{\theta}_t^i - y^i(\boldsymbol{\Phi}_{t-1})\right\|^2\right] + 2\alpha_t\mathbb{E}\left[\left\langle \boldsymbol{\theta}_t^i - y^i(\boldsymbol{\Phi}_{t-1}), \sum_{k=1}^{K} \mathbf{g}(\boldsymbol{\theta}_{t,k-1}^i, \boldsymbol{\Phi}_t) \right\rangle\right]
$$

$$
+ \alpha_t^2\mathbb{E}\left[\left\|\sum_{k=1}^{K} \mathbf{g}(\boldsymbol{\theta}_{t,k-1}^i, \boldsymbol{\Phi}_t) - \sum_{k=1}^{K} \bar{\mathbf{g}}(\boldsymbol{\theta}_{t,k-1}^i, \boldsymbol{\Phi}_t) + \sum_{k=1}^{K} \bar{\mathbf{g}}(\boldsymbol{\theta}_{t,k-1}^i, \boldsymbol{\Phi}_t) - \sum_{k=1}^{K} \bar{\mathbf{g}}(y^i(\boldsymbol{\Phi}_t), \boldsymbol{\Phi}_t)\right\|^2\right]
$$

$$
\leq \mathbb{E}\left[\left\|\boldsymbol{\theta}_t^i - y^i(\boldsymbol{\Phi}_{t-1})\right\|^2\right] + \underbrace{2\alpha_t^2\mathbb{E}\left[\left\|\sum_{k=1}^{K} \mathbf{g}(\boldsymbol{\theta}_{t,k-1}^i, \boldsymbol{\Phi}_t) - \sum_{k=1}^{K} \bar{\mathbf{g}}(\boldsymbol{\theta}_{t,k-1}^i, \boldsymbol{\Phi}_t)\right\|^2\right]}_{\text{Mixing time property in Definition 4.3}}
$$

$$
+ 2\alpha_t^2\mathbb{E}\left[\left\|\sum_{k=1}^{K} \bar{\mathbf{g}}(\boldsymbol{\theta}_{t,k-1}^i, \boldsymbol{\Phi}_t) - \sum_{k=1}^{K} \bar{\mathbf{g}}(y^i(\boldsymbol{\Phi}_t), \boldsymbol{\Phi}_t)\right\|^2\right]
$$

$$+ 2\alpha_t \mathbb{E}\left[\left\langle \boldsymbol{\theta}_t^i - y^i(\boldsymbol{\Phi}_{t-1}), \sum_{k=1}^{K} \mathbf{g}(\boldsymbol{\theta}_{t,k-1}^i, \boldsymbol{\Phi}_t) \right\rangle\right]$$

$$\leq \mathbb{E}\left[\left\|\boldsymbol{\theta}_t^i - y^i(\boldsymbol{\Phi}_{t-1})\right\|^2\right] + 6\alpha_t^2 \delta^2 K^2 \mathbb{E}\left[\left\|\boldsymbol{\Phi}_t - \boldsymbol{\Phi}^*\right\|^2\right] + 6\alpha_t^2 \delta^2 K^2 (1 + B^2) + 2\alpha_t^2 K^2 L^2 B^2$$

$$+ \underbrace{2\alpha_t \mathbb{E}\left[\left\langle \boldsymbol{\theta}_t^i - y^i(\boldsymbol{\Phi}_{t-1}), \sum_{k=1}^{K} \bar{\mathbf{g}}(\boldsymbol{\theta}_{t,k-1}^i, \boldsymbol{\Phi}_t) \right\rangle\right]}_{\text{Term 4.1}}$$

$$+ \underbrace{2\alpha_t \mathbb{E}\left[\left\langle \boldsymbol{\theta}_t^i - y^i(\boldsymbol{\Phi}_{t-1}), \sum_{k=1}^{K} \mathbf{g}(\boldsymbol{\theta}_{t,k-1}^i, \boldsymbol{\Phi}_t) - \sum_{k=1}^{K} \bar{\mathbf{g}}(\boldsymbol{\theta}_{t,k-1}^i, \boldsymbol{\Phi}_t) \right\rangle\right]}_{\text{Term 4.2}}, \qquad (39)$$

where the first inequality holds due to the fact that $\|\mathbf{x} + \mathbf{y}\|^2 \leq 2\|\mathbf{x}\|^2 + 2\|\mathbf{y}\|^2$, and the second inequality is due to the mixing time property of function $\mathbf{g}$ as in Definition 4.3.

Next, we bound Term 4.1 as

$$\text{Term 4.1} = 2\alpha_t \mathbb{E}\left[\left\langle \boldsymbol{\theta}_t^i - y^i(\boldsymbol{\Phi}_{t-1}), \sum_{k=1}^{K} \bar{\mathbf{g}}(\boldsymbol{\theta}_{t,k-1}^i, \boldsymbol{\Phi}_t) \right\rangle\right]$$

$$= 2\alpha_t \mathbb{E}\left[\left\langle \boldsymbol{\theta}_t^i - y^i(\boldsymbol{\Phi}_{t-1}), \sum_{k=1}^{K} \bar{\mathbf{g}}(\boldsymbol{\theta}_t^i, \boldsymbol{\Phi}_{t-1}) \right\rangle\right]$$

$$+ 2\alpha_t \mathbb{E}\left[\left\langle \boldsymbol{\theta}_t^i - y^i(\boldsymbol{\Phi}_{t-1}), \sum_{k=1}^{K} \bar{\mathbf{g}}(\boldsymbol{\theta}_{t,k-1}^i, \boldsymbol{\Phi}_t) - \sum_{k=1}^{K} \bar{\mathbf{g}}(\boldsymbol{\theta}_t^i, \boldsymbol{\Phi}_{t-1}) \right\rangle\right]$$

$$\leq -2\alpha_t K \omega \mathbb{E}\left[\left\|\boldsymbol{\theta}_t^i - y^i(\boldsymbol{\Phi}_{t-1})\right\|^2\right]$$

$$+ 2\alpha_t \mathbb{E}\left[\left\langle \boldsymbol{\theta}_t^i - y^i(\boldsymbol{\Phi}_{t-1}), \sum_{k=1}^{K} \bar{\mathbf{g}}(\boldsymbol{\theta}_{t,k-1}^i, \boldsymbol{\Phi}_t) - \sum_{k=1}^{K} \bar{\mathbf{g}}(\boldsymbol{\theta}_t^i, \boldsymbol{\Phi}_{t-1}) \right\rangle\right]$$

$$\leq -2\alpha_t K \omega \mathbb{E}\left[\left\|\boldsymbol{\theta}_t^i - y^i(\boldsymbol{\Phi}_{t-1})\right\|^2\right] + \beta_{t-1}/\alpha_t \mathbb{E}\left[\left\|\boldsymbol{\theta}_t^i - y^i(\boldsymbol{\Phi}_{t-1})\right\|^2\right]$$

$$+ \alpha_t^3/\beta_{t-1} \mathbb{E}\left[\left\|\sum_{k=1}^{K} \bar{\mathbf{g}}(\boldsymbol{\theta}_{t,k-1}^i, \boldsymbol{\Phi}_t) - \sum_{k=1}^{K} \bar{\mathbf{g}}(\boldsymbol{\theta}_t^i, \boldsymbol{\Phi}_{t-1})\right\|^2\right]. \qquad (40)$$

In particular, we can bound $\mathbb{E}\left[\left\|\sum_{k=1}^{K} \bar{\mathbf{g}}(\boldsymbol{\theta}_{t,k-1}^i, \boldsymbol{\Phi}_t) - \sum_{k=1}^{K} \bar{\mathbf{g}}(\boldsymbol{\theta}_t^i, \boldsymbol{\Phi}_{t-1})\right\|^2\right]$ as

$$\mathbb{E}\left[\left\|\sum_{k=1}^{K} \bar{\mathbf{g}}(\boldsymbol{\theta}_{t,k-1}^i, \boldsymbol{\Phi}_t) - \sum_{k=1}^{K} \bar{\mathbf{g}}(\boldsymbol{\theta}_t^i, \boldsymbol{\Phi}_{t-1})\right\|^2\right]$$

$$\leq 2L^2 \mathbb{E}\left[\left\|\boldsymbol{\Phi}_t - \boldsymbol{\Phi}_{t-1}\right\|^2\right] + 2L^2 \mathbb{E}\left[\left\|\sum_{k=1}^{K} \boldsymbol{\theta}_{t,k-1} - \boldsymbol{\theta}_t\right\|^2\right]$$

$$\leq 2L^2 \mathbb{E}\left[\left\|\boldsymbol{\Phi}_t - \boldsymbol{\Phi}_{t-1}\right\|^2\right] + 2L^2 K \mathbb{E}\left[\sum_{k=1}^{K} \left\|\boldsymbol{\theta}_{t,k-1} - \boldsymbol{\theta}_t\right\|^2\right]$$

$$\leq 2L^2 \mathbb{E}\left[\left\|\boldsymbol{\Phi}_t - \boldsymbol{\Phi}_{t-1}\right\|^2\right] + 2L^2 K^2 B^2. \qquad (41)$$

Substituting (41) back into (40), we have Term 4.1 bounded as

$$\text{Term 4.1} \leq -2\alpha_t K \omega \mathbb{E}\left[\left\|\boldsymbol{\theta}_t^i - y^i(\boldsymbol{\Phi}_{t-1})\right\|^2\right] + \beta_{t-1}/\alpha_t \mathbb{E}\left[\left\|\boldsymbol{\theta}_t^i - y^i(\boldsymbol{\Phi}_{t-1})\right\|^2\right]$$

$$+ \alpha_t^3/\beta_{t-1}(2L^2 \mathbb{E}\left[\left\|\boldsymbol{\Phi}_t - \boldsymbol{\Phi}_{t-1}\right\|^2\right] + 2L^2 K^2 B^2). \qquad (42)$$

We next bound Term 4.2 as

$$
\begin{aligned}
\text{Term 4.2} &= 2\alpha_t \mathbb{E}\left[\left\langle \boldsymbol{\theta}_t^i - y^i(\boldsymbol{\Phi}_{t-1}), \sum_{k=1}^K \mathbf{g}(\boldsymbol{\theta}_{t,k-1}^i, \boldsymbol{\Phi}_t) - \sum_{k=1}^K \bar{\mathbf{g}}(\boldsymbol{\theta}_{t,k-1}^i, \boldsymbol{\Phi}_t) \right\rangle\right] \\
&\leq \beta_{t-1}/\alpha_t \mathbb{E}\left[\left\|\boldsymbol{\theta}_t^i - y^i(\boldsymbol{\Phi}_{t-1})\right\|^2\right] + \alpha_t^3/\beta_{t-1}(3K^2B^2 + 3K^2\delta^2 + 3K^2\delta^2\mathbb{E}[\|\boldsymbol{\Phi}_t - \boldsymbol{\Phi}^*\|^2]) \\
&\leq \beta_{t-1}/\alpha_t \mathbb{E}\left[\left\|\boldsymbol{\theta}_t^i - y^i(\boldsymbol{\Phi}_{t-1})\right\|^2\right] + \alpha_t^3/\beta_{t-1}(3K^2B^2 + 3K^2\delta^2) \\
&\quad + 6K^2\delta^2\alpha_t^3/\beta_{t-1}\mathbb{E}[\|\boldsymbol{\Phi}_{t-1} - \boldsymbol{\Phi}^*\|^2] + 6K^2\delta^2\alpha_t^3/\beta_{t-1}\mathbb{E}[\|\boldsymbol{\Phi}_t - \boldsymbol{\Phi}_{t-1}\|^2] \quad (43)
\end{aligned}
$$

Substituting Term 4.1 and Term 4.2 back into (39), we get the final result

$$
\begin{aligned}
\text{Term 4} &\leq \mathbb{E}\left[\left\|\boldsymbol{\theta}_t^i - y^i(\boldsymbol{\Phi}_{t-1})\right\|^2\right] + 6\alpha_t^2\delta^2K^2\mathbb{E}\left[\|\boldsymbol{\Phi}_t - \boldsymbol{\Phi}^*\|^2\right] + 6\alpha_t^2\delta^2K^2(1+B^2) + 2\alpha_t^2K^2L^2B^2 \\
&\quad + \text{Term 4.1} + \text{Term 4.2} \\
&= \mathbb{E}\left[\left\|\boldsymbol{\theta}_t^i - y^i(\boldsymbol{\Phi}_{t-1})\right\|^2\right] + 6\alpha_t^2\delta^2K^2\mathbb{E}\left[\|\boldsymbol{\Phi}_t - \boldsymbol{\Phi}^*\|^2\right] + 6\alpha_t^2\delta^2K^2(1+B^2) + 2\alpha_t^2K^2L^2B^2 \\
&\quad - 2\alpha_tK\omega\mathbb{E}\left[\|\boldsymbol{\theta}_t^i - y^i(\boldsymbol{\Phi}_{t-1})\|^2\right] + \beta_{t-1}/\alpha_t\mathbb{E}\left[\left\|\boldsymbol{\theta}_t^i - y^i(\boldsymbol{\Phi}_{t-1})\right\|^2\right] \\
&\quad + \alpha_t^3/\beta_{t-1}(2L^2\mathbb{E}\left[\|\boldsymbol{\Phi}_t - \boldsymbol{\Phi}_{t-1}\|^2\right] + 2L^2K^2B^2) \\
&\quad + \beta_{t-1}/\alpha_t\mathbb{E}\left[\left\|\boldsymbol{\theta}_t^i - y^i(\boldsymbol{\Phi}_{t-1})\right\|^2\right] + \alpha_t^3/\beta_{t-1}(3K^2B^2 + 3K^2\delta^2) \\
&\quad + 6K^2\delta^2\alpha_t^3/\beta_{t-1}\mathbb{E}[\|\boldsymbol{\Phi}_{t-1} - \boldsymbol{\Phi}^*\|^2] + 6K^2\delta^2\alpha_t^3/\beta_{t-1}\mathbb{E}[\|\boldsymbol{\Phi}_t - \boldsymbol{\Phi}_{t-1}\|^2] \\
&\leq (1 + 2\beta_{t-1}/\alpha_t - 2\alpha_tK\omega)\mathbb{E}\left[\left\|\boldsymbol{\theta}_t^i - y^i(\boldsymbol{\Phi}_{t-1})\right\|^2\right] \\
&\quad + (12\alpha_t^2\delta^2K^2 + 6K^2\delta^2\alpha_t^3/\beta_{t-1})\mathbb{E}[\|\boldsymbol{\Phi}_{t-1} - \boldsymbol{\Phi}^*\|^2] \\
&\quad + (12\alpha_t^2\delta^2K^2 + 2L^2\alpha_t^3/\beta_{t-1} + 6K^2\delta^2\alpha_t^3/\beta_{t-1})\mathbb{E}[\|\boldsymbol{\Phi}_t - \boldsymbol{\Phi}_{t-1}\|^2] \\
&\quad + 6\alpha_t^2\delta^2K^2(1+B^2) + 2\alpha_t^2K^2L^2B^2 + 2L^2K^2B^2\alpha_t^3/\beta_{t-1} \\
&\quad + \alpha_t^3/\beta_{t-1}(3K^2B^2 + 3K^2\delta^2) \quad (44)
\end{aligned}
$$

This completes the proof. $\square$

Next, we bound Term 5 in the following lemma.

**Lemma F.10.** *With $t \geq \tau$, we have* Term 5 *bounded as*

$$
\text{Term 5} \leq 4\beta_{t-1}^2(L^4 + L^6)\mathbb{E}[\|\boldsymbol{\Phi}^* - \boldsymbol{\Phi}_{t-1}\|^2] + \frac{4\beta_{t-1}^2L^4}{N}\mathbb{E}\left[\sum_{i=1}^N \|\boldsymbol{\theta}_t - y^i(\boldsymbol{\Phi}_{t-1})\|^2\right] + 4L^2\beta_{t-1}^2\delta^2.
\quad (45)
$$

*Proof.* We have

$$
\begin{aligned}
\text{Term 5} &= \mathbb{E}\left[\left\|y^i(\boldsymbol{\Phi}_{t-1}) - y^i(\boldsymbol{\Phi}_t)\right\|^2\right] = L^2\mathbb{E}\left[\|\boldsymbol{\Phi}_t - \boldsymbol{\Phi}_{t-1}\|^2\right] \\
&= \frac{L^2\beta_{t-1}^2}{N^2}\mathbb{E}\left[\left\|\sum_{i=1}^N \mathbf{h}(\boldsymbol{\theta}_t^i, \boldsymbol{\Phi}_{t-1})\right\|^2\right] \\
&\leq 4\beta_{t-1}^2(L^4 + L^6)\mathbb{E}[\|\boldsymbol{\Phi}^* - \boldsymbol{\Phi}_{t-1}\|^2] + \frac{4\beta_{t-1}^2L^4}{N}\mathbb{E}\left[\sum_{i=1}^N \|\boldsymbol{\theta}_t - y^i(\boldsymbol{\Phi}_{t-1})\|^2\right] + 4L^2\beta_{t-1}^2\delta^2,
\end{aligned}
$$
$$
(46)
$$

where the last inequality holds due to Lemma F.2. $\square$

Next, we bound Term 6 in the following lemma.

**Lemma F.11.** *We have* Term 6 *bounded as*

$$\text{Term } 6 \le \beta_{t-1}/\alpha_t \text{Term } 4 + \alpha_t/\beta_{t-1} \text{Term } 5. \tag{47}$$

*Proof.*

$$\text{Term } 6 = 2\mathbb{E}\left[\left\langle \boldsymbol{\theta}_t^i - y^i(\boldsymbol{\Phi}_{t-1}) + \alpha_t \sum_{k=1}^{K} \mathbf{g}(\boldsymbol{\theta}_{t,k-1}^i, \boldsymbol{\Phi}_t), y^i(\boldsymbol{\Phi}_{t-1}) - y^i(\boldsymbol{\Phi}_t) \right\rangle\right]$$

$$\le \beta_{t-1}/\alpha_t \mathbb{E}\left[\underbrace{\left\|\boldsymbol{\theta}_t^i - y^i(\boldsymbol{\Phi}_{t-1}) + \alpha_t \sum_{k=1}^{K} \mathbf{g}(\boldsymbol{\theta}_{t,k-1}^i)\right\|^2}_{\text{Term } 4}\right] + \alpha_t/\beta_{t-1}\underbrace{\mathbb{E}\left[\left\|y^i(\boldsymbol{\Phi}_{t-1}) - y^i(\boldsymbol{\Phi}_t)\right\|^2\right]}_{\text{Term } 5}$$

$$\tag{48}$$

$\square$

Providing Term 4 in Lemma F.9, Term 5 in Lemma F.10, and Term 6 in Lemma F.11, we have the following result.

**Lemma F.12.** *For $t \ge \tau$, the following holds*

$$\mathbb{E}[\|\boldsymbol{\theta}_{t+1}^i - y^i(\boldsymbol{\Phi}_t)\|^2]$$

$$\le \Bigg[(1 + \beta_{t-1}/\alpha_t)\bigg((1 + 2\beta_{t-1}/\alpha_t - 2\alpha_t K\omega)$$

$$+ (12\alpha_t^2\delta^2 K^2 + 2L^2\alpha_t^3/\beta_{t-1} + 6K^2\delta^2\alpha_t^3/\beta_{t-1})\frac{4\beta_{t-1}^2 L^2}{N}\bigg) + (1 + \alpha_t/\beta_{t-1})\frac{4\beta_{t-1}^2 L^4}{N}\Bigg]$$

$$\cdot \mathbb{E}\left[\left\|\boldsymbol{\theta}_t^i - y^i(\boldsymbol{\Phi}_{t-1})\right\|^2\right]$$

$$+ \Bigg[(1 + \beta_{t-1}/\alpha_t)\bigg((12\alpha_t^2\delta^2 K^2 + 6K^2\delta^2\alpha_t^3/\beta_{t-1})$$

$$+ (12\alpha_t^2\delta^2 K^2 + 2L^2\alpha_t^3/\beta_{t-1} + 6K^2\delta^2\alpha_t^3/\beta_{t-1})(4\beta_{t-1}(L^2 + L^4))\bigg)$$

$$+ (1 + \alpha_t/\beta_{t-1})4\beta_{t-1}^2(L^4 + L^6)\Bigg] \cdot \mathbb{E}[\|\boldsymbol{\Phi}^* - \boldsymbol{\Phi}_{t-1}\|^2]$$

$$+ (1 + \beta_{t-1}/\alpha_t)\bigg((12\alpha_t^2\delta^2 K^2 + 2L^2\alpha_t^3/\beta_{t-1} + 6K^2\delta^2\alpha_t^3/\beta_{t-1})4\beta_{t-1}^2\delta^2$$

$$+ 6\alpha_t^2\delta^2 K^2(1 + B^2) + 2\alpha_t^2 K^2 L^2 B^2 + 2L^2 K^2 B^2\alpha_t^3/\beta_{t-1} + \alpha_t^3/\beta_{t-1}(3K^2 B^2 + 3K^2\delta^2)\bigg)$$

$$+ (1 - \alpha_t/\beta_{t+1}) \cdot 4L^2\beta_{t-1}^2\delta^2. \tag{49}$$

*Proof.* According to (36), we have

$$\mathbb{E}[\|\boldsymbol{\theta}_{t+1}^i - y^i(\boldsymbol{\Phi}_t)\|^2] = Term_4 + Term_5 + Term_6$$

$$\overset{\text{Lemma F.11}}{\le} (1 + \beta_{t-1}/\alpha_t)Term_4 + (1 + \alpha_t/\beta_{t-1})Term_5$$

$$\le (1 + \beta_{t-1}/\alpha_t)\bigg((1 + 2\beta_{t-1}/\alpha_t - 2\alpha_t K\omega)\mathbb{E}\left[\left\|\boldsymbol{\theta}_t^i - y^i(\boldsymbol{\Phi}_{t-1})\right\|^2\right]$$

$$+ (12\alpha_t^2\delta^2 K^2 + 6K^2\delta^2\alpha_t^3/\beta_{t-1})\mathbb{E}[\|\boldsymbol{\Phi}_{t-1} - \boldsymbol{\Phi}^*\|^2]$$

$$+ (12\alpha_t^2\delta^2 K^2 + 2L^2\alpha_t^3/\beta_{t-1} + 6K^2\delta^2\alpha_t^3/\beta_{t-1})\mathbb{E}[\|\boldsymbol{\Phi}_t - \boldsymbol{\Phi}_{t-1}\|^2]$$

$$+ 6\alpha_t^2\delta^2 K^2(1 + B^2) + 2\alpha_t^2 K^2 L^2 B^2 + 2L^2 K^2 B^2\alpha_t^3/\beta_{t-1} + \alpha_t^3/\beta_{t-1}(3K^2 B^2 + 3K^2\delta^2)\bigg)$$

$$+ (1 + \alpha_t/\beta_{t-1})\bigg( 4\beta_{t-1}^2(L^4 + L^6)\mathbb{E}[\|\mathbf{\Phi}^* - \mathbf{\Phi}_{t-1}\|^2]$$

$$+ \frac{4\beta_{t-1}^2 L^4}{N}\mathbb{E}\left[\sum_{i=1}^N \|\boldsymbol{\theta}_t - y^i(\mathbf{\Phi}_{t-1})\|^2\right] + 4L^2\beta_{t-1}^2\delta^2\bigg)$$

$$\overset{\text{Lemma F.10}}{\leq} (1 + \beta_{t-1}/\alpha_t)\bigg( (1 + 2\beta_{t-1}/\alpha_t - 2\alpha_t K\omega)\mathbb{E}\left[\|\boldsymbol{\theta}_t^i - y^i(\mathbf{\Phi}_{t-1})\|^2\right]$$

$$+ (12\alpha_t^2\delta^2 K^2 + 6K^2\delta^2\alpha_t^3/\beta_{t-1})\mathbb{E}[\|\mathbf{\Phi}_{t-1} - \mathbf{\Phi}^*\|^2]$$

$$+ (12\alpha_t^2\delta^2 K^2 + 2L^2\alpha_t^3/\beta_{t-1} + 6K^2\delta^2\alpha_t^3/\beta_{t-1})$$

$$\cdot \bigg( 4\beta_{t-1}^2(L^2 + L^4)\mathbb{E}[\|\mathbf{\Phi}^* - \mathbf{\Phi}_{t-1}\|^2] + \frac{4\beta_{t-1}^2 L^2}{N}\mathbb{E}\left[\sum_{i=1}^N \|\boldsymbol{\theta}_t - y^i(\mathbf{\Phi}_{t-1})\|^2\right] + 4\beta_{t-1}^2\delta^2\bigg)$$

$$+ 6\alpha_t^2\delta^2 K^2(1 + B^2) + 2\alpha_t^2 K^2 L^2 B^2 + 2L^2 K^2 B^2\alpha_t^3/\beta_{t-1} + \alpha_t^3/\beta_{t-1}(3K^2 B^2 + 3K^2\delta^2)\bigg)$$

$$+ (1 + \alpha_t/\beta_{t-1})\bigg( 4\beta_{t-1}^2(L^4 + L^6)\mathbb{E}[\|\mathbf{\Phi}^* - \mathbf{\Phi}_{t-1}\|^2]$$

$$+ \frac{4\beta_{t-1}^2 L^4}{N}\mathbb{E}\left[\sum_{i=1}^N \|\boldsymbol{\theta}_t - y^i(\mathbf{\Phi}_{t-1})\|^2\right] + 4L^2\beta_{t-1}^2\delta^2\bigg)$$

$$= \bigg[(1 + \beta_{t-1}/\alpha_t)\bigg( (1 + 2\beta_{t-1}/\alpha_t - 2\alpha_t K\omega)$$

$$+ (12\alpha_t^2\delta^2 K^2 + 2L^2\alpha_t^3/\beta_{t-1} + 6K^2\delta^2\alpha_t^3/\beta_{t-1})\frac{4\beta_{t-1}^2 L^2}{N}\bigg) + (1 + \alpha_t/\beta_{t-1})\frac{4\beta_{t-1}^2 L^4}{N}\bigg]$$

$$\cdot \mathbb{E}\left[\|\boldsymbol{\theta}_t^i - y^i(\mathbf{\Phi}_{t-1})\|^2\right]$$

$$+ \bigg[(1 + \beta_{t-1}/\alpha_t)\bigg( (12\alpha_t^2\delta^2 K^2 + 6K^2\delta^2\alpha_t^3/\beta_{t-1})$$

$$+ (12\alpha_t^2\delta^2 K^2 + 2L^2\alpha_t^3/\beta_{t-1} + 6K^2\delta^2\alpha_t^3/\beta_{t-1})(4\beta_{t-1}(L^2 + L^4))\bigg)$$

$$+ (1 + \alpha_t/\beta_{t-1})4\beta_{t-1}^2(L^4 + L^6)\bigg] \cdot \mathbb{E}[\|\mathbf{\Phi}^* - \mathbf{\Phi}_{t-1}\|^2]$$

$$+ (1 + \beta_{t-1}/\alpha_t)\bigg( (12\alpha_t^2\delta^2 K^2 + 2L^2\alpha_t^3/\beta_{t-1} + 6K^2\delta^2\alpha_t^3/\beta_{t-1})4\beta_{t-1}^2\delta^2$$

$$+ 6\alpha_t^2\delta^2 K^2(1 + B^2) + 2\alpha_t^2 K^2 L^2 B^2 + 2L^2 K^2 B^2\alpha_t^3/\beta_{t-1} + \alpha_t^3/\beta_{t-1}(3K^2 B^2 + 3K^2\delta^2)\bigg)$$

$$+ (1 + \alpha_t/\beta_{t-1}) \cdot 4L^2\beta_{t-1}^2\delta^2. \tag{50}$$

This completes the proof. $\qquad\square$

### F.1.3 Final Step of Proof for Theorem 4.13

**Now, we are ready to proof the desired result in Theorem 4.13.**

According to the definition of Lyapunov function in (14), We have

$$M(\{\boldsymbol{\theta}_{t+2}^i\}, \mathbf{\Phi}_{t+1}) = \|\mathbf{\Phi}_{t+1} - \mathbf{\Phi}^*\|^2 + \frac{\beta_t}{\alpha_{t+1}} \cdot \frac{1}{N}\sum_{i=1}^N \|\boldsymbol{\theta}_{t+2}^i - y^i(\mathbf{\Phi}_{t+1})\|^2$$

$$\leq (1 + 4\beta_t^2(L^2 + L^4) + (7\beta_t/\alpha_t + 2\beta_t\alpha_t L^2 + 6\beta_t\alpha_t\delta^2)4\tau^2\beta_0^2/\alpha_0^2$$

$$
\begin{aligned}
&+ (6\beta_t/\alpha_t + 6\beta_t\alpha_t\delta^2(1+L^2) + 4\beta_t\alpha_t L^2(3+4L^2)) + \beta_t(L/\alpha_t - 2\omega))\mathbb{E}[\|\mathbf{\Phi}_t - \mathbf{\Phi}^*\|^2] \\
&+ \frac{4\beta_t^2 L^2 + \beta_t\alpha_t L + 16\beta_t\alpha_t L^2 + 6\beta_t\alpha_t\delta^2}{N}\mathbb{E}\left[\sum_{i=1}^{N}\|\boldsymbol{\theta}_{t+1}^i - y^i(\mathbf{\Phi}_t)\|^2\right] \\
&+ (7\beta_t/\alpha_t + 2\beta_t\alpha_t L^2 + 6\beta_t\alpha_t\delta^2)(8\beta_0^2 L^2 B^2\tau^2 + 8\beta_0^2\delta^2\tau^2) + 4\beta_t^2\delta^2 + 11\beta_t\alpha_t\delta^2
\end{aligned}
$$

$$
\begin{aligned}
&+ \frac{\beta_t}{\alpha_{t+1}} \cdot \Bigg[(1+\beta_t/\alpha_{t+1})\Bigg((1+2\beta_t/\alpha_{t+1} - 2\alpha_{t+1}K\omega) \\
&+ (12\alpha_{t+1}^2\delta^2 K^2 + 2L^2\alpha_{t+1}^3/\beta_t + 6K^2\delta^2\alpha_{t+1}^3/\beta_t)\frac{4\beta_t^2 L^2}{N}\Bigg) + (1+\alpha_{t+1}/\beta_t)\frac{4\beta_t^2 L^4}{N}\Bigg] \\
&\qquad\qquad \cdot \frac{1}{N}\mathbb{E}\left[\sum_{i=1}^{N}\|\boldsymbol{\theta}_{t+1}^i - y^i(\mathbf{\Phi}_t)\|^2\right] \\
&+ \Bigg[(1+\beta_t/\alpha_{t+1})\Bigg((12\alpha_{t+1}^2\delta^2 K^2 + 6K^2\delta^2\alpha_{t+1}^3/\beta_t) \\
&+ (12\alpha_{t+1}^2\delta^2 K^2 + 2L^2\alpha_{t+1}^3/\beta_t + 6K^2\delta^2\alpha_{t+1}^3/\beta_t)(4\beta_t(L^2+L^4))\Bigg) \\
&+ (1+\alpha_{t+1}/\beta_t)4\beta_t^2(L^4+L^6)\Bigg] \cdot \mathbb{E}[\|\mathbf{\Phi}^* - \mathbf{\Phi}_t\|^2] \\
&+ (1+\beta_t/\alpha_{t+1})\Bigg((12\alpha_{t+1}^2\delta^2 K^2 + 2L^2\alpha_{t+1}^3/\beta_t + 6K^2\delta^2\alpha_{t+1}^3/\beta_t)4\beta_t^2\delta^2 \\
&+ 6\alpha_{t+1}^2\delta^2 K^2(1+B^2) + 2\alpha_{t+1}^2 K^2 L^2 B^2 + 2L^2 K^2 B^2\alpha_{t+1}^3/\beta_t + \alpha_{t+1}^3/\beta_t(3K^2 B^2 + 3K^2\delta^2)\Bigg) \\
&+ (1+\alpha_{t+1}/\beta_t)\cdot 4L^2\beta_t^2\delta^2\Bigg]. \tag{51}
\end{aligned}
$$

To simplify the notations, we define

$$
\begin{aligned}
D_1 :=\ & (4\beta_t^2(L^2+L^4) + (7\beta_t/\alpha_t + 2\beta_t\alpha_t L^2 + 6\beta_t\alpha_t\delta^2)4\tau^2\beta_0^2/\alpha_0^2 \\
&+ (6\beta_t/\alpha_t + 6\beta_t\alpha_t\delta^2(1+L^2) + 4\beta_t\alpha_t L^2(3+4L^2)) + \beta_t L/\alpha_t) \\
&+ \frac{\beta_t}{\alpha_{t+1}}\Bigg[(1+\beta_t/\alpha_{t+1})\Bigg((12\alpha_{t+1}^2\delta^2 K^2 + 6K^2\delta^2\alpha_{t+1}^3/\beta_t) \\
&+ (12\alpha_{t+1}^2\delta^2 K^2 + 2L^2\alpha_{t+1}^3/\beta_t + 6K^2\delta^2\alpha_{t+1}^3/\beta_t)(4\beta_t(L^2+L^4))\Bigg) \\
&+ (1+\alpha_{t+1}/\beta_t)4\beta_t^2(L^4+L^6)\Bigg], \tag{52}
\end{aligned}
$$

and

$$
\begin{aligned}
D_2 :=\ & 4\beta_t^3/\alpha_{t+1}L^2 + \alpha_t^2 L + 16\alpha_t^2 L^2 + 6\alpha_t\alpha_t\delta^2 \\
&+ \Bigg[\Bigg((2\beta_t/\alpha_{t+1}) + (12\alpha_{t+1}^2\delta^2 K^2 + 2L^2\alpha_{t+1}^3/\beta_t + 6K^2\delta^2\alpha_{t+1}^3/\beta_t)\frac{4\beta_t^2 L^2}{N}\Bigg) + (1+\alpha_{t+1}/\beta_t)\frac{4\beta_t^2 L^4}{N}\Bigg] \\
&+ \Bigg[\beta_t/\alpha_{t+1}\Bigg((1+2\beta_t/\alpha_{t+1} - 2\alpha_{t+1}K\omega) \\
&+ (12\alpha_{t+1}^2\delta^2 K^2 + 2L^2\alpha_{t+1}^3/\beta_t + 6K^2\delta^2\alpha_{t+1}^3/\beta_t)\frac{4\beta_t^2 L^2}{N}\Bigg) + (1+\alpha_{t+1}/\beta_t)\frac{4\beta_t^2 L^4}{N}\Bigg]. \tag{53}
\end{aligned}
$$

Since $D_1$ is of higher orders of $o(\beta_t)$ and $D_2$ is of higher order of $o(\alpha_{t+1})$, we can let $D_1 \leq \omega\beta_t$ and $D_2 \leq K\omega\alpha_{t+1}$. Therefore, we have

$$M(\{\boldsymbol{\theta}_{t+2}^i\}, \boldsymbol{\Phi}_{t+1}) \leq (1 - \omega\beta_t)M(\{\boldsymbol{\theta}_{t+1}^i\}, \boldsymbol{\Phi}_t)$$

$$+ (144\tau^2 K^2 L^2 \delta^2 + 4L^4/N)\beta_t\alpha_{t+1} \left[\mathbb{E}[\|\boldsymbol{\Phi}_t - \boldsymbol{\Phi}^*\|^2] + \frac{1}{N}\mathbb{E}\left[\sum_{i=1}^N \left\|\boldsymbol{\theta}_{t+1}^i - y^i(\boldsymbol{\Phi}_t)\right\|^2\right]\right]$$

$$+ 4\alpha_{t+1}\beta_t K^2(3\delta^2(1 + B^2) + L^2 B^2) + 2\alpha_{t+1}^2(3K^2 B^2 + 3K^2\delta^2 + 2L^2 K^2 B^2) + 8\alpha_{t+1}\beta_t\delta^2$$

$$\leq (1 - \omega\beta_t)M(\{\boldsymbol{\theta}_{t+1}^i\}, \boldsymbol{\Phi}_t)$$

$$+ (144\tau^2 K^2 L^2 \delta^2 + 4L^2/N)\beta_t\alpha_t \left[\mathbb{E}[\|\boldsymbol{\Phi}_t - \boldsymbol{\Phi}^*\|^2] + \frac{1}{N}\mathbb{E}\left[\sum_{i=1}^N \left\|\boldsymbol{\theta}_{t+1}^i - y^i(\boldsymbol{\Phi}_t)\right\|^2\right]\right]$$

$$+ 4\alpha_t\beta_t K^2(3\delta^2(1 + B^2) + L^2 B^2) + 2\alpha_t^2(3K^2 B^2 + 3K^2\delta^2 + 2L^2 K^2 B^2) + 8\alpha_t\beta_t\delta^2, \quad (54)$$

where the first inequality holds by omitting the higher order of learning rates, and the second inequality holds due to the decreasing learning rates of $\alpha_t$.

We now set the proper decaying learning rates. Let $\alpha_t = \alpha_0/(t + 2)^{5/6}$ and $\beta_t = \beta_0/(t + 2)$. We then have

$$(t + 2)^2 \cdot (1 - \omega\beta_t) = (t + 2)^2(1 - \omega\beta_0)/(t + 2) \leq (t + 1)^2, \quad (55)$$

if $\omega\beta_o < 2$. In addition, we have the following inequalities

$$(t + 2)^2 \cdot \alpha_t\beta_t \leq \alpha_0\beta_0(t + 2)^{1/3},$$

$$(t + 2)^2 \cdot \alpha_t^2 = \alpha_0^2(t + 2)^2.$$

Hence, multiplying both sides with $(t + 2)^2$, we have

$$(t + 2)^2 M(\{\boldsymbol{\theta}_{t+2}^i\}, \boldsymbol{\Phi}_{t+1}) \leq (t + 1)^2 M(\{\boldsymbol{\theta}_{t+1}^i\}, \boldsymbol{\Phi}_t)$$

$$+ (144\tau^2 K^2 L^2 \delta^2 + 4L^2/N)\alpha_0\beta^0(t + 2)^{1/3} \left[\mathbb{E}[\|\boldsymbol{\Phi}_t - \boldsymbol{\Phi}^*\|^2] + \frac{1}{N}\mathbb{E}\left[\sum_{i=1}^N \|\boldsymbol{\theta}_{t+1}^i - y^i(\boldsymbol{\Phi}_t)\|^2\right]\right]$$

$$+ (4\alpha_0\beta_0 K^2(3\delta^2(1 + B^2) + L^2 B^2) + 2\alpha_0^2(3K^2 B^2 + 3K^2\delta^2 + 2L^2 K^2 B^2) + 8\alpha_0\beta_0\delta^2)(t + 2)^{1/3}.$$

Summing the above equation from $t = 0, \ldots, T$, we have

$$(T + 2)^2 M(\{\boldsymbol{\theta}_{t+2}^i\}, \boldsymbol{\Phi}_{t+1}) \leq M(\{\boldsymbol{\theta}_1^i\}, \boldsymbol{\Phi}_0)$$

$$+ (144\tau^2 K^2 L^2 \delta^2 + 4L^2/N)\alpha_0\beta^0(T + 2)^{4/3} \left[\mathbb{E}[\|\boldsymbol{\Phi}_0 - \boldsymbol{\Phi}^*\|^2] + \frac{1}{N}\mathbb{E}\left[\sum_{i=1}^N \|\boldsymbol{\theta}_1^i - y^i(\boldsymbol{\Phi}_0)\|^2\right]\right]$$

$$+ (4\alpha_0\beta_0 K^2(3\delta^2(1 + B^2) + L^2 B^2) + 2\alpha_0^2(3K^2 B^2 + 3K^2\delta^2 + 2L^2 K^2 B^2) + 8\alpha_0\beta_0\delta^2)(T + 2)^{4/3}.$$

Dividing both sides by $(T + 2)^2$, we have

$$M(\{\boldsymbol{\theta}_{t+2}^i\}, \boldsymbol{\Phi}_{t+1}) \leq \frac{M(\{\boldsymbol{\theta}_1^i\}, \boldsymbol{\Phi}_0)}{(T + 2)^2}$$

$$+ (144\tau^2 K^2 L^2 \delta^2 + 4L^2/N)\alpha_0\beta_0(T + 2)^{-2/3} \left[\mathbb{E}[\|\boldsymbol{\Phi}_0 - \boldsymbol{\Phi}^*\|^2] + \frac{1}{N}\mathbb{E}\left[\sum_{i=1}^N \|\boldsymbol{\theta}_1^i - y^i(\boldsymbol{\Phi}_0)\|^2\right]\right]$$

$$+ (4\alpha_0\beta_0 K^2(3\delta^2(1 + B^2) + L^2 B^2) + 2\alpha_0^2(3K^2 B^2 + 3K^2\delta^2 + 2L^2 K^2 B^2) + 8\alpha_0\beta_0\delta^2)(T + 2)^{-2/3}.$$

This completes the proof.

### F.2 PROOF OF COROLLARY 4.15

If $\alpha_0 = \beta_0 = o(N^{-1/3}K^{-1/2})$, we have

$$M(\{\boldsymbol{\theta}_{t+2}^i\}, \boldsymbol{\Phi}_{t+1}) \leq \mathcal{O}\left(\frac{1}{(T + 2)^2} + \frac{1}{N^{2/3}(T + 2)^{2/3}} + \frac{1}{K^2 N^{5/3}(T + 2)^{2/3}} + \frac{1}{K^2 N^{2/3}(T + 2)^{2/3}}\right),$$

which is dominated by $\mathcal{O}\left(\frac{1}{N^{2/3}(T+2)^{2/3}}\right)$ if $T^2 > N$.

Table 2: Parameter setting

| Parameter | Description |
|---|---|
| Input size | 4 |
| Hidden size | $128 \times 128 \times 128$ |
| Output size | 2 |
| Activation function | ReLu |
| Number of episodes | 500 |
| Batch size | 64 |
| Discount factor | 0.98 |
| $\epsilon$ greedy parameter | 0.01 |
| Target update | 30 |
| Buffer size | 10000 |
| Minimal size | 500 |
| Learning rate | 0.002, decays every 100 episodes |

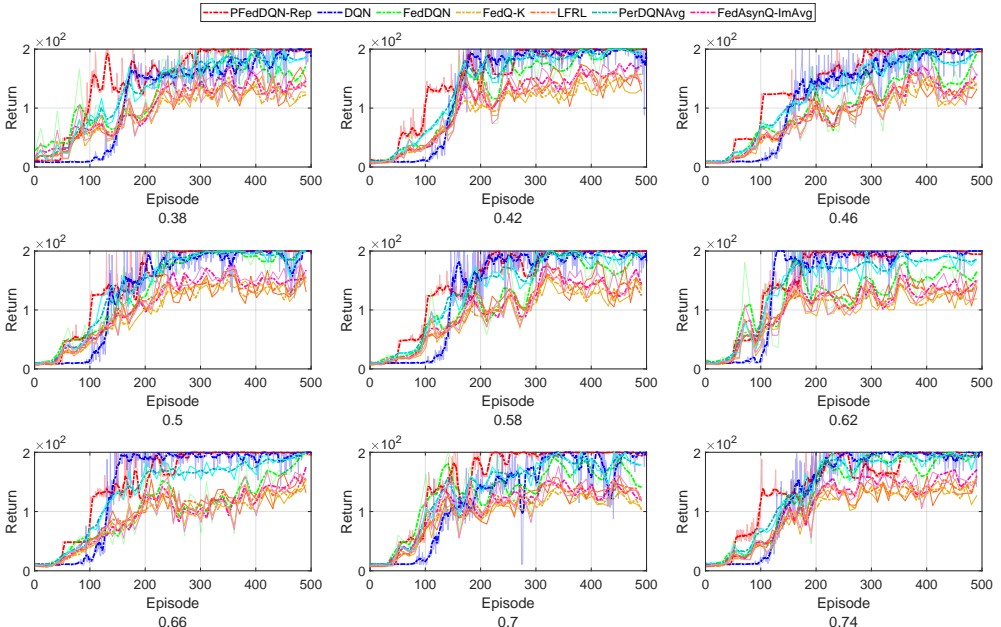

Figure 8: Comparison of control by DQN, FedDQN and PFEDDQN-REP in Cartpole Environments.

## G ADDITIONAL EXPERIMENT DETAILS

**Compute resources.** The experiments are performed on a computer with Intel 14900k CPU with 48GB of RAM. No GPU is involved.

**PFEDDQN-REP in the CartPole environment.** We evaluate the performance PFEDDQN-REP in a modified CartPole environment (Brockman et al., 2016). Similar to Jin et al. (2022), we change the length of pole to create different environments. Specifically, we consider 10 agents with varying pole length from 0.38 to 0.74 with a step size of 0.04. We compare PFEDDQN-REP with (i) a conventional DQN that each agent learns its own environment independently; and (ii) a federated version DQN (FedDQN) that allows all agents to collaboratively learn a single policy (without personalization). We randomly choose one agent and present its performance in Figure 3(top)(a). The results of the other agents are presented in Figure 8. Again, we observe that our PFEDDQN-REP achieves the maximized return much faster than the conventional DQN due to leveraging shared representations among agents; and obtains larger reward than FedDQN, thanks to our personalized policy. We further evaluate the effectiveness of shared representation learned by PFEDDQN-REP when generalizes it to a new agent. As shown in Figure 3(top)(b), our PFEDDQN-REP generalizes quickly to the new environment. Detailed parameter settings can be found in Table 2.

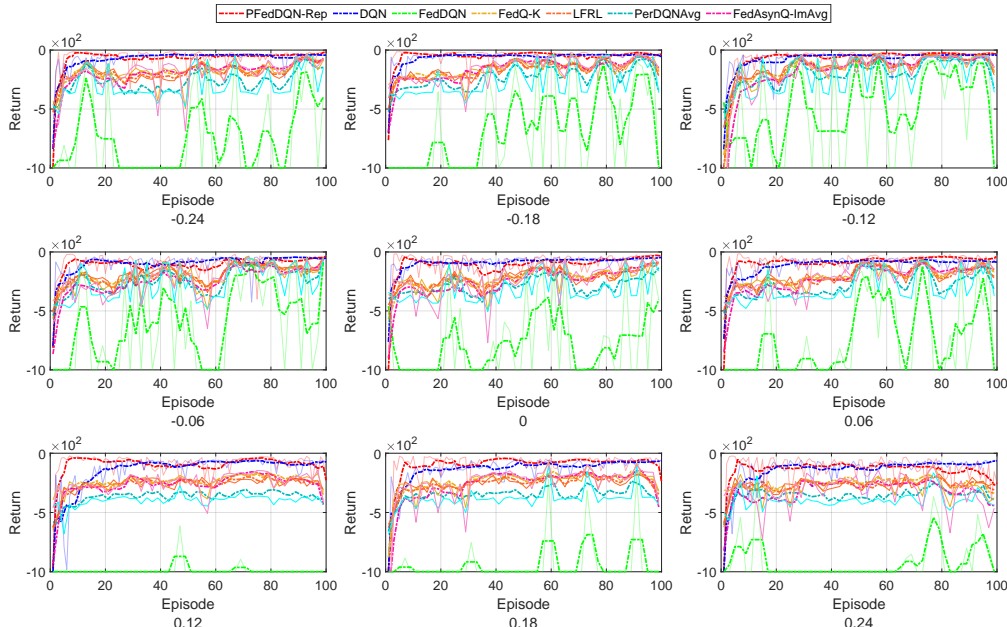

Figure 9: Comparison of control by DQN, FedDQN and PFEDDQN-REP in Acrobot Environments.

**PFEDDQN-REP in Acrobot environment.** We further evaluate FEDDQN-REP in a modified Acrobot environment (Brockman et al., 2016). The pole length is adjusted with [-0.3, 0.3] with a step size of 0.06, and the pole mass with be adjusted accordingly (Jin et al., 2022). The same two benchmarks are compared as in Figure 3(top). The parameter setting remains the same except number of episodes decreases to 100. Similar observations can be made from Figure 3(bottom) and Figure 9 as those for the Cartpole enviroments.

### G.1 MORE COMPLEX ENVIRONMENT: HOPPER

We consider another environment, Hopper from Gym, whose state and action space are both continuous. To induce heterogeneity within between the agents' environments, we vary the length of legs to be $0.02 + 0.001 \cdot i$, where $i$ is the $i$-th agent, while keeping other parameters (such as healthy reward, forward reward, and control cost (the $l2$ cost function to penalize large actions), the same. We increase the number of agents to 20, and plot the return with respect to the number of frames. In addition to training, we also generate a new sampled transition to validate the algorithms' ability to generalize.

In order to fit the algorithm to the continuous setting, we modified the proposed algorithm to a DDPG-based algorithm, similar to the DQN-related benchmarks. For FedQ-K, LFRL and FedAsynQ-ImAvg, we discretize the state and action spaces. Similar to Cartpole and Acrobot environments, our proposed PFedDDPG-Rep achieves the best reward and generalizes to new environments quickly, as shown in Figure 10.

### G.2 VERIFYING THE LINEAR SPEEDUP RESULT

We now verify the main theoretical result empirically. In the personalized setting, verifying this result is not as straightforward as in the non-personalized setting because Theorem 4.13 and Corollary 4.15 hold when parameters defined *across* environments (e.g., $\tau_\delta$, $C$, and others) remain constant as the number of agents (environments) increases.

To properly address this issue, we design an experiment that duplicates 2 initial environments with pole lengths 0.36 and 0.42. We duplicate these two environments with 2, 3, 4, and 5 times, thereby obtaining situations with $N = 2, 4, 6, 8, 10$. Because of this duplication, we know that the across-environment parameters (e.g., $\tau_\delta$, $C$, and others) remain constant.

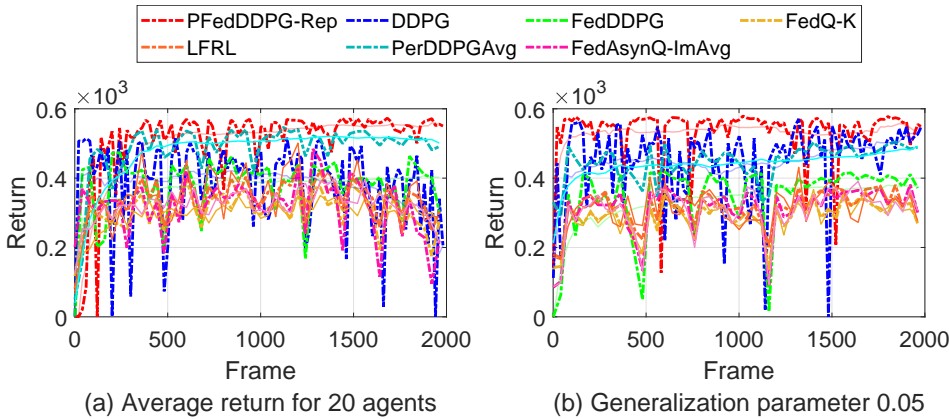

(a) Average return for 20 agents      (b) Generalization parameter 0.05

Figure 10: Hopper environment.

As shown in Figure 11, as we increase the number of agents, we see an approximate linear relationship in the convergence time. Of course, note that in practice, there is certain amount of unavoidable overhead in the experiments, which means the speedup will not be exactly as efficient as predicted by theory.

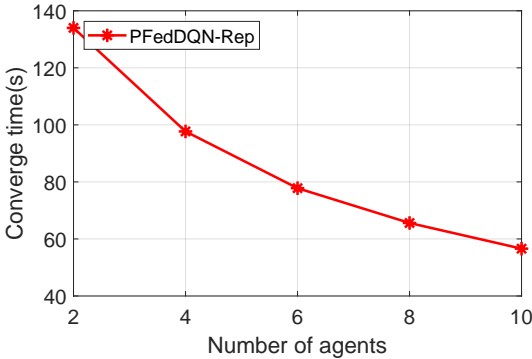

Figure 11: Linear speedup for cartpole with duplicates of environments.

### G.3 COMPUTATION AND WORST CASE PERSONALIZATION ERROR TRADE OFF

We now show another experiment that quantifies the tradeoff between computation and *personalization quality*, which we define as the worst case personalization error across agents:

$$\max_{i \in [N]} \mathbb{E}_{s \sim \mu^i, \pi^i} \left\| f^i(\boldsymbol{\theta}^i, \boldsymbol{\Phi}(s)) - V^{i,\pi^i}(s) \right\|^2. \tag{56}$$

The intuition behind this metric is that if all agents achieve good estimation error, then this metric is small (meaning we have personalized well), but if some agents perform poorly while others perform well, then this metric will be large (detecting that we did not personalize well).

We vary the number of agents from 2 to 10 and examine naive DQN that runs independently on each environment, FedDQN (no personalization), and PFedDQN-Rep (our approach). In the left panel of Figure 12, we show the computational resources needed to run each algorithm, while the right panel shows the personalization quality. We notice that for naive DQN, we can achieve no personalization error at the cost of high computation. At the other end of the spectrum, FedDQN leverages parallelization and reduces the computation, but has high personalization error. Finally, our algorithm, PFedDQN-Rep achieves the best of both worlds: low computation, while attaining low personalization error.

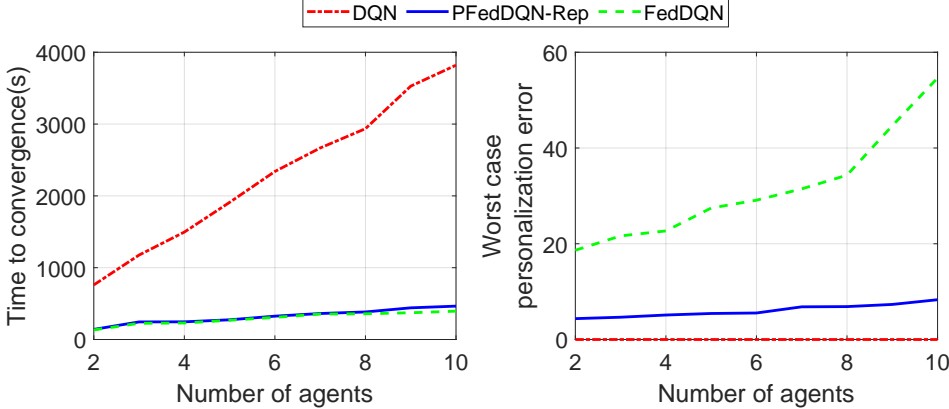

Figure 12: Computation versus worst case personalization error trade off.

### G.4 THE EFFECT OF ENVIRONMENT DISCREPANCY ON PERSONALIZATION ERROR

Recall that in the previous Hopper experiment, we vary the length of legs to be $0.02 + 0.001 \cdot i$, where $i$ is the $i$-th agent and $0.001 \cdot 10 = 0.01$ is the maximum *pole length discrepancy between environments*. In this section, we vary the maximum pole length discrepancy between 0 (all 10 environments are identical) to 0.04 (the environments have substantial differences).

We compare the performance of the three algorithms that include personalization, PerDQNAvg, FedAsymQ-ImAvg, and PFedDQN-Rep (ours). The results are in Figure 13.

We notice that as the discrepancy increases, all algorithm encounter degradations in personalization quality (measured in terms of the worst case personalization error defined in (56)), but our proposed algorithm achieves the least degradation.

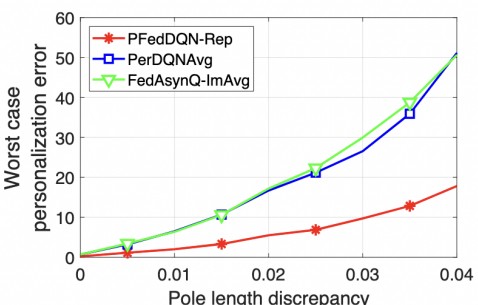

Figure 13: Worst case personalization error with varying pole length discrepancy across environments.

### G.5 ANOTHER LOOK AT PERSONALIZATION

Here, we give another look at how personalization is achieved by PFedRL-Rep. We first compute the cosine similarity matrix of transition probabilities between pairs of agents. This represents the similarity of environments between any two agents. After the algorithm converges, we compute the cosine similarity matrix for the policy layer (last layer) of the neural network. This represents the similarity of the learned policy between any two agents.

In Figures 14 and 15, we observe that all agents reach their unique personalization, and the cosine similarity of personalization layer shows close distribution as the similarity of transition probability. This essentially means for environments of similar transition probability matrices, their personalization layer will reach similar stage. Since the agents share the representation layer, the heatmap stays identical.

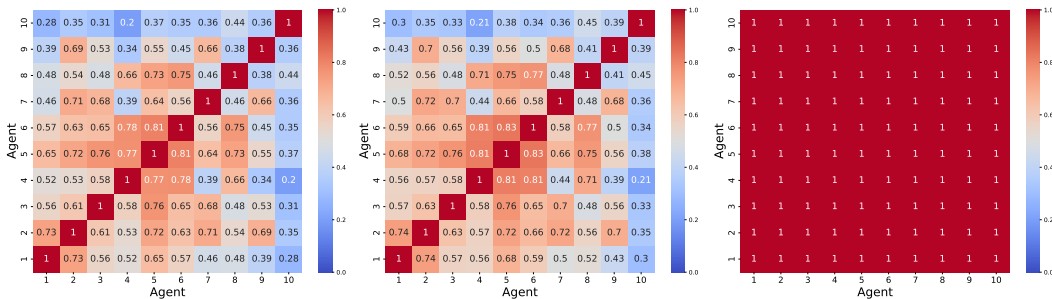

(a) Transition probability heatmap  (b) Personalization layer heatmap.  (c) Representation layer heatmap.

Figure 14: Heatmap of Cartpole environment.

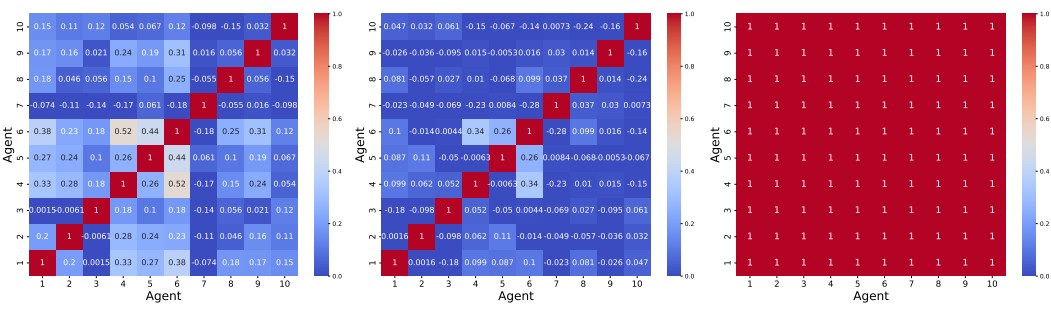

(a) Transition probability heatmap  (b) Personalization layer heatmap.  (c) Representation layer heatmap.

Figure 15: Heatmap of Acrobot environment.

## G.6 MORE STATISTICS OF THE EMPIRICAL RESULTS

We report the return average, variance average, return median and total running time for 10 environments for Cartpole and Acrobot environments. Among all algorithms, our PFedDQN-Rep achieves the best return average and median, with top variance and running time, as summarized in Tables 3 and 4. We also provide a zoom-in shortened plot for both environments to show the quick adaptation speed when sharing representations as in Figure 16.

Table 3: Statistics for Cartpole environment.

| Algorithm | Return average | Variance average | Return median | Total running time(s) |
|---|---|---|---|---|
| PFedDQN-Rep | 143 | 43 | 154 | 466 |
| DQN | 135 | 54 | 127 | 3840 |
| FedDQN | 101 | 67 | 88 | 387 |
| FedQ-K | 112 | 34 | 107 | 490 |
| LFRL | 117 | 47 | 99 | 434 |
| PerDQNAvg | 127 | 48 | 131 | 520 |
| FedAsynQ-ImAvg | 119 | 51 | 117 | 501 |

Table 4: Statistics for Acrobot environment.

| Algorithm | Return average | Variance average | Return median | Total running time(s) |
|---|---|---|---|---|
| PFedDQN-Rep | -42 | 37 | -29 | 618 |
| DQN | -63 | 67 | -57 | 5854 |
| FedDQN | -714 | 162 | -625 | 571 |
| FedQ-K | -213 | 41 | -202 | 621 |
| LFRL | -207 | 58 | -194 | 676 |
| PerDQNAvg | -295 | 64 | -277 | 602 |
| FedAsynQ-ImAvg | -191 | 36 | -186 | 664 |

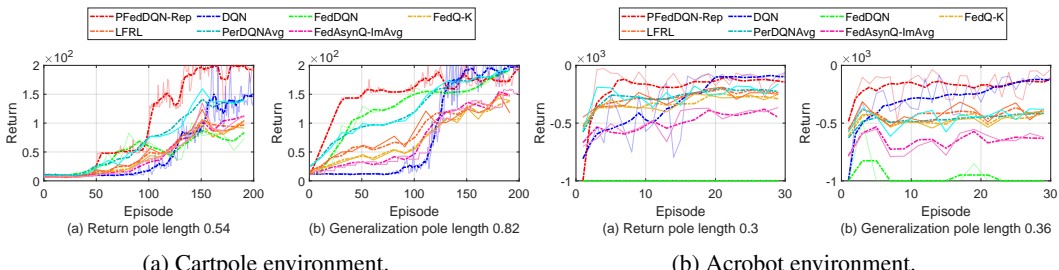

(a) Cartpole environment.          (b) Acrobot environment.

Figure 16: Shortened plot for cartpole and acrobot environment.

