# OpenReview forum: "On the Linear Speedup of Personalized Federated Reinforcement Learning with Shared Representations"
_ICLR.cc/2025/Conference — ICLR 2025 Poster_

### Official Review · Reviewer_mFXK · 2024-11-02

**Soundness:** 3
**Presentation:** 3
**Contribution:** 3
**Rating:** 6
**Confidence:** 4

**Summary:**

The manuscript introduces PFEDRL, a framework for personalized federated reinforcement learning (FedRL) aimed at addressing heterogeneity across agent environments. The authors propose PFEDTD-REP, a specific instantiation of PFEDRL with temporal difference (TD) learning. Notably, they claim a linear speedup in convergence proportional to the number of agents, a desirable characteristic in large-scale federated RL systems. Experimental results in both value-based learning (CliffWalking) and control tasks (CartPole, Acrobot) validate the framework's advantages, with promising outcomes in personalization and convergence speedup.

**Strengths:**

+The paper is practically relevant and addresses real-world heterogeneity in federated RL environments

+The manuscript made some innovations in FedRL theories:

- First work to prove linear speedup in personalized federated RL with shared representations under Markovian noise

- Rigorous analysis of convergence rates using two-timescale stochastic approximation theory

**Weaknesses:**

while there are merits in the paper’s theoretical contributions, I am concerned with a few critical points:

1. regarding the motivation on personalization:

a. the paper lacks formal definition of what constitutes successful personalization. The authors should consider to design metrics to quantify personalization quality

b. thus, no theoretical guarantees that learned personalization (via agent-specific parameters in the paper) captures meaningful environment-specific adaptations. What happens to personalization quality when environments are very different from each other?

c. how does agent count N affects personalization? if we add more agents by increasing N, do we have better personlization or worse?

d. is there a tradeoff between personalization and global performance or the speedup?

2. regarding the problem formulation and approach:

a. does sharing this common representation breach privacy preservation?

b. In section 2.2, why does the transition from (1) to (2) preserve the problem properties?

3. regarding theoretical rigor

a. corollary 4.15 claims that linear speedup w.r.t agent count N is justified by “we can proportionally decrease T as N increases while keeping the same convergence rate”. However, this is not a precise claim as linear speedup generally implies that adding more agents N directly enhances the convergence rate, without requiring adjustments to T. Here, the convergence rate remains fixed only by reducing T, which does not reflect true linear acceleration in convergence. The paper could be misleading readers into believing that the convergence rate inherently improves with more agents, rather than simply adjusting the number of communication rounds to balance computational costs

b. related to 3.a) above, if the authors claims linear speedup w.r.t agent count N, they should provide comprehensive experimental validation showing how convergence behavior scales with varying numbers of agents. Notable prior work making similar speedup claims, such as Fan et al. [1] which is one of the earliest FedRL works missing from the related work, included thorough ablation studies demonstrating the impact of agent count N on convergence. The absence of such analysis is particularly concerning given the centrality of the linear speedup claim to the paper's contributions.

c. otherwise, how to determine the optimal number of agents? If we add more agents, will we get faster convergence? what about personalization? intuitively, more agents should increase the personalization complexity.

4. regarding experimental evaluation.

a. Limited diversity in test environments (only classic control tasks) and no statistical significance is assessed

b. how to empirically verify the personalization achieved?

c. ablation on agent count N should be conducted.

---
[1] Fan, X., Ma, Y., Dai, Z., Jing, W., Tan, C., & Low, B. K. H. (2021). Fault-tolerant federated reinforcement learning with theoretical guarantee. *Advances in Neural Information Processing Systems*, *34*, 1007-1021.

**Questions:**

1. how does agent count N affects personalization? if we add more agents by increasing N, do we have better personlization or worse?
2. is there a tradeoff between personalization and global performance or the speedup?
3. how to determine the optimal number of agents? If we add more agents, will we get faster convergence? what about personalization? intuitively, more agents should increase the personalization complexity.

---

> ### Author Response · Authors · 2024-11-21
> **Official Response by Authors (1/4)**
>
> Thank you very much for your review and constructive comments. Here we would like to address the reviewer's concerns and hope that can help raise the rating of our paper.
>
> **Weakness \#1a:  Regarding the motivation on personalization:
> a. the paper lacks formal definition of what constitutes successful personalization. The authors should consider to design metrics to quantify personalization quality.**
>
> **Response:** Thank you for your comments. However, we believe there is a misunderstanding here; let us respectfully clarify. Our definition of personalization is given in Equation (2):
> \begin{align}\nonumber
> \min_{\pmb{\Phi}} \frac{1}{N}\sum_{i=1}^N \min_{\pmb{\theta}^i} \mathbb{E}_{s\sim \mu^{i,\pi^i}}\left\|f^i(\pmb{\theta}^i, \pmb{\Phi}(s))-V^{i,\pi^i}(s)\right\|^2.
> \end{align}
>
> Note here that the “min” over $\pmb{\theta}^i$ is inside of the “sum” operator, which means that for each agent i, we are computing the error of its personalized estimate $f^i(\pmb{\theta}^i, \pmb{\Phi})$ to the target value $V^{i, \pi^i}$. However, also note that the “min” over $\pmb{\Phi}$ (the shared feature representation) is outside of the “sum”, representing the fact that $\pmb{\Phi}$ should be a “good” feature representation for all agents.
> We can contrast this with the non-personalized formulation in Equation (1):
> \begin{align}\nonumber
>   \min_{\pmb{\theta}}\frac{1}{N}\sum_{i=1}^N\mathbb{E}_{s\sim \mu^{i, \pi}}  \left\|\pmb{\Phi}(s)\,\pmb{\theta}-V^{i,\pi}(s)\right\|^2,
>  \end{align}
> where the “min” over $\theta$ is taken outside of the “sum”, meaning that this $\theta$ is not personalized.
> Therefore, “successful personalization” can be considered to be achieved when the expected value under a personalized estimate is better than a non-personalized estimate.
>
>
> In Fig 2 (a), we show this via a numerical example. Here, blue can be considered the ground truth value estimated by TD independently in each environment, while the orange bars represent the non-personalized estimates and green bars represent the personalized estimates. Note that the difference |personalized (green) - ground truth (blue)| is smaller than |non-personalized (orange) - ground truth (blue)|, indicating that personalization was successful.
>
> Thank you for pointing this out, we believe we could have been more clear about these definitions in the paper.
>
>
> **Weakness \#1b: thus, no theoretical guarantees that learned personalization (via agent-specific parameters in the paper) captures meaningful environment-specific adaptations. What happens to personalization quality when environments are very different from each other?**
>
> **Response:** We believe this is a misunderstanding. Let us clarify a bit. Our definition of personalization does indeed capture environment-specific adaptations. We refer the reviewer to Equation (2), where the environment-specific value is given by $V^{i, \pi^i}$ (where $i$ denotes the specific environment). Therefore, an agent’s ability to adapt to $V^{i, \pi^i}$ via the agent-specific $\pmb{\theta}^i$ is precisely what we aim to measure. Our theoretical results directly build upon this definition, so we would argue that our theory is capturing a meaningful quantity. Please let us know if you have follow-ups and we would be happy to discuss more.
>
> Certainly, when environments are very different from each other, we expect that it will be harder to learn a good estimate of the environment value functions (since it will be harder to learn a useful feature representation $\pmb{\Phi}$). However, if this happens, it will be precisely measured by the quantity defined in Equation (2): in those situations, we expect to see an increase in the error defined by Equation (2).
>
> At the same time, we would like to point out that when environments are very different, our personalized framework (Equation (2)) will still perform better than the non-personalized framework (Equation (1)) since it allows for agent-specific parameters.

---

> ### Author Response · Authors · 2024-11-21
> **Official Response by Authors (2/4)**
>
> **Weakness \#1c:  ... count N affects personalization ..?**
>
> **Response:** This is an insightful question. We don’t believe there is a clear-cut answer to this, but we definitely believe this is an interesting discussion to include in the paper. First, we note again the definition of personalization:
> \begin{align}\nonumber
> \min_{\pmb{\Phi}} \frac{1}{N}\sum_{i=1}^N \min_{\pmb{\theta}^i} \mathbb{E}_{s\sim \mu^{i,\pi^i}}\left\|f^i(\pmb{\theta}^i, \pmb{\Phi}(s))-V^{i,\pi^i}(s)\right\|^2.
> \end{align}
> Critically, we point out that there is another dimension to this question: environment heterogeneity. In the equation above, this is represented by how different the $V^{i,\pi^i}$'s are across agents. It could go both ways: for a fixed iteration $T$, if we add more agents and the environments are similar, then this presents an opportunity for better learning of the shared feature; but if we add more agents with dissimilar environments, then personalization could be harder.
>
> The natural follow-up question now is how does environment heterogeneity show up in our analysis? We did not clearly emphasize it in the paper, but we refer the reviewer to Assumption 4.3 and Definition 4.5. In Assumption 4.3, higher environment heterogeneity shows up in a larger constant $C$. In Definition 4.5, we define the mixing time $\tau_\delta$ of the system. Note here that we take a maximum over a sequence of environment-specific discrepancy terms, so $\tau_\delta$ also increases as the level of environment heterogeneity increases.
>
> To circle back to your original question, if the level of environment heterogeneity is fixed (i.e., the terms in our analysis remain the same), then as we add more agents $N$, we are able to *improve* personalization.
>
> Thank you for this great question. We plan to add this discussion into the paper's appendix per the reviewer's approval.
>
> **Weakness \#1d: is there a tradeoff ...?**
>
> **Response:** Thanks for your question. First let us clarify that in our paper, ''global performance'' and ''personalization'' are synonymous in that our main objective, Equation (2), accounts for both. This is because the objective is defined as a global average over each agent's personalized performance. Second, the ''speedup'' refers to the speed at which we can learn optimal solutions to the main objective. Therefore, from this perspective,
> we should not claim that there is a tradeoff between ``personalization'' and speedup because they are related quantities.
>
> In our opinion, the correct question to ask is whether there is a tradeoff between ``environment heterogeneity'' and speedup (this is somewhat related to the previous question). Since higher heterogeneity in the environments leads to increased values of certain constants in the analysis, it is clear that there is indeed a tradeoff here. Higher environment heterogeneity leads to a worse speedup, which intuitively makes sense.
>
> We hope this answer addresses your high-level concerns. If we misunderstood your question, please feel free to follow-up.
>
> **Weakness \#2:  ... formulation and approach:
> a. ... privacy preservation...?
> b. ... (1) to (2) preserve ... properties?**
>
> **Response:**  Thank you for your comment.
>
> Our answer is no! Similar as the communication paradigm in standard FedRL (Khodadadian et al., 2022; Dal Fabbro et al., 2023;  Jin et al., 2022), agents only send the gradient of the parameter to the server, not for the trajectories or any other information.  Moreover, we divided the entire model into two parts, and agents only share one representation part $\pmb{\Phi}$ while keeping local personalized head $\pmb{\theta}$ in private. This is even better than existing work in terms of privacy-preservation.
>
> We did not completely understand what the reviewer means by "problem properties". However, let us clarify that Eq. (1) and Eq. (2) can be thought of as loss functions to our algorithm and therefore don't depend on particular assumptions of the underlying problem. We suspect that the reviewer may be wondering if we make assumptions about the underlying environments, which we don't. Therefore, our algorithm applies to a wide range of settings: it can be applied in settings where the environments are completely dissimilar (the algorithm will resort to strong personalization) or to environments with shared structure (the algorithm will automatically discover and exploit this structure through the shared global feature learning). Please also see our response to **Question #1 of Reviewer Um15**, where we were asked a similar question. We hope this answer helps.
>
> The only difference is that the feature $\pmb{\Phi}$ is known and the value function is approximated linearly by $\pmb{\Phi}(s)\pmb{\theta}$ in the conventional formulation in Eq. (1); while $\pmb{\Phi}$ is assumed to be unknown and the value function is approximated by a general function  $f(\pmb{\Phi}(s),\pmb{\theta})$.
>
> If we misunderstood the "problem property", please let us know.

---

> ### Author Response · Authors · 2024-11-21
> **Official Response by Authors (3/4)**
>
> **Weakness \#3a: regarding theoretical rigor
> a. corollary 4.15 ... linear speedup ...**
>
> **Response**: We believe there is a misunderstanding here and would like to respectfully clarify.
> First, let us clarify how speedup is computed in general in the literature. Consider an arbitrary algorithm with convergence rate $\mathcal{O}(1/\sqrt{T})$. To attain $\epsilon$ accuracy for an algorithm, it needs to take $\mathcal{O}(1/\epsilon^2)$ steps. Now consider another algorithm with rate $\mathcal{O}(1/\sqrt{NT})$ (the hidden constant in Big-O is the same), it needs $\mathcal{O}(1/(N\epsilon^2))$ steps to attain $\epsilon$ accuracy. The factor of $N$ is the linear speedup.
>
> Now, let us get back to Corollary 4.15,
> if we let $M\leq \epsilon$, we have the sample complexity of $\mathcal{O}(N^{-1}\epsilon^{-3/2})$, which is $N$ times faster than complexity $\mathcal{O}(\epsilon^{-3/2})$ with one client. This phenomenon is exactly the linear speedup!
>
>
> **Weakness \#3b: b. related to 3.a) above, if the authors claims linear speedup w.r.t agent count N, they should provide comprehensive experimental validation showing how convergence behavior scales with varying numbers of agents. Notable prior work making similar speedup claims, such as Fan et al. [1] which is one of the earliest FedRL works missing from the related work, included thorough ablation studies demonstrating the impact of agent count N on convergence. The absence of such analysis is particularly concerning given the centrality of the linear speedup claim to the paper's contributions.**
>
> **Response:** Thank you for your suggestion. We added additional experimental results to support the provably linear speedup results. We vary the number of agents from 2 to 10 using a specialized experimental setup. Please see the new results in Appendix H2, where we observe that the speedup (convergence time) is almost linearly increasing (decreasing)
> as the number of clients increases.
>
> We thank the reviewer's reminder for bringing [1] to our attention. We have included [1] in our paper and discussed [1] in Section A in the Appendix.
>
>
> **Weakness \#3c:
> c. otherwise, how to determine the optimal number of agents? If we add more agents, will we get faster convergence? what about personalization? intuitively, more agents should increase the personalization complexity.**
>
> **Response:** Thank you for the comment. If all agents operate in identical (or very similar) environments, adding more agents generally leads to faster convergence and improved performance due to the increased availability of collaborative learning data. However, in the case of heterogeneous environments, the relationship between the number of agents, convergence speed, and personalization performance becomes more complex (see our answer above to your previous question about this point). In such scenarios, there is no straightforward correlation, as the added heterogeneity can introduce challenges that may affect both the speed of convergence and the quality of personalization. The impact depends on the extent of diversity among the agents and how effectively the shared representations capture commonalities while allowing for individualized adaptations.

---

> ### Author Response · Authors · 2024-11-21
> **Official Response by Authors (4/4)**
>
> **Weakness \#4: regarding experimental evaluation.
> a. Limited diversity in test environments (only classic control tasks) and no statistical significance is assessed
> b. how to empirically verify the personalization achieved?
> c. ablation on agent count N should be conducted.**
>
> **Response:** Thank you for your suggestions.
>
> First, we report the statistical significances in Tables 3 and 4 in Appendix H6. Specifically, we report the return average, variance average, return median and total running time for 10 environments for Cartpole and Acrobot environments. By comparing with DQN algorithm without personalization, we can validate the linear speedup in running time. Among all algorithms, our PFedDQN-Rep achieves the best return average and median, with top variance and running time, as summarized in Tables 3 and 4. We also provide a zoom-in shortened plot for both environments to show the quick adaptation speed when sharing representations as in Figure 16. All results and discussions are highlighted in blue.
>
>
> Second, to validate that the personalization is reached, we first compute the cosine similarity matrix of transition probabilities in both environments. After the algorithm converges, we compute the cosine similarity matrix of policy layer (last layer) in the neuron network in Appendix H5. We notice that while the nearby agents might share similarity in their policy, personalization is reached corresponds to their transition probabilities. The shared representation layer stays identical. All results and discussions are highlighted in blue.

---

> ### Author Response · Authors · 2024-11-25
> **Additional Feedback?**
>
> Dear reviewer,
>
> Since the discussion period is almost over, we would to politely check if our response has addressed your concerns & questions. If you have additional feedback, please let us know. We've also posted a new revision of the paper with additional experiments and summarized the changes in a comment at the top of OpenReview.
>
> Thanks again for your valuable feedback.
>
> Authors

---

> > ### Comment · Reviewer_mFXK · 2024-11-25
> >
> > Thank the authors for the detailed revisions and comprehensive response. The revised manuscript is indeed clearer in presenting its theoretical contributions, especially the formal definition of successful personalization and the theoretical guarantees regarding environment-specific adaptations. These additions significantly enhance the paper’s clarity and scholarly contribution.
> >
> > However, I have several remaining concerns:
> >
> > 1. Ablation on Agent Count $N$: The analysis regarding how the number of agents $N$ affects both personalization quality and convergence speed is still limited. While the theoretical results highlight the linear speedup claim, comprehensive experimental ablations are missing in the main text. It would be highly beneficial for the paper to include:
> > - A detailed empirical evaluation of how increasing $N$ impacts personalization complexity and convergence speed.
> > - Specific studies in heterogeneous environments where $N$ introduces varying levels of agent-environment discrepancy.
> > Incorporating such experiments in the main body (maybe in future work) would provide stronger empirical validation and align with precedent works in the FedRL literature.
> >
> > 2. Tradeoff Between Personalization and Global Performance/Speedup: The rebuttal suggests that “global performance” and “personalization” are treated synonymously within the objective, which may constrain the broader appeal and practical relevance of the work. However, since the convergence speed in the proposed framework is tied to the averaging of parameters via the central server, global performance cannot be entirely equivalent to personalization. In practice, an increased focus on personalization is likely to impact the global model's effectiveness or convergence properties. For instance, prioritizing personalized adaptations for highly heterogeneous environments might introduce conflicts with the shared representation’s utility for all agents.
> > - To address this, the manuscript would benefit from a more nuanced discussion of the potential tradeoffs between personalization and global performance, especially in the presence of high environment heterogeneity
> >
> > 3. Clarity on Linear Speedup Claim: While Corollary 4.15 asserts linear speedup, the rebuttal clarifies that this is achieved by proportionally reducing $T$ (communication rounds). This is different from a conventional linear speedup (as in the FedRL papers) where adding agents inherently improves the convergence rate without requiring adjustments to $T$.
> > - In conventional FedRL papers (such as those you referenced), linear speedup means that increasing N (number of agents) directly improves the convergence rate. This improvement comes "for free" - you don't need to adjust other parameters
> >   - If you double $N$, you roughly halve the time to convergence, all else being equal
> >   - What's actually happening in Corollary 4.15 is putting a constraint on $T$ w.r.t $N$,
> >   - which means you can't freely increase $N$ without also adjusting $T$
> >   - The "speedup" isn't purely from parallelization, as claimed
> > - Please correct me if I made further misunderstanding. Otherwise, this is a significant oversight in the interpretation of the results. While the mathematical bounds themselves may be correct, the interpretation as a "linear speedup" is potentially misleading as it suggests a simpler and more favourable scaling than what's actually achieved. Alternatively, providing additional context and explicit comparisons with prior FedRL works would help avoid potential misunderstandings.
> >
> > These points, while critical, are primarily focused on improving the completeness and robustness of the paper. I appreciate the authors' efforts in addressing my earlier comments, and I am inclined to revise my scores upon further discussion with the other reviewers, especially if the authors plan to incorporate the suggested experiments and analyses into the final version.

---

> ### Author Response · Authors · 2024-11-27
> **Personalization quality tradeoff**
>
> Thank you for this comment! We've thought through this carefully and believe that you make a great point. We've now defined a new metric to capture personalization quality, the "worst case personalization error among N agents." In other words, rather than averaging over N agents' performance, we also examine the maximum error over the N agents:
>
> $$\max_{i \in [N]}  \mathbb{E}_{s\sim \mu^{i,\pi^i}}\left\|f^i(\pmb{\theta}^i, \pmb{\Phi}(s))-V^{i,\pi^i}(s)\right\|^2.$$
>
> This would be a measure of personalization quality. The intuition behind this metric is that if all agents achieve good estimation error, then this metric is small (meaning we have personalized well), but if some agents perform poorly while others perform well, then this metric will be large (detecting that we did not personalize well).
>
> In **Appendix H3**, we have added a new experiment to understand the tradeoff between computation time and personalization quality. The important takeaways are reproduced below, but please refer to the revision for the figure and more details:
> * We notice that for naive DQN, we can achieve no personalization error at the cost of high computation.
> * At the other end of the spectrum, FedDQN leverages parallelization and reduces the computation, but has high personalization error.
> * Finally, our algorithm, PFedDQN-Rep achieves the best of both worlds: low computation, while attaining low personalization error.
>
> We believe this is a great addition to the paper---thank you for your comments which made us realize this fact.

---

> > ### Comment · Reviewer_mFXK · 2024-11-28
> > **response to comment regarding Personalization quality tradeoff**
> >
> > I thank the authors for this thoughtful addition that addresses personalization quality. The new worst-case metric provides valuable insight into personalization performance, and I look forward to seeing these discussions in the final version.

---

> > > ### Comment · Reviewer_mFXK · 2024-11-28
> > >
> > > I thank the authors for their detailed response. I have revised my evaluation and updated the scores. Good luck.

---

> > > > ### Author Response · Authors · 2024-11-29
> > > > **Thank you!**
> > > >
> > > > We are happy to hear that we have adequately addressed all your concerns. Thank you for your acknowledgement and raising the rating of our paper. Much appreciated!

---

> ### Author Response · Authors · 2024-11-27
> **Ablation on Agent Count**
>
> Thank you for the comment.
> * **Agent count vs convergence time.** In **Appendix H2**, we have added an experiment to validate the linear speedup as in our theorems. Note that because there is the added dimension of environment heterogeneity, we designed a special experiment that holds environment heterogeneity constant as we increase the number of agents. This is achieved by duplicating a set of 2 environments 2, 3, 4, and 5 times, thereby obtaining situations with N=2, 4, 6, 8, 10. Please see Appendix H2 for the detailed plot and discussion. We were able to verify a nearly linear speedup (but note that in practice, there is certain overhead that prevents the speedup to be as efficient as predicted by t heory). We would be happy to move this to the main text if the reviewer prefers.
>
> * **Agent count vs personalization.** As discussed above, in **Appendix H3**, we present a new experiment to explore the tradeoff between agent count and personalization. We now report the “worst case personalization error among N agents” to properly measure how personalization quality may degrade as we increase the number of agents.
>
> * **Environment heterogeneity/discrepancy vs personalization.** In **Appendix H4**, we examine how environment discrepancy affects personalization quality, measured in terms of worst case personalization error among N agents. We fix the number of agents to be 10. While keeping the average pole length fixed, we adjust the discrepancy in pole length between environments, which allows us to see the effect of environment heterogeneity on our results. We observe that while all algorithms show degradation in personalization quality as environment heterogeneity increases (as expected), our approach degrades the smallest amount.

---

> > ### Author Response · Authors · 2024-11-27
> > **Clarity on the linear speedup claim**
> >
> > Thank you for following up! Let us give some more clarity. Our computation of linear speedup is quite straightforward: we simply compute the # of iterations it takes to reach a certain solution quality. Mathematically, this means we compute the $T(\epsilon)$ (# of iterations) such that the error falls below a threshold ($\epsilon$). In other words, $T(\epsilon)$ is the time to convergence, where “convergence” is defined by $\epsilon$.
> >
> > We believe that this matches the reviewer’s intuition exactly (“if you double $N$, you roughly halve the time to convergence”)! Our result says precisely this.
> >
> > (Note that we are not “constraining” $T$ in any sense; the result is achieved by some simple algebraic manipulations. Therefore, the speedup is indeed purely from parallelization. We will clarify the writing to avoid this potential misinterpretation.)
> >
> > A few comments:
> > 1. Many papers in the FedRL literature give the same style of results:
> >   * Please see Theorem 4.1 of [1] (link below), a well-known paper in FedRL. They state “We achieve $\mathbf{E}[error] \le \epsilon$ within $T = O(1/(N \epsilon))$ iterations,” exactly the same logic as ours.
> >   * Please see Theorem 3.1 of [2] (link below), another well-known paper in FedRL. They say “Theorem 3.1 suggests that to achieve an $\epsilon$-accurate Q-function estimate in an $l_\infty$ sense, the number of samples required at each agent is no more than $\tilde{O}(|S| * |A| / (K (1-\gamma)^5 \epsilon^2))$.” In this paper, $K$ is the number of agents. This is also exactly the same logic that we use.
> >
> > 2. We believe the reviewer may be referring to some papers which provide results of the form “Algorithm error $\le O(1/(NT))$”. In these papers, this is directly claimed to be “linear speedup.” As an example, please see Corollary 2.2 of [3] or Theorem 2 of [4].
> > In these papers, they are not explicit about *why* this is a linear speedup. But if we look carefully, it is also the same logic:
> > To reach epsilon in Algorithm error, we need $1/(NT) \le \epsilon$, which implies that we need $T \ge 1/(N \epsilon)$ to reach convergence. This is the same logic as us and also papers [1] and [2]!
> >
> > 3. The natural next question is “why do our results not look as straightforward as papers [3] and [4]?” This is because our setting is more difficult (Markovian noise + heterogeneous environments), so our result (Corollary 4.15) has more complex terms ($N^{2/3}$ and $T^{2/3}$) than papers [3] and [4]. But once we do the algebraic manipulation, it turns out we achieve the same linear speedup in terms of time to convergence!
> >
> > 4. We plan to clarify all of the above in the final version of the paper. Indeed, it deserves some more explanation. Thanks to the reviewer for pointing it out!
> >
> > References:
> >
> > [1] https://proceedings.mlr.press/v162/khodadadian22a/khodadadian22a.pdf
> >
> > [2] https://proceedings.mlr.press/v202/woo23a/woo23a.pdf
> >
> > [3] https://arxiv.org/pdf/2401.15273
> >
> > [4] https://arxiv.org/pdf/2302.02212

---

> > > ### Comment · Reviewer_mFXK · 2024-11-28
> > >
> > > I appreciate the detailed explanation and this clarification significantly changes my previous assessment. My previous critique was off-base because I misinterpreted $T^2 > N$ as a limiting constraint rather than a condition in the analysis.

---

### Official Review · Reviewer_LerK · 2024-11-04

**Soundness:** 2
**Presentation:** 3
**Contribution:** 3
**Rating:** 8
**Confidence:** 3

**Summary:**

This paper introduces a personalized federated reinforcement learning (FedRL) framework, PFEDRL-REP, that incorporates shared representations to improve learning in heterogeneous environments. PFEDRL-REP collaboratively learns a shared feature representation among agents while maintaining agent-specific weight vectors for personalization. The authors analyze PFEDTD-REP, a variant using temporal difference learning, and prove it achieves a linear convergence speedup in terms of the number of agents, demonstrating scalability benefits. Experiments in both policy evaluation and control settings show that PFEDTD-REP enhances convergence and generalization in heterogeneous environments compared to non-personalized FedRL methods.

**Strengths:**

1. PFEDRL-REP is an innovative approach to FedRL, addressing a major challenge in heterogeneous environments by introducing a shared representation while allowing for agent-level personalization.
2. The paper provides a rigorous theoretical foundation, including proofs of convergence speedup under Markovian noise using a two-timescale stochastic approximation framework.
3. The paper is well-structured and clear, with detailed explanations of the problem formulation, the PFEDRL-REP framework, and the two-timescale convergence analysis.

**Weaknesses:**

1. The experimental evaluation could be extended to include more complex environments, such as those with sparse rewards or high-dimensional state spaces, to better assess the scalability of PFEDRL-REP.
2. The applicability of PFEDRL-REP to all types of environmental heterogeneity is not fully guaranteed, as the combination of shared feature representations and personalized weight vectors may not capture all nuances of diverse environments.

**Questions:**

1. Is PFEDRL-REP universally applicable across diverse heterogeneous federated RL problems? What if a shared feature representation could not represent the similarity between different environments and the heterogeneity could not be distinguished by the agent-specific weight vector?
2. Does PFEDDQN-REP maintain strong performance on more complex RL tasks, such as those with sparse rewards or high-dimensional state spaces? Additional experimental results on more challenging environments would provide valuable insights.

---

> ### Author Response · Authors · 2024-11-21
> **Official Response by Authors (1/2)**
>
> Thank you very much for your review and constructive comments, as well as giving the positive rating of our work. Here we would like to address the reviewer's concerns and hope that can help raise the rating of our paper.
>
> **Weakness \#1:  The experimental evaluation could be extended to include more complex environments, such as those with sparse rewards or high-dimensional state spaces, to better assess the scalability of PFEDRL-REP.**
>
>
> **Response:** Thank you for your valuable feedback regarding the experimental evaluation.
>
> Our current experiments were designed to clearly illustrate the benefits of PFedRL-Rep in environments that highlight its ability to handle personalization and shared representations among heterogeneous agents. This includes showing improvements over baseline methods and validating theoretical findings through controlled setups. We agree that extending the experiments to more complex environments would provide deeper insights into the robustness and scalability of PFedRL-Rep. In environments with sparse rewards, for example, the ability to learn shared representations could potentially enhance exploration by pooling agent experiences. Similarly, for high-dimensional state spaces, our framework's ability to learn a common low-dimensional feature representation could mitigate the complexity and improve convergence.  We are the first to present this novel framework for personalized federated reinforcement learning via leveraging shared representations, which has the potential to handle a wider range of scenarios.
>
>
> Per the reviewer's suggestion, we conduct experiments on another enviroment named Hopper from gym, whose state and action space are both continuous. We vary the length of legs to be $0.02 + 0.001*i$, where $i$ is the i-th indexed agent, while keeping the same parameters such as healthy reward, forward reward and ctrl cost (l2 cost function to penalize large actions). We increase the number of agents to 20, and plot the return with respect to frames. We generate a new sampled transition to validate the generalization nature of the algorithms. In order to fit the algorithm to continuous setting, we modified the proposed algorithm to a DDPG based algorithm, similar to any DQN related benchmarks. For FedQ-K, LFRL and FedAsynQ-Imavg, we discretize the state and action space. Similar to Cartpole and Acrobot environment, our proposed PFedDDPG-Rep achieves the best reward and generalize to new environments quickly as in Appendix H1.

---

> ### Author Response · Authors · 2024-11-21
> **Official Response by Authors (2/2)**
>
> **Weakness \#2: The applicability of PFEDRL-REP to all types of environmental heterogeneity is not fully guaranteed, as the combination of shared feature representations and personalized weight vectors may not capture all nuances of diverse environments.**
>
>
>
>
> **Response:** Thank you for your thoughtful observation regarding the applicability of PFedRL-Rep to diverse forms of environmental heterogeneity. It is important to clarify that the motivation of this work is not to address scenarios where each agent's environment is significantly different but rather to target settings where agents share a substantial amount of common structure. This focus is well-aligned with the motivations described in lines 42-50 and the illustrative motivating examples provided in Figure 6 of our paper. Also note that this is also the motivation of the study of federated reinforcement learning (FedRL) as in many existing works in this area (e.g., some are summarized in Table 1 and discussed in Appendix A). The agents can face heterogeneous environments in FedRL but should benefit the learning from collaborations in FedRL.
> We can take the view that if the agent is unaware of the level of heterogeneity in the environment, the algorithm will automatically adjust itself: if there's dramatic differences, the algorithm will focus on personalization (reducing to the independent case); if the environments are similar, the algorithm will automatically learn a useful feature.
> The critical point is that we don't make any specific assumption on a shared structure in the environments.
>
>
>
>
> From a theoretical perspective, some related works have discussed the necessity of a bounded divergence assumption for environments, both for FedRL setting (Jin et al., 2022) and the widely studied federated learning (a supervised learning framework) settings (Collins et al., 2022, Xiong et al, 2024). In our framework, we assume a bounded mixing time (Assumption 4.3, Lemma 4.12), which inherently limits the extent of heterogeneity among the agents' environments. This ensures that the environments are not arbitrarily different, allowing the shared feature representations and personalized weight vectors in our PFedRL-Rep framework to effectively capture both commonalities and individual nuances across agents.

---

> ### Author Response · Authors · 2024-11-25
> **Additional Feedback?**
>
> Dear reviewer,
>
> Since the discussion period is almost over, we would to politely check if our response has addressed your concerns & questions. If you have additional feedback, please let us know. We've also posted a new revision of the paper with additional experiments and summarized the changes in a comment at the top of OpenReview.
>
> Thanks again for your valuable feedback.
>
> Authors

---

> > ### Comment · Reviewer_LerK · 2024-12-03
> >
> > Dear authors,
> >
> > Thank you for your detailed explanation and efforts in the new experiment. My concerns are addressed, and I have raised my score to 8. Good luck!

---

> > > ### Author Response · Authors · 2024-12-03
> > > **Thank you!**
> > >
> > > We are happy to hear that we have adequately addressed all your concerns. Thank you for your acknowledgement and raising the rating of our paper. Much appreciated!

---

### Official Review · Reviewer_Um15 · 2024-11-05

**Soundness:** 3
**Presentation:** 3
**Contribution:** 2
**Rating:** 6
**Confidence:** 2

**Summary:**

This work propose a Personalized FedRL approach (similar to PFL but with RL) allowing local/per-agent learnable parameters for use in heterogeneous FedRL settings with convergence results. The authors find a linear relationship between the number of agent and the convergence timestep.

**Strengths:**

- The theoretical analysis have a proof sketch and the assumptions are clearly stated.
- I think the two timescale approximation result is novel
- Interesting results from cliff-walking and cartpole

**Weaknesses:**

- Although the setup hold promise, evaluation is rather on the simple side. I consider it understandable for now since the main focus in on theory.
- If I understand correctly, the linear speed up is not a particular exciting result, since sample collection in unit time grows linearly with N.
- There is no comparison with other PFL methods, or parameter-sharing MARL methods.

**Questions:**

Is the problem definition of PFedRL includes "shared common structure", how does it affects learning? or anything else in the theory? In theory they can be completely different problem without any shared structure and the results still stand?

---

> ### Author Response · Authors · 2024-11-21
> **Official Response by Authors (1/2)**
>
> Thank you very much for your review and constructive comments, as well as giving the positive rating of our work. Here we would like to address the reviewer's concerns and hope that can help raise the rating of our paper.
>
> **Weakness \#1:  If I understand correctly, the linear speed up is not a particular exciting result, since sample collection in unit time grows linearly with N.**
>
> **Response:** We appreciate the reviewer's question. We would like to respectfully clarify our perspective on this point. The reviewer is correct that linear speedup corresponds to the linear growth in sample collection with $N$. However, it is worth noting the practical advantage that sample collection is now parallelized across $N$. Intuitively, the linear speedup result says that we can parallelize without significant loss in solution quality, and hence is highly desirable since one can efficiently leverage the massive parallelism in large-scale decentralized systems.
>
> Perhaps more importantly, achieving linear convergence speedup is widely recognized as a significant technical contribution in the field of federated learning (FL), both in conventional FL (in supervised learning settings) and in the more recent and increasingly studied domain of federated reinforcement learning (FedRL). This is highlighted by numerous existing works (e.g., some listed in Table 1). To maintain consistency with this line of research and to provide a fair comparison with these well-established baselines, we also focus on demonstrating linear convergence speedup in our newly proposed personalized federated reinforcement learning (PFedRL) framework. We hope that this clarification helps.
>
> Secondly, we would like to point out that proving linear speedup in the PFedRL setting is more theoretically challenging than the conventional FL case, requiring the
> handling of Markovian dynamics, which ensures convergence in the presence of non-stationary data generated from agents' interactions with their (heterogeneous) environments. In addition, the call for personalization in PFedRL introduces an additional layer of complexity. Although Jin et al. 2022 proposed a heuristic personalized FedRL algorithm, its theoretical performance guarantee remains unknown. Our theoretical contribution in PFedRL navigates these unique challenges by establishing convergence guarantees in settings characterized by Markovian noise and the need for personalization.

---

> ### Author Response · Authors · 2024-11-21
> **Official Response by Authors (2/2)**
>
> **Weakness \#2:  There is no comparison with other PFL methods, or parameter-sharing MARL methods.**
>
> **Response:** First, as highlighted in Table 1, the most recent works in federated reinforcement learning (FedRL) with some theoretical performance guarantees do not focus on personalization within the FedRL context. In our evaluations, in particular the applications to control problem (Section 5), we did include comparisons with two heuristic personalized federated reinforcement learning methods, PerDQNAvg (Jin et al., 2022) and FedAsynQ-ImAvg (Woo et al., 2023), which incorporate elements of personalization in their designs. Our experiments demonstrate the superior performance of our PFedRL-Rep framework, particularly in its ability to provide personalized learning while leveraging shared representations for heterogeneous environments. In addition, this paper considers the personalized federated RL (PFedRL) setting, rather than the personalized federated learning (PFL) setting (a supervised learning framework), and hence we did not compare with the large body of works in PFL. We discussed the related works in PFL in Appendix A.
>
> Second, FedRL and MARL represent fundamentally different frameworks with distinct objectives and structures. FedRL focuses on collaborative learning across decentralized agents without sharing local trajectories, prioritizing privacy and decentralized data aggregation. In contrast, MARL typically involves agents interacting within a shared environment, emphasizing coordination and competition between agents. Given these divergent goals and settings, a direct comparison between FedRL and MARL methods would be inherently unfair and not reflective of their respective aims or challenges, and hence were not considered in the experiments.
>
> **Question \#1: Is the problem definition of PFedRL includes "shared common structure", how does it affects learning? or anything else in the theory? In theory they can be completely different problem without any shared structure and the results still stand?**
>
>
> **Response:** Thank you for this insightful question.
>
> No, we don't make any explicit assumption that there's shared common structure between the environments, which makes our algorithm applicable in a range of settings. If there is shared common structure, then our learning algorithm can automatically discover it and exploit it. If there is absolutely no relationship between the environments, then our learning algorithm will resort to more personalization. In either case, we expect that our algorithm will work.
> In particular, the similarity level of environments will affect the mixing time (see Assumption 4.3, Definition 4.5, and Lemma 4.12) of the entire system, which has a significant impact on the convergence speed.
>
> If we misunderstood this question, please let us know.

---

> ### Author Response · Authors · 2024-11-25
> **Additional Feedback?**
>
> Dear reviewer,
>
> Since the discussion period is almost over, we would to politely check if our response has addressed your concerns & questions. If you have additional feedback, please let us know. We've also posted a new revision of the paper with additional experiments and summarized the changes in a comment at the top of OpenReview.
>
> Thanks again for your valuable feedback.
>
> Authors

---

### Author Response · Authors · 2024-11-25
**Revision posted**

Dear reviewers,

Thanks for the valuable feedback. We wanted to make sure you are aware that we've uploaded a revision of the paper before the end of the discussion phase. The new results are in blue and located in Appendix H:

* **More complex environment.** We've added a more complex environment (Hopper), under the recommendation of Reviewer LerK. Hopper has continuous state and action spaces, so we adapted our personalized FedRL framework to DDPG, resulting in a new instantiation of the algorithm PFedDDPG, which performs well compared to baselines (See Appendix H1 in the revision). In addition, this result illustrates that the PFedRL framework can be used quite broadly (TD, DQN, DDPG).

* **Linear speedup wrt to agent count N.** As suggested by Reviewer mFXK, we've conducted an experiment varying the number of agents from 2 through 10 to verify our theoretical results. Indeed, we see that the convergence time decreases nearly linearly as the number of agents increases. Please see Appendix H2.

Thanks again!

---

### Author Response · Authors · 2024-11-27
**Another revision posted**

Dear reviewers,

Given our discussion with Reviewer mFXK, we have posted another revision of the paper. Please refer to the latest revision because we have formatted the new results to be easier to read (on separate pages and different sections). Our revision includes several new experiment results:
* A more complex environment (Hopper) in **Appendix H1**
* Verifying the linear speedup theoretical result in **Appendix H2**
* Examining the tradeoff between computation and personalization quality in **Appendix H3**
* Examining the effect of environment discrepancy on personalization error in **Appendix H4**

As always, thanks for your comments and please follow up if we can answer or clarify additional points as the discussion period winds down soon.

---

### Meta-Review · Area_Chair_JhP1 · 2024-12-21

**Metareview:**

This paper studies personalized federated reinforcement learning (FedRL) with shared representations, presenting theoretical convergence results and demonstrating linear speedup in the proposed framework, PFedRL-Rep, under Markovian noise. The results are supported by rigorous theoretical analysis and experimental validation, including performance improvements in heterogeneous environments. The reviewers highlighted the novelty and soundness of the theoretical contributions, particularly in establishing linear speedup and addressing personalization in FedRL. During the rebuttal phase, the authors effectively addressed concerns raised by the reviewers, providing additional experimental ablations and clarified theoretical guarantees. Overall, the reviewers reached a positive consensus, and the paper makes a substantial theoretical contribution to the field, making it acceptable for publication.

**Additional Comments On Reviewer Discussion:**

During the rebuttal period, reviewers raised several points regarding theoretical clarity, experimental validation, and personalization metrics. Reviewer Um15 sought better explanation of how shared structure affects learning and comparisons with related methods, which the authors addressed by clarifying theoretical contributions and including new comparisons in the appendix. Reviewer LerK suggested expanding experimental evaluations to more complex environments, which was addressed by adding results on the Hopper environment and discussing generalization to heterogeneous settings. Reviewer mFXK requested a formal definition of personalization, clarification on the linear speedup claim, and additional ablation studies on agent count and personalization quality. The authors added a "worst-case personalization error" metric, detailed experiments validating speedup, and an analysis of tradeoffs between agent count and personalization. Each concern was thoroughly addressed, leading to increased reviewer confidence.

---

### Decision · Program_Chairs · 2025-01-22

Accept (Poster)